# Learning Steadily: Accumulating Relative Point Margin Scores for Face Image Quality Assessment

## Abstract

Face Image Quality Assessment (FIQA) is essential for reliable biometric systems, as it determines whether captured face images are suitable for automated recognition tasks. Current state-of-the-art FIQA approaches that integrate with face recognition (FR) training often suffer from unstable quality targets due to the evolving feature space during training. We introduce *CARPM-FIQA*, a novel approach that addresses this limitation by learning to predict accumulated relative point margin scores across training epochs rather than relying on single-epoch estimates. Our method quantifies image quality by tracking and accumulating the ratio between intra-class compactness and inter-class separation for each sample throughout training, providing a temporally-stable measure of sample utility. We demonstrate theoretically and empirically that this cumulative averaging approach significantly reduces variance in quality estimates and improves reliability in sample ranking compared to non-cumulative alternatives. To enable quality assessment for unseen images, we extend standard FR architectures with a regression layer that predicts these accumulated values. Through extensive evaluation on eight challenging benchmarks (LFW, AgeDB-30, CFP-FP, CALFW, Adience, CPLFW, XQLFW, and IJB-C) and across four state-of-the-art FR models, we show that *CARPM-FIQA* consistently ranks among the top-performing methods, with particular strength on datasets with quality variations. Our approach offers a more robust alternative to existing FIQA techniques while maintaining tight integration with the FR pipeline.

## 1 Introduction

FIQA evaluates face image utility for face recognition (FR) processing, i.e. *recognition utility* or *utility for identity recognition* ISO/IEC JTC 1/SC 37 Biometrics (2024); Grother et al. (Sep. 2021). The goal of FIQA is not to judge an image based on its visual appeal (e.g., sharpness, noise), but rather to estimate how useful a face image is for face verification—such as deciding whether it should be accepted from a live capture or passport scan for identity matching, which ensures reliable and accurate FR Grother et al. (Sep. 2021); Schlett et al. (2022). Unlike IQA methods that assess quality from human perception Mittal et al. (2012; 2013); Liu et al. (2017b), FIQA specifically evaluates face image suitability for automated recognition. As demonstrated in Fu et al. (2022), high perceived quality does not always correlate with FR utility, particularly when factors like facial occlusions are present. This distinction explains why FIQA approaches consistently outperform general IQA methods for FR applications, as confirmed by multiple studies Boutros et al. (2023); Meng et al. (2021a); Ou et al. (2021); Babnik et al. (2023a).

Existing FIQA approaches fall into four groups: (1) Label-generation approaches Hernandez-Ortega et al. (2020); Ou et al. (2021); Best-Rowden & Jain (2018) train regression networks using quality labels from human assessments Best-Rowden & Jain (2018), ICAO-compliance comparisons Hernandez-Ortega et al. (2020), or distribution distances Ou et al. (2021). RankIQ Chen et al. (2015) uses learning-to-rank to predict quality rankings via FR performance metrics across datasets, better capturing relative differences. A notable limitation of these approaches is that they often decouple FIQA from its primary application (FR), as they typically employ shallower networks that don't extract comprehensive facial features. (2) Non-FR model approaches include DifFIQA Babnik et al. (2023b), which uses diffusion models to explore embedding stability, and eDifFIQA Babnik et al. (2024), which applies knowledge distillation for efficiency. One of the main limitations of this approach is the extremely high computational cost induced by the base model, e.g., the diffusion model Babnik et al. (2023b). (3) Pre-trained FR analysis approaches Terhörst et al. (2020); Kolf et al. (2024) include SER-FIQ Terhörst et al. (2020), which evaluates embedding consistency with varied dropout patterns, and GraFIQs Kolf et al. (2024), which uses gradient magnitude during

backpropagation. (4) FR-integrated approaches Boutros et al. (2023); Meng et al. (2021a); Shi & Jain (2019) include MagFace Meng et al. (2021a), using embedding magnitude as quality, and PFE Shi & Jain (2019), modeling face embeddings as Gaussian distributions with uncertainty indicating quality. CR-FIQA Boutros et al. (2023) estimates quality by predicting a sample's relative classifiability based on its position in the embedding space. The advantage of these approaches is their direct coupling with the FR application, creating a more cohesive quality assessment framework. Approaches Meng et al. (2021a); Boutros et al. (2023) in this category led to SOTA performances in the main challenging benchmarks Yang et al. (2025). Our *CARPM-FIQA*, as we present through the paper, belongs to the methods in the fourth group.

Despite recent advancements in FIQA, especially in the approaches in the fourth group (FR-training-required methods), challenges remain. Such methods can suffer from unstable quality estimates due to the dynamic nature of the training process. As the feature space evolves, the induced clustering structure changes, leading to fluctuations in quality estimates over training iterations, which will be analyzed later in this work. This instability makes it difficult to consistently identify high-quality face images that remain distinguishable throughout training, creating a moving target. To address this challenge, we propose *CARPM-FIQA*, which incorporates three key innovations:

1. A quality stabilization approach accumulating relative point margin values—measuring the separation of samples and competing identities—across training epochs, providing more robust quality estimates, capturing long-term discriminative properties of face images.
2. A theoretical framework showing accumulation reduces variance and increases stability in quality estimation compared to single-epoch measurements, with empirical validation.
3. A regression model, based on the presented approach, that effectively predicts accumulated quality scores for unseen face images, generalizing the temporal stability patterns observed during training to new samples without requiring access to their training history.

Our extensive experiments across multiple benchmarks demonstrate that *CARPM-FIQA* provides consistent, stable quality assessments, generalizing well across different FR models, ranking among the top three state-of-the-art methods, effectively addressing the instability issues present in existing FR-integrated FIQA approaches while maintaining the advantage of tight coupling with the FR task.

## 2 METHODOLOGY

This section presents our proposed *CARPM-FIQA* approach that leverages quality training targets accumulated over multiple training epochs to achieve stable FIQA. Unlike methods that rely on single-epoch measurments Meng et al. (2021a); Shi & Jain (2019); Boutros et al. (2023), our approach aggregates relative point margin values—measuring temporally, providing a more robust estimate of their intrinsic quality.

**(Background) Silhouette Scores and Class Separability:** The silhouette score Rousseeuw (1987) evaluates the quality of clustering results Lovmar et al. (2005); Januzaj et al. (2023); Ogbuabor & Ugwoke (2018); Shahapure & Nicholas (2020). It quantifies how well a data point is assigned to its cluster by considering both intra-cluster cohesion and inter-cluster separation. The score provides an interpretable measure of clustering validity. Since our task involves classification rather than clustering, we use the terms class instead of cluster and class-belonging/classification instead of clustering from now on. Formally, the silhouette score for a given sample $i$ is defined as:

$$s(i) = \frac{b(i) - a(i)}{\max(a(i), b(i))},$$
(1)

where $a(i)$ represents the average distance between the sample $i$ and all other points within the same class, indicating intra-class compactness. The term $b(i)$ denotes the smallest average distance between $i$ and all points in the nearest different class, reflecting inter-class separation. $s(i) \in [-1, +1]$ is the silhouette score. The interpretation of the silhouette score per sample is straightforward: values close to $+1$ indicate that the sample is well-classified and far from other classes, values around $0$ suggest that the sample lies near the decision boundary, and negative values imply that the sample may have been misclassified, as it is closer to a different class than its own.

In FR models, the classification structure of deep feature representations is influenced by losses that explicitly encourage intra-class compactness and inter-class separation. One such family of loss functions is the Angular Margin Penalty-based Softmax Loss, widely used for FR models Deng et al. (2019); Huang et al. (2020); Meng et al. (2021a); Boutros et al. (2022); Wang et al. (2018); Liu et al. (2017a); Meng et al. (2021b). Such loss modifies the standard softmax loss, a cross-entropy loss over a softmax layer, by introducing an angular margin penalty between the deep features and their corresponding class centers. In this work, we take the ArcFace loss Deng et al. (2019) as an

example, a specific instantiation of the angular margin penalty-based softmax. The ArcFace loss is defined as:

$$\mathcal{L}_{Arc} = \frac{1}{N} \sum_{i \in N} - \log \frac{e^{s(\cos(\theta_{y_i} + m))}}{e^{s(\cos(\theta_{y_i} + m))} + \sum_{j=1, j \neq y_i}^{C} e^{s(\cos(\theta_j))}}, \tag{2}$$

where $N$ is the batch size, $C$ is the number of classes, and $y_i$ is the class label for sample $i$. The angle $\theta_{y_i}$ is defined as the angle between the feature vector $x_i$ and the corresponding class center $w_{y_i}$. The feature vector $x_i$ is the deep embedding from the last fully connected layer, and the weight vector $w_{y_i}$ corresponds to the class center. The margin $m > 0$ is the additive angular margin and $s$ is the scaling parameter Wang et al. (2018).

The class centers $w_{y_i}$ are always accessible, as they are stored in the weight matrix of the classification layer by design. Instead of explicitly computing the average distance between a sample $i$ and all other points within the same class, we can directly compute the distance between the sample $i$ and its corresponding class center $w_{y_i}$. This provides an efficient way to measure intra-class cohesion, as the class center represents the mean feature vector of the corresponding identity class. We redefine the intra-class distance $a(i)$ and the inter-class distance $b(i)$ using the cosine similarity as follows:

$$a(i) := \cos(\theta_{y_i}). \tag{3}$$

The cosine similarity between the deep feature $x_i$ and the class center $W_{y_i}$ is computed using the formula $\cos(\theta_{y_i}) = x_i \cdot W_{y_i}$, where $\cdot$ denotes the dot product between the two vectors, $x_i$ is the deep feature of sample $i$, and $W_{y_i}$ is the corresponding class center. Similarly, the inter-class distance $b(i)$ is redefined as the largest similarity between $x_i$ and any class center $W_{y \neq y_i}$:

$$b(i) := \max(\cos(\theta_{y \neq y_i})) = \max_{y \neq y_i} \cos(\theta_y). \tag{4}$$

This value is similarly computed by finding the maximum cosine similarity between the sample and the negative class centers in the embedding space, allowing to effectively compute the silhouette scores per sample using the class centers, leveraging the optimization by the loss.

**(Background) Relative Point Margin for Face Quality Assessment:** With the reformulation of the silhouette score in Equation 1, we now analyze its behavior under different cases for the intra-class $a(i)$ and inter-class distances $b(i)$. The silhouette score $s(i)$ for a given sample $i$ can be written as a single equation that accounts for both distances:

$$s(i) = \frac{b(i) - a(i)}{\max(a(i), b(i))} = \begin{cases} 1 - \frac{a(i)}{b(i)}, & \text{if } a(i) < b(i) \\ \frac{b(i)}{a(i)} - 1, & \text{if } a(i) > b(i) \\ 0, & \text{if } a(i) = b(i) \end{cases}. \tag{5}$$

From these cases, we observe that the information obtained from the silhouette score is derived from the ratio of intra-class and inter-class distances, i.e. $a(i)/b(i)$ or $b(i)/a(i)$. This aligns with the concept of the *Relative Point Margin* introduced by Ackerman *et al.*Ben-David & Ackerman (2008). They define the Relative Point Margin of a sample $x$ as:

$$RPM(x) = \frac{d(x, c_x)}{d(x, c_{x_0})}, \tag{6}$$

where $c_x$ is the closest center to $x$ and $c_{x_0}$ is the second closest center. They also introduce a class-belonging quality measure called relative margin, calculated from the RPM, satisfying Kleinberg's axioms, i.e. Function Scale Invariance, Function Consistency, and Function Richness Kleinberg (2002); Ben-David & Ackerman (2008). This reformulated silhouette-based metric has been employed in FIQA, particularly in Boutros et al. (2023). Their approach utilizes the same intra-class and inter-class distance terms but adapts them to the face quality domain. In their framework, the Closest Class Similarity (CCS) and Nearest Non-Class Center Similarity (NNCCS) are defined analogously to redefined $a(i)$ and $b(i)$ terms:

$$\text{CCS}(x) = \cos(x, W_{y_x}), \text{NNCCS}(x) = \max_{y \neq y_x} \cos(x, W_y), \tag{7}$$

where $W_{y_x}$ is the class center corresponding to the ground truth identity of $x$, and $W_y$ represents the nearest competing class center. The key insight in their method is that the CCS/NNCCS reflects the separability of a sample in the embedding space, making it a useful proxy for FIQ in terms of utility. They attempt to learn $f(\cdot)$ with the labels being:

$$q(\cdot) = \frac{\text{CCS}(\cdot)}{\text{NNCCS}(\cdot)}, \tag{8}$$

where $f(x)$ represents the predicted FIQ, and $q(x)$ represents the pseudo-label for the quality for the sample $x$. By optimizing this formulation, their method estimates FIQ in a way that aligns with the induced class clustering structure of the learned feature space.

**Limitations of Single-Epoch Quality Estimation:** During training, each epoch generates a different induced class clustering of the data, making quality estimation a moving target—i.e., $q_j(\cdot)$ varies for each epoch $j$. This causes not only fluctuating target values but also unstable ranking of image qualities, creating training instability as learning objectives constantly shift Grill et al. (2020); Sutskever et al. (2013). Additionally, FIQA methods requiring FR training, including CR-FIQA Boutros et al. (2023), typically select optimal checkpoints based on small benchmark performance and report these on larger datasets, introducing selection bias, which might be suboptimal in real operation. Our cumulative approach naturally addresses both issues by aggregating information across epochs, reducing dependency on specific checkpoints.

**(Proposed Approach) Cumulative Average of Relative Point Margin (CARPM):** A cumulative average ($CA$), is a running average of a sequence of values that is updated incrementally as new data points are added Ya-Lun (1963). At each step, the cumulative average is computed as the average of all previous values up to that point. Mathematically, for a sequence of values $x_1, x_2, \ldots, x_n$, the cumulative average at step $n$ is given by:

$$CA_n = \frac{1}{n} \sum_{i=1}^{n} x_i = \frac{x_n + (n-1)CA_{n-1}}{n}. \tag{9}$$

In the context of training a FR model, we may view the model's training process as an induced dynamic class clustering. Since each iteration generates a different induced class clustering due to the constantly evolving nature of the learned feature space, the target values, such as the relative point margin Ben-David & Ackerman (2008), are subject to change. This results in fluctuating rankings of image qualities, making it difficult to establish stable target values for optimization. To mitigate this issue and stabilize the training, we propose using the cumulative average of the relative point margin (CARPM) as the target during training. By averaging the target values over multiple epochs, we smooth out the fluctuations caused by the dynamic nature of the induced class clustering. Specifically, for each sample $x$ at epoch $n$, we calculate its cumulative average quality score $Q(x)_n$ as follows:

$$Q(x)_n = \frac{1}{n} \sum_{j=1}^{n} q_j(x), \tag{10}$$

where $q_j(x)$ is the quality score for sample $x$ at epoch $j$. Given the dynamic nature of induced feature clustering during training, we can model the quality score at epoch $n$ as:

$$q_n(x) = Q^*(x) + \varepsilon_n, \tag{11}$$

where $Q^*(x)$ represents the "true" quality value and $\varepsilon_n$ is the noise from induced clustering variations at epoch $n$ and sample $x$. We can conceptually characterize this noise as if we were to train our model $m$ times with different initializations, which would yield $m$ different quality scores $q_n^1(x), q_n^2(x), ..., q_n^m(x)$ for the sample $x$ at epoch $n$. Such variations would allow to empirically estimate $\varepsilon_n$ distribution.

**Theoretical Analysis of CARPM:** We now provide a theoretical analysis of the cumulative average of relative point margin (CARPM), demonstrating its statistical advantages over single-epoch quality estimates. We begin with a formal characterization of the quality estimation problem.

**Assumption 1** (Quality Score Noise Model). *Assuming both $q_n(x)$ and $Q^*(x)$ are appropriately normalized, we model the quality score at epoch $n$ as $q_n(x) = Q^*(x) + \varepsilon_n$, where $\varepsilon_n \sim \mathcal{N}(0, \sigma_n^2)$ is a normally distributed error term with zero mean.*

This noise model captures the stochastic nature of the training process and the resulting variability in quality estimates. The zero-mean property implies that the quality score $q_n(x)$ is an unbiased estimate of the true quality $Q^*(x)$, i.e., $\mathbb{E}[q_n(x)] = \mathbb{E}[Q^*(x)]$. This is reasonable because during training, the network optimizes to minimize classification loss, tending to produce embeddings that correctly represent the identities. While estimations fluctuate, these fluctuations are equally likely to over or underestimate the true quality rather than showing systematic bias in either direction.

**Assumption 2** (Independent Errors). *The noise terms are independent across different epochs. This means that $\varepsilon_i$ and $\varepsilon_j$ are uncorrelated for $i \neq j$, and thus $\mathbb{E}[\varepsilon_i \varepsilon_j] = \mathbb{E}[\varepsilon_i] \mathbb{E}[\varepsilon_j] = 0$.*

While the independence of errors $\varepsilon_i$ and $\varepsilon_j$ across epochs is debatable due to sequential parameter updates, this assumption is supported by: (1) stochasticity from mini-batch selection and data augmentation; (2) empirically observed decreasing correlations between temporally distant epochs; and (3) the fact that averaging benefits persist even with some correlation, though potentially with less dramatic variance reduction. Our analysis therefore provides useful, if conservative, bounds.

Based on these assumptions, we can establish several important properties of our CARPM approach.

**Property 1** (Unbiasedness of CARPM). *Under Assumption 1, the cumulative average quality score $Q(x)_n$ is an unbiased estimator of the true quality $Q^*(x)$.*

*Proof.* The cumulative average quality score after $n$ epochs is:

$$Q(x)_n = \frac{1}{n}\sum_{j=1}^{n} q_j(x) = \frac{1}{n}\sum_{j=1}^{n}(Q^*(x) + \varepsilon_j) = Q^*(x) + \frac{1}{n}\sum_{j=1}^{n}\varepsilon_j. \qquad (12)$$

Assuming the noise terms $\varepsilon_j$ have zero mean and finite variance $\sigma_j^2$, the expected value of this estimate becomes:

$$\mathbb{E}[Q(x)_n] = \mathbb{E}\left[Q^*(x) + \frac{1}{n}\sum_{j=1}^{n}\varepsilon_j\right] = \mathbb{E}[Q^*(x)] + \frac{1}{n}\sum_{j=1}^{n}\mathbb{E}[\varepsilon_j]$$

$$= Q^*(x) + 0 = Q^*(x).$$

Hence, our proposed metric $Q(x)_n$ is unbiased if $q_n(x)$ is an unbiased estimate of the true quality $Q^*(x)$. $\qquad \square$

**Property 2** (Reduced Variance Property). *Under Assumptions 1, and 2, the variance of $Q(x)_n$ decreases at a rate proportional to $1/n$ as the number of epochs increases.*

*Proof.* As $Var(aX + b) = a^2 Var(X)$, and $Var(X + Y) = Var(X) + Var(Y) + 2Cov(X, Y)$, the variance of the cumulative average is:

$$Var(Q(x)_n) = Var(Q^*(x)) + Var\left(\frac{1}{n}\sum_{j=1}^{n}\varepsilon_j\right)$$

$$= 0 + \frac{1}{n^2}\sum_{j=1}^{n}\sigma_j^2 + \frac{2}{n^2}\sum_{i<j\leq n}Cov(\varepsilon_i, \varepsilon_j) = \frac{1}{n^2}\sum_{j=1}^{n}\sigma_j^2.$$

Since the noise terms are independent for $\varepsilon_i, \varepsilon_j$, $Cov(\varepsilon_i, \varepsilon_j) = 0$ for all $i, j$, thus, the third term becomes zero. We note that $Var(q_n(x)) = \sigma_n^2$, and show:

$$\mathbb{E}[Var(Q(x)_n)] = \mathbb{E}\left[\frac{1}{n^2}\sum_{j=1}^{n}\sigma_j^2\right] = \frac{1}{n^2}\sum_{j=1}^{n}\mathbb{E}[\sigma_j^2]$$

$$= \frac{1}{n}\mathbb{E}[\sigma_j^2] := \frac{\bar{\sigma}^2}{n} \leq \bar{\sigma}^2 = \mathbb{E}[Var(q_n(x))].$$

This shows that as $n$ increases, the expected value of variance of our quality estimate decreases proportionally to $1/n$, providing a more stable target for optimization. $\qquad \square$

**Property 3** (Lower Mean Squared Error). *Under Assumptions 1, and 2, the mean squared error of $Q(x)_n$ in approximating $Q^*(x)$ is lower than that of any single-epoch estimate $q_n(x)$ by a factor of $1/n$.*

*Proof.* For any estimator using a single epoch's measurement $q_n(x)$, the expected squared error is:

$$\mathbb{E}[(q_n(x) - Q^*(x))^2] = \mathbb{E}[\varepsilon_n^2] := \bar{\varepsilon}^2. \qquad (13)$$

In contrast, the expected squared error for the cumulative average $\mathbb{E}[(Q(x)_n - Q^*(x))^2]$ is:

$$\mathbb{E}\left[\left(\frac{1}{n}\sum_{j=1}^{n}\varepsilon_j\right)^2\right] = \mathbb{E}\left[\frac{1}{n^2}\left(\sum_{j=1}^{n}\varepsilon_j^2 + 2\sum_{i<j\leq n}\varepsilon_i\varepsilon_j\right)\right]$$

$$= \frac{1}{n^2}\left(\sum_{j=1}^{n}\mathbb{E}[\varepsilon_j^2] + 2\sum_{i<j\leq n}\mathbb{E}[\varepsilon_i\varepsilon_j]\right) = \frac{1}{n^2}\sum_{j=1}^{n}\bar{\varepsilon}^2$$

$$= \frac{\bar{\varepsilon}^2}{n} < \bar{\varepsilon}^2 = \mathbb{E}[(q_n(x) - Q^*(x))^2].$$

The cumulative average approach reduces estimation error by a factor of $n$ compared to using single-epoch measurements. $\qquad \square$

**Proposition 1** (Convergence of Ranking Probability). *Under Assumptions 1, and 2, for any two samples $x_a$ and $x_b$ with true qualities $Q^*(x_a) > Q^*(x_b)$, the probability of incorrect ranking using $Q(x)_n$ decreases at a rate of at least $1/n$ and approaches zero as $n \to \infty$.*

*Proof.* The proof is provided in Appendix B. $\qquad \square$

By incorporating the cumulative average, we achieve **reduced variance in quality estimates** (Property 2), **lower mean squared error in approximating true quality** (Property 3), and **more consistent and reliable ranking of samples** (Proposition 1). These properties ensure that target values remain more stable throughout training, reducing the impact of transient changes in the induced class clustering structure and enabling the model to learn a more consistent representation of FIQ Grill et al. (2020); Sutskever et al. (2013).

**Quality Regression Model for Inference:** While the CARPM provides a robust quality measure, it is only available for samples in the training dataset where class centers are known. In practical applications, we need to assess the quality of previously unseen face images. To address this limitation, we develop a regression model that can predict the CARPM for any given face image. Our approach simultaneously trains a FR model and a quality regression branch. The FR model learns identity-discriminative features through ArcFace loss, while the quality regression branch learns to predict the cumulative average quality scores from the embeddings. During training, we track and accumulate the relative point margin values for each sample across epochs, providing increasingly stable quality targets as training progresses. Specifically, we extend an FR backbone network with a regression head that takes the face embeddings as input and outputs predicted quality scores. For each training sample $x_i$ at epoch $n$, we compute: (a) The current relative point margin $q_n(x_i)$, (b) the cumulative average quality score $Q(x_i)_n$ using all previous epochs, (c) the predicted quality score from the regression head $\hat{Q}(x_i)_n$. The model is trained with a combined loss function:

$$\mathcal{L} = \mathcal{L}_{Arc} + \lambda \mathcal{L}_Q, \tag{14}$$

where $\mathcal{L}_{Arc}$ is the ArcFace loss as in Equation 2, and $\mathcal{L}_Q$ is the quality prediction loss measuring the difference between the predicted scores and the cumulative average targets. We use Smooth L1 loss for $\mathcal{L}_Q$ due to its robustness to outliers and stable convergence properties by following Boutros et al. (2023):

$$\mathcal{L}_Q = \frac{1}{N} \sum_{i=1}^{N} \ell(Q(x_i)_n, \hat{Q}(x_i)_n), \tag{15}$$

where the loss function $\ell$ for each sample is defined as:

$$\ell(x, y) = \begin{cases} \frac{0.5(x-y)^2}{\beta}, & \text{if } |x - y| < \beta \\ |x - y| - 0.5\beta, & \text{otherwise} \end{cases}, \tag{16}$$

with $\beta = 0.5$ chosen as the threshold parameter that determines the transition between L1 and L2 behaviour, and $\lambda$ is a hyperparameter that balances the recognition and quality prediction objectives. We set $\lambda = 10$ for all experiments based on preliminary studies showing this provides an appropriate balance between the two learning tasks Boutros et al. (2023). This training paradigm enables our model to predict CARPM values for unseen face images. By learning from the aggregated quality measurements captured during FR training, the quality regression branch can effectively generalize to new samples.

## 3 EXPERIMENTAL SETUP

**Model training and implementation details:** We evaluate our proposed method, *CARPM-FIQA*, under two distinct protocols, small and large, based on the choice of training datasets and model architectures, following Meng et al. (2021a); Boutros et al. (2023); Kolf et al. (2024). Both protocols utilize common FR architectures, ResNet50 and ResNet100 He et al. (2016), with modifications as specified in Section 2. The small protocol employs ResNet50 trained on CASIA-WebFace Yi et al. (2014) (denoted as *CARPM-FIQA* (S)), while the large protocol utilizes ResNet100 trained on MS1MV2 Guo et al. (2016); Deng et al. (2019) (denoted as *CARPM-FIQA* (L)). The MS1MV2 dataset, a refined version of MS-Celeb-1M Guo et al. (2016) by Deng et al. (2019), consists of around 5.82 million images of 85,742 identities, whereas CASIA-WebFace contains 494,414 images of 10,575 identities Yi et al. (2014). The training follows ArcFace settings Deng et al. (2019), using a scale parameter $s$ of 64 and a margin $m$ of 0.5. Models are trained with Stochastic Gradient Descent (SGD) at an initial learning rate of 1e-1, a mini-batch size of 512, a momentum of 0.9, and a weight decay 5e-4. Data augmentation includes only random horizontal flipping with a probability of 0.5. For *CARPM-FIQA* (S), the learning rate is reduced by a factor of 10 at 20K and 28K iterations, with training stopping at 32K iterations. For *CARPM-FIQA* (L), the learning rate reduction occurs at 100K and 160K iterations, with training ending at 180K iterations. All images are aligned and cropped to $112 \times 112$ Deng et al. (2019) and normalized to pixel values between -1 and 1. All experiments are conducted using 4 Nvidia HGX A100 GPUs with 40GB VRAM each, 256 CPU cores, and 1024GB of RAM. **Evaluation benchmarks and metrics:** To assess the generalizability of *CARPM-FIQA*, we evaluate it on eight challenging benchmarks: Labeled Faces in the Wild

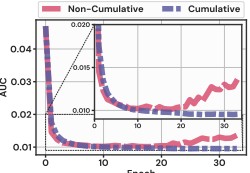 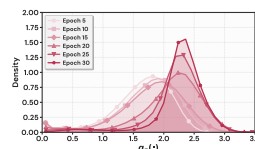 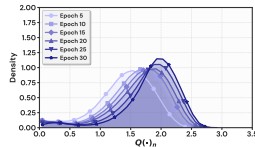

(a) EDC curves across epochs for non-cumulative (red) and cumulative (blue) targets.

(b) AUC-EDC trends. Lower values indicate better performance.

(c) Non-cumulative target distribution over epochs. Scores drift rightward and compress, reducing sample differentiation.

(d) Cumulative target distributions remain more stable and well-spread across epochs, preserving sample differentiation.

Figure 1: Comparison of cumulative and non-cumulative quality estimation. (a) EDC curves and (b) AUC-EDC trends show cumulative averaging improves estimates consistently, while non-cumulative degrades after epoch 20. (c, d) Quality target distributions highlight non-cumulative compression versus cumulative stability.

(LFW) Huang et al. (2007), AgeDB-30 Moschoglou et al. (2017), Celebrities in Frontal-Profile in the Wild (CFP-FP) Sengupta et al. (2016), Cross-Age LFW (CALFW) Zheng et al. (2017), Adience Eidinger et al. (2014), Cross-Pose LFW (CPLFW) Zheng & Deng (2018), Cross-Quality LFW (XQLFW) Knoche et al. (2021), and IJB-C Maze et al. (2018). These benchmarks facilitate comparisons with SOTA FIQA methods Boutros et al. (2023); Meng et al. (2021a); Babnik et al. (2023a; 2024) and provide insights into the robustness of *CARPM-FIQA*. Performance is measured using Error-versus-Discard Characteristic (EDC) curves Grother & Tabassi (2007), which assess the impact of discarding low-quality face images on face verification performance. The False Non-Match Rate (FNMR) is evaluated at fixed False Match Rate (FMR) thresholds ISO/IEC JTC1 SC37 Biometrics (2021), specifically at $1e-3$, recommended for border control by Frontex Frontex (2015), and $1e-4$. Additionally, the Area Under the Curve (AUC) and partial Area Under the Curve (pAUC) of EDC is reported to quantify verification performance across rejection rates. pAUC quantifies verification performance by considering only a specific portion of the EDC, up to a rejection rate of 30% by following Babnik et al. (2023b; 2024); Ou et al. (2024); Schlett et al. (2024). To examine the impact of FIQA across different FR models, we test *CARPM-FIQA* on four SOTA FR solutions: ArcFace Deng et al. (2019), ElasticFace (ElasticFace-Arc) Boutros et al. (2022), MagFace Meng et al. (2021a), and CurricularFace Huang et al. (2020) (some presented in the supplementary material). Each model processes $112 \times 112$ aligned images to generate 512-dimensional feature embeddings. Officially released ResNet-100 models pretrained on MS1MV2 by each FR solution are used. Note that all evaluations are performed under cross-model settings, i.e., the models used to learn FIQA are different than the ones used to extract feature representation of face images, demonstrating the generalizability of our approach. **Comparisons with SOTA FIQ:** We compare *CARPM-FIQA* against fourteen quality assessment methods. These include three general image quality assessment (IQA) techniques—BRISQUE Mittal et al. (2012), RankIQA Liu et al. (2017b), and DeepIQA Bosse et al. (2018), which have been shown to correlate with face utility Fu et al. (2022). Additionally, we benchmark against nine SOTA FIQA methods: RankIQ Chen et al. (2015), PFE Shi & Jain (2019), SER-FIQ Terhörst et al. (2020), FaceQnet (v1) Hernandez-Ortega et al. (2019; 2020), MagFace Meng et al. (2021a), SDD-FIQA Ou et al. (2021), CR-FIQA Boutros et al. (2023), DifFIQA Babnik et al. (2023b), eDifFIQA Babnik et al. (2024), GRAFIQS Kolf et al. (2024), and CLIB-FIQA Ou et al. (2024). All methods are evaluated using their official implementations and pretrained models as described in their respective works.

## 4 RESULTS

**Non-Cumulative vs Cumulative:** Before training our quality prediction model, we conducted a thorough analysis comparing the effectiveness of cumulative and non-cumulative quality training targets. This analysis aimed to validate our hypothesis that cumulative averaging provides a more stable and reliable training target than single-epoch estimates. We first trained a ResNet50 without the quality prediction branch, with the experimental settings mentioned in 3, and recorded the RPM scores for each sample at every epoch. These scores served as proxy quality labels of the training samples of CASIA-WebFace Yi et al. (2014) dataset, which we calculated using both the non-cumulative approach (single-epoch estimates) as in Eq. 8 and our proposed cumulative approach (across epochs) as in Eq. 9.

Table 1: Comparison of AUC-EDC and pAUC-EDC (lower is better) for non-cumulative and cumulative quality estimation across models and benchmarks (FMR: $1e-3$, $1e-4$). Best results are **bolded**. Cumulative methods outperform non-cumulative, though CALFW (age) and CPLFW (pose) remain challenging. pAUC-EDC on IJB-C—our most demanding metric and benchmark—highlights the benefit of cumulative estimation.

| Metric | FR | Quality Metric | Adience (Eidinger et al. 2014) | | AgeDB-30 (Moschoglou et al. 2017) | | CFP-FP (Sengupta et al. 2016) | | LFW (Huang et al. 2007) | | CALFW (Zheng et al. 2017) | | CPLFW (Zheng & Deng 2018) | | XQLFW (Knoche et al. 2021) | | IJB-C (Maze et al. 2018) | | |
|---|---|---|---|---|---|---|---|---|---|---|---|---|---|---|---|---|---|---|---|
| | | | $1e{-}3$ | $1e{-}4$ | $1e{-}3$ | $1e{-}4$ | $1e{-}3$ | $1e{-}4$ | $1e{-}3$ | $1e{-}4$ | $1e{-}3$ | $1e{-}4$ | $1e{-}3$ | $1e{-}4$ | $1e{-}3$ | $1e{-}4$ | $1e{-}2$ | $1e{-}3$ | $1e{-}4$ |
| AUC-EDC | ArcFace Deng et al. (2019) | non-cumulative | 0.0255 | 0.0587 | **0.0177** | **0.0224** | 0.0082 | 0.0137 | 0.0024 | 0.0029 | **0.0551** | **0.0585** | 0.0461 | 0.0640 | 0.2728 | 0.310 | 0.0119 | **0.0175** | 0.0260 |
| | | cumulative | **0.0223** | **0.0440** | 0.0187 | 0.0261 | **0.0058** | **0.0084** | **0.0017** | **0.0023** | 0.0582 | 0.0619 | **0.0432** | **0.0607** | **0.2171** | **0.2681** | **0.0118** | 0.0176 | **0.0258** |
| | ElasticFace Boutros et al. (2022) | non-cumulative | 0.0287 | 0.0523 | 0.0183 | 0.0195 | 0.0084 | 0.0115 | 0.0019 | 0.0025 | **0.0538** | **0.0551** | 0.0424 | **0.0545** | 0.2306 | 0.2974 | 0.0122 | 0.0174 | 0.0256 |
| | | cumulative | **0.0239** | **0.0412** | **0.0180** | **0.0192** | **0.0057** | **0.0078** | **0.0016** | **0.0022** | 0.0566 | 0.0581 | **0.0395** | 0.0634 | **0.2062** | **0.2477** | **0.0121** | **0.0173** | **0.0253** |
| | MagFace Meng et al. (2021a) | non-cumulative | 0.0260 | 0.0577 | **0.0201** | **0.0340** | 0.0105 | 0.0129 | 0.0025 | 0.0030 | **0.0540** | **0.0554** | 0.0466 | **0.0806** | 0.3007 | 0.3312 | **0.0134** | 0.0204 | 0.0295 |
| | | cumulative | **0.0228** | **0.0440** | 0.0213 | 0.0384 | **0.0070** | **0.0088** | **0.0017** | **0.0023** | 0.0576 | 0.0586 | **0.0426** | 0.0973 | **0.2505** | **0.2993** | **0.0134** | **0.0202** | **0.0288** |
| | CurricularFace Huang et al. (2020) | non-cumulative | 0.0240 | 0.0513 | 0.0198 | 0.0238 | 0.0102 | 0.0147 | 0.0025 | 0.0030 | **0.0545** | **0.0564** | 0.0397 | 0.0854 | 0.2309 | 0.2703 | **0.0120** | **0.0168** | **0.0245** |
| | | cumulative | **0.0213** | **0.0388** | **0.0195** | **0.0224** | **0.0071** | **0.0101** | **0.0017** | **0.0023** | 0.0574 | 0.0603 | **0.0373** | **0.0670** | **0.1886** | **0.2200** | 0.0121 | 0.0171 | **0.0244** |
| pAUC-EDC | ArcFace Deng et al. (2019) | non-cumulative | 0.0105 | 0.0262 | 0.0076 | 0.0113 | 0.0040 | 0.0074 | 0.0008 | 0.0009 | **0.0178** | **0.0196** | **0.0218** | **0.0340** | 0.1394 | 0.1597 | **0.0041** | 0.0063 | 0.0094 |
| | | cumulative | **0.0093** | **0.0238** | **0.0073** | **0.0112** | **0.0033** | **0.0053** | **0.0006** | **0.0008** | 0.0185 | 0.0205 | 0.0224 | 0.0347 | **0.1219** | **0.1473** | **0.0041** | **0.0061** | **0.0093** |
| | ElasticFace Boutros et al. (2022) | non-cumulative | 0.0118 | 0.0231 | 0.0073 | 0.0078 | 0.0039 | 0.0059 | 0.0007 | 0.0009 | **0.0172** | **0.0178** | 0.0205 | **0.0304** | 0.1277 | 0.1547 | 0.0042 | 0.0060 | 0.0092 |
| | | cumulative | **0.0105** | **0.0212** | **0.0070** | **0.0075** | **0.0031** | **0.0046** | **0.0005** | **0.0008** | 0.0177 | 0.0184 | **0.0199** | 0.0420 | **0.1213** | **0.1462** | **0.0041** | **0.0059** | **0.0090** |
| | MagFace Meng et al. (2021a) | non-cumulative | 0.0107 | 0.0256 | 0.0079 | 0.0167 | 0.0058 | 0.0076 | 0.0008 | 0.0010 | **0.0174** | **0.0183** | 0.0232 | **0.0479** | 0.1476 | **0.1552** | 0.0046 | 0.0072 | 0.0107 |
| | | cumulative | **0.0097** | **0.0237** | **0.0077** | **0.0164** | **0.0046** | **0.0059** | **0.0006** | **0.0007** | 0.0182 | 0.0184 | **0.0230** | 0.0693 | **0.1337** | 0.1630 | **0.0045** | **0.0070** | **0.0104** |
| | CurricularFace Huang et al. (2020) | non-cumulative | 0.0091 | 0.0217 | 0.0081 | 0.0097 | 0.0048 | 0.0072 | 0.0008 | 0.0009 | **0.0173** | **0.0184** | 0.0187 | **0.0315** | 0.1167 | 0.1319 | 0.0041 | 0.0060 | 0.0087 |
| | | cumulative | **0.0083** | **0.0196** | **0.0076** | **0.0094** | **0.0040** | **0.0059** | **0.0006** | **0.0008** | 0.0179 | 0.0194 | **0.0182** | 0.0457 | **0.1093** | **0.1284** | **0.0040** | **0.0059** | **0.0086** |

Figure 1 illustrates a fundamental difference between the non-cumulative and cumulative approaches through their quality training target distributions, respectively $q_n(x)$ and $Q(x)_n$. The non-cumulative approach (Fig. 1c) exhibits two critical limitations as training progresses: (1) a pronounced rightward shift of the entire distribution, indicating that scores tend to increase over time regardless of actual sample quality; and (2) a compression effect where the distribution becomes increasingly dense, reducing the discriminative power between samples of varying quality. The rightward shift occurs because the model becomes more confident in its classifications, and the separation between samples and their competing classes generally increases, artificially inflating quality scores even for samples that may not be intrinsically high-quality. This effect is particularly problematic in later epochs when the network begins focusing on optimizing difficult samples, leading to inflated quality estimates for these challenging cases. In contrast, the cumulative approach (Fig. 1d) maintains a more consistent and stretched distribution throughout training. By averaging quality estimates across epochs, it prevents the artificial inflation of scores and preserves meaningful differentiation between samples of varying quality.

To evaluate the effectiveness of these quality training targets, if hypothetically used as a quality score for the training data, in improving verification performance, we plotted EDC curves for several epochs (5, 10, 15, 20, 25, and 30), as shown in Figure 1a. These curves illustrate how verification error rates change as we progressively remove samples with the lowest quality scores. Our analysis revealed a critical limitation of the non-cumulative approach: there exists a "sweet spot" around epoch 20, after which the quality estimates begin to deteriorate. This deterioration probably occurs because, in later training epochs, the model has already learned to correctly classify easy samples and focuses primarily on optimizing difficult samples. This shift artificially inflates the quality estimates for these difficult samples, reducing the reliability of the non-cumulative quality estimates. In contrast, the cumulative approach incorporates information from all previous epochs, maintaining a more consistent quality ranking that reflects each sample's overall utility throughout the training process. To quantify this difference, we calculated the AUC of EDC curves (AUC-EDC) for each epoch and plotted the results in Figure 1b. As shown in Figure 1b, the AUC-EDC values for the non-cumulative approach initially decrease (improve) until approximately epoch 20, but then begin to increase, indicating worsening performance. In contrast, the AUC-EDC values for the cumulative approach consistently decrease throughout training, demonstrating the superior stability and reliability of cumulative quality estimation. This empirical analysis strongly supports our theoretical findings in Section 2, confirming that cumulative averaging of quality scores produces more stable and reliable quality estimates than single-epoch estimates. We have further provided the sample images, specifically the 10 highest and 10 lowest quality samples for three different identities according to both the non-cumulative and cumulative approach, to inspect how the proxy quality labels evolve during the training for each approach in the supplementary material.

**FIQA with Non-Cumulative vs Cumulative quality target:** The AUC-EDC and pAUC-EDC results of training two instances of ResNet using the setting described in Section 3 are summarized in Table 1. The results confirm that the cumulative approach consistently outperforms the non-cumulative approach, even in the most demanding metric (pAUC-EDC) and challenging benchmark (IJB-C). However, we observed certain limitations with challenging cross-attribute datasets. For datasets with significant age variations (CALFW) and extreme pose variations (CPLFW), the performance gap between cumulative and non-cumulative approaches narrows, and the cumulative approach performs comparably or slightly worse. This phenomenon might be attributed to the fact that these cross-attribute datasets represent more complex recognition challenges where the model's

Table 2: The pAUC-EDC achieved by *CARPM-FIQA* and the SOTA methods under different experimental settings. The notions of $1e-3$ and $1e-4$ indicate the value of the fixed FMR at which the EDC curves (FNMR vs. reject) were calculated. The results are compared to three IQA and eleven FIQA approaches. The XQLFW dataset uses SER-FIQ (marked with *) as FIQ labeling method.

| FR | Method | Adience Eidinger et al. (2014) | | AgeDB-30 Moschoglou et al. (2017) | | CFP-FP Sengupta et al. (2016) | | LFW Huang et al. (2007) | | CALFW Zheng et al. (2017) | | CPLFW Zheng & Deng (2018) | | XQLFW Knoche et al. (2021) | | IJB-C Maze et al. (2018) | |
|---|---|---|---|---|---|---|---|---|---|---|---|---|---|---|---|---|---|
| | | 1e-3 | 1e-4 | 1e-3 | 1e-4 | 1e-3 | 1e-4 | 1e-3 | 1e-4 | 1e-3 | 1e-4 | 1e-3 | 1e-4 | 1e-3 | 1e-4 | 1e-3 | 1e-4 |
| ElasticFace Boutros et al. (2022) — IQA | BRISQUE Mittal et al. (2012) | 0.0160 | 0.0302 | 0.0090 | 0.0099 | 0.0082 | 0.0107 | 0.0007(3) | 0.0010 | 0.0195 | 0.0203 | 0.0427 | 1.1055 | 0.1412 | 0.1638 | 0.0069 | 0.0108 |
| | RankIQA Liu et al. (2017b) | 0.0138 | 0.0274 | 0.0085 | 0.0096 | 0.0082 | 0.0105 | 0.0007(3) | 0.0010 | 0.0203 | 0.0209 | 0.0433 | 0.1086 | 0.1428 | 0.1661 | 0.0068 | 0.0106 |
| | DeepIQA Bosse et al. (2018) | 0.0162 | 0.0308 | 0.0088 | 0.0097 | 0.0074 | 0.0100 | 0.0008 | 0.0010 | 0.0201 | 0.0208 | 0.0431 | 0.1082 | 0.1379 | 0.1621 | 0.0070 | 0.0108 |
| FIQA | RankIQC Chen et al. (2015) | 0.0139 | 0.0276 | 0.0089 | 0.0097 | 0.0067 | 0.0085 | 0.0005(1) | 0.0008(2) | 0.0182 | 0.0188 | 0.0291 | 0.0394 | 0.1163 | 0.1342 | 0.0065 | 0.0100 |
| | PFEShi & Jain (2019) | 0.0106 | 0.0211 | 0.0064 | 0.0069 | 0.0049 | 0.0065 | 0.0006(2) | 0.0008(2) | 0.0181 | 0.0186 | 0.0219 | 0.0682 | 0.1180 | 0.1401 | 0.0060 | 0.0091 |
| | SER-FIQTerhörst et al. (2020) | 0.0114 | 0.0227 | 0.0064 | 0.0072 | 0.0031(2) | 0.0044(2) | 0.0006(2) | 0.0008(2) | 0.0177 | 0.0184 | 0.0185 | 0.0292 | 0.1057(1)* | 0.1283(3)* | 0.0054(1) | 0.0083(1) |
| | FaceQnetHernandez-Ortega et al. (2019; 2020) | 0.0143 | 0.0274 | 0.0075 | 0.0082 | 0.0071 | 0.0084 | 0.0007(3) | 0.0009(3) | 0.0189 | 0.0196 | 0.0371 | 0.0951 | 0.1428 | 0.1645 | 0.0068 | 0.0103 |
| | MagFaceMeng et al. (2021a) | 0.0110 | 0.0211 | 0.0060(2) | 0.0064(2) | 0.0043 | 0.0059 | 0.0005(1) | 0.0007(1) | 0.0173(3) | 0.0177(3) | 0.0237 | 0.0345 | 0.1331 | 0.1445 | 0.0058 | 0.0089 |
| | SDD-FIQAOu et al. (2021) | 0.0115 | 0.0231 | 0.0074 | 0.0080 | 0.0054 | 0.0067 | 0.0006(2) | 0.0008(2) | 0.0181 | 0.0186 | 0.0255 | 0.0377 | 0.1336 | 0.1564 | 0.0059 | 0.0090 |
| | CR-FIQA(S) Boutros et al. (2023) | 0.0112 | 0.0223 | 0.0067 | 0.0073 | 0.0038 | 0.0057 | 0.0005(1) | 0.0008(2) | 0.0172(2) | 0.0176(2) | 0.0197 | 0.0301 | 0.1166 | 0.1411 | 0.0058 | 0.0087 |
| | CR-FIQA(L) Boutros et al. (2023) | 0.0105 | 0.0206 | 0.0064 | 0.0069 | 0.0031(2) | 0.0045(3) | 0.0006(2) | 0.0009(3) | 0.0171(1) | 0.0175(1) | 0.0178(3) | 0.0275(2) | 0.1094(2) | 0.1265(1) | 0.0055(2) | 0.0084(2) |
| | DifFIQA(R) Babnik et al. (2023b) | 0.0113 | 0.0227 | 0.0076 | 0.0082 | 0.0031(2) | 0.0045(3) | 0.0006(2) | 0.0007(1) | 0.0180 | 0.0186 | 0.0174(1) | 0.0271(1) | 0.1094(2) | 0.1279(2) | 0.0060 | 0.0091 |
| | eDifFIQA(L) Babnik et al. (2024) | 0.0099(2) | 0.0205(3) | 0.0058(1) | 0.0062(1) | 0.0029(1) | 0.0044(2) | 0.0006(2) | 0.0007(1) | 0.0171(1) | 0.0176(2) | 0.0179 | 0.0280 | 0.1129(3) | 0.1322 | 0.0056(3) | 0.0086(3) |
| | GraFIQs (S) Kolf et al. (2024) | 0.0132 | 0.0257 | 0.0088 | 0.0098 | 0.0067 | 0.0083 | 0.0007(3) | 0.0010 | 0.0185 | 0.0190 | 0.0271 | 0.0523 | 0.1255 | 0.1381 | 0.0057 | 0.0088 |
| | GraFIQs (L) Kolf et al. (2024) | 0.0101(3) | 0.0203(2) | 0.0066 | 0.0073 | 0.0034 | 0.0047 | 0.0006(2) | 0.0009(3) | 0.0176 | 0.0181 | 0.0194 | 0.0415 | 0.1270 | 0.1528 | 0.0056(3) | 0.0086(3) |
| | CLIB-FIQA Ou et al. (2024) | 0.0103 | 0.0214 | 0.0062 | 0.0066 | 0.0032(3) | 0.0048 | 0.0005(1) | 0.0007(1) | 0.0171(1) | 0.0176(2) | 0.0179 | 0.0276(3) | 0.1140 | 0.1532 | 0.0056(3) | 0.0084(2) |
| | *CARPM-FIQA* (S) | 0.0105 | 0.0212 | 0.0070 | 0.0075 | 0.0031(2) | 0.0046 | 0.0005(1) | 0.0008(2) | 0.0177 | 0.0184 | 0.0199 | 0.0420 | 0.1213 | 0.1462 | 0.0059 | 0.0090 |
| | *CARPM-FIQA* (L) | **0.0096(1)** | **0.0195(1)** | 0.0061(3) | 0.0065(3) | **0.0029(1)** | **0.0042(1)** | 0.0006(2) | 0.0008(2) | 0.0179 | 0.0184 | 0.0182 | 0.0278 | 0.1159 | 0.1310 | 0.0055(2) | 0.0086(3) |
| MagFace et al. (2021a) — IQA | BRISQUE Mittal et al. (2012) | 0.0148 | 0.0334 | 0.0101 | 0.0207 | 0.0117 | 0.0205 | 0.0009 | 0.0013 | 0.0199 | 0.0211 | 0.0700 | 0.1672 | 0.1601 | 0.1727 | 0.0084 | 0.0131 |
| | RankIQA Liu et al. (2017b) | 0.0128 | 0.0291 | 0.0092 | 0.0212 | 0.0119 | 0.0207 | 0.0009 | 0.0013 | 0.0208 | 0.0217 | 0.0518 | 0.1695 | 0.1619 | 0.1744 | 0.0083 | 0.0130 |
| | DeepIQA Bosse et al. (2018) | 0.0149 | 0.0335 | 0.0100 | 0.0204 | 0.0111 | 0.0199 | 0.0009 | 0.0013 | 0.0206 | 0.0215 | 0.0710 | 0.1691 | 0.1576 | 0.1706 | 0.0085 | 0.0131 |
| FIQA | RankIQC Chen et al. (2015) | 0.0125 | 0.0302 | 0.0100 | 0.0199 | 0.0096 | 0.0178 | 0.0007(2) | 0.0010 | 0.0188 | 0.0198 | 0.0336 | 0.1133 | 0.1392 | 0.1514 | 0.0077 | 0.0117 |
| | PFEShi & Jain (2019) | 0.0098 | 0.0239 | 0.0074 | 0.0161 | 0.0066 | 0.0092 | 0.0007(2) | 0.0008(2) | 0.0181 | 0.0189 | 0.0253 | 0.1178 | 0.1386 | 0.1558 | 0.0072 | 0.0106 |
| | SER-FIQTerhörst et al. (2020) | 0.0107 | 0.0241 | 0.0074 | 0.0160 | 0.0045(3) | 0.0099 | 0.0007(2) | 0.0011 | 0.0183 | 0.0187 | 0.0219 | 0.0541(3) | 0.1264(1)* | 0.1440(1)* | 0.0066(1) | 0.0097(1) |
| | FaceQnetHernandez-Ortega et al. (2019; 2020) | 0.0133 | 0.0292 | 0.0082 | 0.0159 | 0.0096 | 0.0162 | 0.0008(3) | 0.0010 | 0.0193 | 0.0198 | 0.0602 | 0.1589 | 0.1584 | 0.1681 | 0.0080 | 0.0120 |
| | MagFaceMeng et al. (2021a) | 0.0100 | 0.0233 | 0.0066(1) | 0.0134 | 0.0057 | 0.0096 | 0.0006(1) | 0.0008(2) | 0.0178(2) | 0.0184(3) | 0.0268 | 0.0579 | 0.1496 | 0.1603 | 0.0070 | 0.0104 |
| | SDD-FIQAOu et al. (2021) | 0.0106 | 0.0257 | 0.0081 | 0.0122(3) | 0.0083 | 0.0128 | 0.0007(2) | 0.0010 | 0.0186 | 0.0194 | 0.0284 | 0.0834 | 0.1525 | 0.1656 | 0.0071 | 0.0106 |
| | CR-FIQA(S) Boutros et al. (2023) | 0.0103 | 0.0246 | 0.0074 | 0.0137 | 0.0054 | 0.0068(3) | 0.0007(2) | 0.0010 | 0.0177(1) | 0.0182(1) | 0.0225 | 0.0548 | 0.1339 | 0.1619 | 0.0070 | 0.0102 |
| | CR-FIQA(L) Boutros et al. (2023) | 0.0100 | 0.0210(1) | 0.0071 | 0.0128 | 0.0048 | 0.0061(2) | 0.0007(2) | 0.0008(2) | 0.0177(1) | 0.0183(2) | 0.0209(2) | 0.0454(2) | 0.1296 | 0.1506 | 0.0067(2) | 0.0098(2) |
| | DifFIQA(R) Babnik et al. (2023b) | 0.0103 | 0.0251 | 0.0086 | 0.0171 | 0.0047 | 0.0104 | 0.0006(1) | 0.0008(2) | 0.0185 | 0.0194 | 0.0207(1) | 0.0598 | 0.1280(2) | 0.1501(3) | 0.0072 | 0.0107 |
| | eDifFIQA(L) Babnik et al. (2024) | 0.0093(2) | 0.0224(3) | 0.0076 | 0.0113(1) | 0.0042(1) | 0.0097 | 0.0006(1) | 0.0008(2) | 0.0177(1) | 0.0182(1) | 0.0213(3) | 0.0596 | 0.1323 | 0.1498(2) | 0.0068(3) | 0.0102 |
| | GraFIQs (S) Kolf et al. (2024) | 0.0125 | 0.0273 | 0.0097 | 0.0204 | 0.0100 | 0.0181 | 0.0008(3) | 0.0013 | 0.0191 | 0.0201 | 0.0344 | 0.1017 | 0.1386 | 0.1557 | 0.0070 | 0.0104 |
| | GraFIQs (L) Kolf et al. (2024) | 0.0097(3) | 0.0217(2) | 0.0070 | 0.0136 | 0.0049 | 0.0111 | 0.0008(3) | 0.0011 | 0.0179(3) | 0.0185 | 0.0230 | 0.0693 | 0.1414 | 0.1576 | 0.0068(3) | 0.0101(3) |
| | CLIB-FIQA Ou et al. (2024) | 0.0098 | 0.0234 | 0.0071 | 0.0119(2) | 0.0048 | 0.0105 | 0.0007(2) | 0.0009(2) | 0.0177(1) | 0.0183 | 0.0213(2) | 0.0599 | 0.1284(3) | 0.1554 | 0.0067(2) | 0.0098(2) |
| | *CARPM-FIQA* (S) | 0.0097(3) | 0.0237 | 0.0077 | 0.0164 | 0.0046 | **0.0059(1)** | 0.0006(1) | 0.0007(1) | 0.0182 | 0.0186 | 0.0230 | 0.0693 | 0.1337 | 0.1630 | 0.0070 | 0.0104 |
| | *CARPM-FIQA* (L) | **0.0092(1)** | **0.0210(1)** | 0.0070(3) | 0.0138 | 0.0044(2) | 0.0080 | 0.0007(2) | 0.0008(2) | 0.0185 | 0.0192 | 0.0212(3) | 0.0451(1) | 0.1368 | 0.1507 | 0.0067(2) | 0.0101(3) |

focus on difficult samples during later training epochs can actually be beneficial. In such cases, the non-cumulative approach's tendency to inflate quality scores for difficult samples may temporarily align with the actual utility of images for these specific recognition tasks. Nevertheless, even in these challenging scenarios, the cumulative approach generally provides more stable and consistent quality estimates across the training process, as evidenced by the smoother improvement trends in AUC-EDC values shown in Figure 1b. The validation trends of our method across different training stages (Epochs 5, 10, 15, 20, 25, and 30), similar to Figure 1b, using the same benchmarks of Table 1 are detailed in the supplementary material.

**Comparison to SOTA:** Having validated the effectiveness of our cumulative approach, we now compare our *CARPM-FIQA* method against three IQA and eleven SOTA FIQA methods. Table 2 presents the quantitative results across different quality metrics and FR models (ElasticFace Boutros et al. (2022) and MagFace Meng et al. (2021a)). Our method consistently achieves competitive or superior performance across multiple evaluation criteria. Specifically, for both ElasticFace and MagFace, *CARPM-FIQA* demonstrates lower pAUC-EDC values in most evaluated metrics, highlighting its robustness in quality estimation. Notably, *CARPM-FIQA* (L) attains the best or near-best results. For ElasticFace, *CARPM-FIQA* (S) ranks first once and second once, while *CARPM-FIQA* (L) achieves first place in Adience and CFP-FP, second in LFW/IJB-C, and third in AgeDB-30. For MagFace, *CARPM-FIQA* (S) ranks first in CFP-FP/LFW and third in Adience, while *CARPM-FIQA* (L) achieves first place in Adience/CPLFW, second in LFW/CFP-FP/IJB-C, and third in AgeDB-30. Due to page limitations, full results for all models (ArcFace and CurricularFace included) are in supplementary material; we focus on widely-adopted, top-performing ElasticFace and MagFace here. The supplementary also contains AUC-EDC scores, EDC for FNMR@FMR=$1e-3$/$1e-4$ across all models/datasets, and score distributions. These additional experiments lead to similar conclusions.

## 5 CONCLUSION

This paper presented *CARPM-FIQA*, a novel FIQA method accumulating Relative Point Margin scores across epochs for stable, reliable FIQ estimation. By leveraging temporal accumulation, our approach addresses instability issues common in existing FIQA methods that rely on assessing face images from a single point of the training, often from the final epoch. As we have theoretically and empirically proved, the cumulative nature of *CARPM-FIQA* offers several main advantages over non-cumulative approaches, including lower variance in quality estimates (i.e. lower movement in the training target), reduced error in approximating true quality, and more reliable sample ranking. Furthermore, *CARPM-FIQA* enhances the consistency of quality rankings throughout training and achieves superior performance over the non-cumulative approach across multiple evaluation benchmarks. Through extensive experiments across multiple FIQA benchmarks, four different FR models, and in comparison to the recent SOTA FIQA, our *CARPM-FIQA* consistently ranked among the top SOTA methods, achieving robust performance across diverse and challenging datasets as well as across different FR models.

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

## A    DETAILED EXPERIMENTAL RESULTS

This supplementary material provides comprehensive experimental results and analyses that extend the main paper. We include comparisons of cumulative versus non-cumulative quality estimation using AUC-EDC and pAUC-EDC metrics (Sec. A.1, A.2), evaluations across multiple face recognition models and loss functions (Sec. A.2), and comparisons with state-of-the-art FIQA methods (Sec. A.3). Additionally, we present EDC curves (Sec. A.4), quality score distributions (Sec. A.5), qualitative examples of ranking stability (Sec. A.6), IJB-C verification with quality-weighted embeddings (Sec. A.7), and an analysis of *CARPM-FIQA* as a feature extractor (Sec. A.8). We also investigate post-hoc versus joint training of the quality regression head (Sec. A.9) and compare SOTA FIQA training conditions (Sec. A.10). Finally, we provide a proof of convergence for cumulative ranking probabilities (Sec. B) and discuss potential societal impacts of our FIQA method (Sec. C).

### A.1    COMPARISON OF AUC-EDC AND pAUC-EDC FOR NON-CUMULATIVE AND CUMULATIVE

The complete table, including the results for ArcFace and CurricularFace, for the comparison of pAUC-EDC values for non-cumulative and cumulative quality estimation is provided in Table 6. Table 3 presents the results with the AUC-EDC metric.

### A.2    AUC-EDC AND pAUC-EDC ACROSS FACE RECOGNITION MODELS

To further validate the generalizability of our method across different FR loss functions, we conducted additional experiments using both ArcFace Deng et al. (2019) and CurricularFace Huang et al. (2020) loss functions during training our *CARPM-FIQA*. We trained two instances of *CARPM-FIQA* with ResNet50 on CASIA-WebFace Yi et al. (2014) using two different losses, namely ArcFace Deng et al. (2019) and CurricularFace Huang et al. (2020), and then evaluated their performance across all seven benchmarks using four different FR models (ArcFace Deng et al. (2019), ElasticFace Boutros et al. (2022), MagFace Meng et al. (2021a), and CurricularFace Huang et al. (2020)). These cross-model evaluations are particularly challenging as they test whether quality estimates learned under one loss function can generalize to embeddings produced by models trained with different loss functions. The comprehensive results, including both AUC-EDC and pAUC-EDC metrics, are provided in Tables 4 and 7. The experiments confirm that our cumulative approach maintains consistent performance advantages regardless of the training loss function used, with only minor variations across different FR models. This cross-loss function generalization demonstrates that our method captures fundamental quality characteristics that transcend the specific optimization objective used during training, making it robust for real-world deployment where target FR systems may differ from those used during quality model development.

### A.3    COMPARISON WITH STATE-OF-THE-ART METHODS

Table 5 extends our comparison to include AUC-EDC metric for all state-of-the-art methods evaluated in our study. These detailed results complement the pAUC-EDC metric presented in the main paper and provide further evidence of our method's effectiveness across different operating thresholds (FMR=$1e-3$ and FMR=$1e-4$). We further provided the full comparison table, in Table 8, for the pAUC-EDC metric, including the results for ArcFace and CurricularFace in addition to ElasticFace and MagFace presented in the main paper.

### A.4    VISUALIZATION OF EDC CURVES

Figures 2 and 3 visualize the Error-versus-Discard Characteristic (EDC) curves for all evaluated methods across multiple datasets and FR models. These curves demonstrate how verification errors decrease as low-quality samples are progressively removed.

### A.5 QUALITY SCORE DISTRIBUTIONS

Figure 4 illustrates the distribution of quality scores assigned by different methods across the evaluation benchmarks.

### A.6 VISUAL ANALYSIS OF QUALITY ESTIMATES

Figures 5-10 provide a qualitative analysis of how our non-cumulative and cumulative approaches rank face images from the same identity across training epochs. For three representative identities, we show both the highest and lowest quality samples as determined by each approach.

These visualizations reveal that the non-cumulative approach exhibits significant instability in quality rankings across epochs, with high-quality samples in early epochs potentially becoming ranked as lower quality in later epochs. In contrast, the cumulative approach provides more stable and consistent rankings that better align with intuitive notions of face quality. The consistent color coding across epochs allows tracking individual samples and observing how their relative quality assessments evolve during training.

Particularly noteworthy are cases where the non-cumulative approach inconsistently ranks challenging samples (e.g., profile views or partially occluded faces) across different epochs, while the cumulative approach provides more reliable assessments that account for the sample's performance throughout the entire training process.

### A.7 IJB-C VERIFICATION USING QUALITY-WEIGHTED EMBEDDINGS

Table 9 shows the verification performance on the IJB-C 1:1 mixed verification benchmark Maze et al. (2018) when using quality scores to weight image embeddings, as defined in Maze et al. (2018). For each face recognition model, the first row reports the baseline results obtained using the standard evaluation protocol and official model Deng et al. (2019); Boutros et al. (2022); Meng et al. (2021a); Huang et al. (2020), i.e., without incorporating FIQ scores. In contrast, the subsequent rows of Table 9 show results when quality scores are applied as weighting factors during template feature aggregation. Each image embedding is multiplied by its normalized quality score prior to aggregation into media- and template-level features. This ensures that higher-quality images contribute more strongly to the final template representation, improving the robustness and verification performance compared to the unweighted baseline.

### A.8 *CARPM-FIQA* AS A FACE RECOGNIZER

We report the performance of the *CARPM-FIQA* (L) backbone when used for feature extraction on standard face recognition benchmarks in Table 10, although this is not the primary focus of our work. The evaluated datasets include LFW Huang et al. (2007), AgeDB-30 Moschoglou et al. (2017), CFP-FP Sengupta et al. (2016), CALFW Zheng et al. (2017), CPLFW Zheng & Deng (2018), and IJB-C Maze et al. (2018). Evaluation metrics were adopted as defined by each benchmark: accuracy for LFW, CALFW, CPLFW, CFP-FP, and AgeDB-30, and TAR at FAR $1e - 4$ for IJB-C. While our approach is designed as a face image quality assessment (FIQA) method rather than a feature extractor, the results in Table 10 show that *CARPM-FIQA* achieves performance comparable to recent state-of-the-art models trained under similar conditions using only a face recognition loss.

### A.9 POST-HOC VS. JOINT TRAINING OF THE QUALITY REGRESSION HEAD

To investigate whether training the quality regression head after face recognition (FR) convergence is enough, we conducted a post-hoc experiment where the backbone embeddings were frozen, and only the quality head was trained. We evaluated the impact on both AUC-EDC and pAUC-EDC across multiple FR backbones and benchmarks. Table 11 reports AUC-EDC for FMR=$1e-3$ and $1e-4$, while Table 12 reports pAUC-EDC. The "Frozen FR" setting uses a pre-trained backbone with frozen embeddings, whereas "Cumulative FR" denotes joint online training of the FR backbone and quality head using CARPM. These results demonstrate that post-hoc training, which removes access to the temporal evolution of embeddings and quality targets, leads to inferior performance, confirming that embedding dynamics are crucial for learning meaningful quality scores.

### A.10 COMPARISON OF SOTA FIQA TRAINING CONDITIONS

Table 13 provides a detailed overview of the training setups and architectural choices across representative general Image Quality Assessment (IQA) and Face Image Quality Assessment (FIQA) methods. The table contrasts datasets, backbone architectures, additional modules, supervision strategies, and loss functions.

## B PROOF OF CONVERGENCE OF RANKING PROBABILITY

**Proposition 1** (Convergence of Ranking Probability). *Under Assumptions 1, and 2, for any two samples $x_a$ and $x_b$ with true qualities $Q^*(x_a) > Q^*(x_b)$, the probability of incorrect ranking using $Q(x)_n$ decreases at a rate of at least $1/n$ and approaches zero as $n \to \infty$.*

*Proof.* Consider two samples $x_a$ and $x_b$ with true qualities $Q^*(x_a) > Q^*(x_b)$, and define $\delta = Q^*(x_a) - Q^*(x_b) > 0$ as their true quality difference. For a single epoch $j$, the probability of incorrect ranking is:

$$P(q_j(x_a) < q_j(x_b)) = P(\varepsilon_j^b - \varepsilon_j^a > \delta). \tag{17}$$

If we denote $Z_j = \varepsilon_j^b - \varepsilon_j^a$, then $\mathbb{E}[Z_j] = 0$ by unbiased assumption and $Var(Z_j) := \sigma_Z^2$. Using Chebyshev's inequality A. Papoulis (1991):

$$P(Z_j > \delta) \leq P(|Z_j| > \delta) \leq \frac{\sigma_Z^2}{\delta^2}. \tag{18}$$

This establishes a lower bound on the ranking accuracy for a single epoch, which depends solely on the ratio of noise variance to the squared true quality difference. Whereas, the probability of incorrect ranking for cumulative averaging is:

$$P(Q(x_a)_n < Q(x_b)_n) = P\left(\frac{1}{n}\sum_{j=1}^{n}(\varepsilon_j^b - \varepsilon_j^a) > \delta\right). \tag{19}$$

Define $\bar{Z}_n = \frac{1}{n}\sum_{j=1}^{n} Z_j$. Since $\mathbb{E}[Z_j] = 0$ from our unbiased assumption, by the weak law of large numbers (WLLN) Bhattacharya et al. (2016):

$$\lim_{n\to\infty} P(|\bar{Z}_n| > \alpha) = 0 \text{ for any } \alpha > 0. \tag{20}$$

For incorrect ranking to occur, we need $\bar{Z}_n > \delta$. Since $\delta > 0$, by choosing $\alpha = \delta$ in the WLLN:

$$\lim_{n\to\infty} P(\bar{Z}_n > \delta) \leq \lim_{n\to\infty} P(|\bar{Z}_n| > \delta) = 0. \tag{21}$$

Therefore, the probability of incorrect ranking approaches zero as $n$ increases. While the WLLN establishes convergence, we can quantify the rate by Chebyshev's inequality:

$$P(|\bar{Z}_n - \mathbb{E}[\bar{Z}_n]| \geq \varepsilon) \leq \frac{Var(\bar{Z}_n)}{\varepsilon^2} = \frac{\sigma_Z^2}{n\varepsilon^2}, \tag{22}$$

where $\sigma_Z^2$ is the variance of $Z_j$. Setting $\varepsilon = \delta$ and noting that $\mathbb{E}[\bar{Z}_n] = 0$:

$$P(|\bar{Z}_n| \geq \delta) \leq \frac{\sigma_Z^2}{n\delta^2}. \tag{23}$$

Since incorrect ranking requires $\bar{Z}_n > \delta$, we have:

$$P(Q(x_a)_n < Q(x_b)_n) = P(\bar{Z}_n > \delta) \leq P(|\bar{Z}_n| \geq \delta) \leq \frac{\sigma_Z^2}{n\delta^2}. \tag{24}$$

This establishes that the probability of incorrect ranking decreases at least as fast as $1/n$. $\square$

## C POTENTIAL SOCIETAL IMPACTS

Our work on FIQA has several potential societal impacts that require careful consideration. We discuss both positive and negative impacts below.

**Positive societal impacts:** Our advancements in FIQA aim to improve security, convenience, and quality of life across multiple domains. Specifically:

- *Enhanced accessibility and convenience:* More reliable facial recognition systems enabled by better quality assessment can provide seamless access to financial and health services e-Aadhaar - Unique Identification Authority of India (2015), particularly benefiting individuals in remote areas or with limited documentation.

- *Improved security:* Better quality assessment in biometric systems can enhance security at borders, critical infrastructure, and secure facilities while reducing false rejections of legitimate users Frontex (2015).

- *Reduced fraud:* Reliable FIQA can help prevent identity fraud by ensuring high-quality face images are used in verification processes, potentially reducing financial crimes and protecting individuals.

- *Efficient public services:* Government services using biometric verification can operate more efficiently with improved FIQA, potentially reducing wait times and administrative burdens.

**Negative societal impacts:** We acknowledge that face recognition technologies, including FIQA, present several potential risks:

- *Privacy concerns:* More effective face recognition systems could facilitate unwanted surveillance or tracking of individuals without their knowledge or consent Meden et al. (2021).

- *Algorithmic bias:* If the training data used for FIQA methods is not sufficiently diverse, quality assessment algorithms may perform inconsistently across demographic groups, potentially leading to higher error rates for certain populations Kärkkäinen & Joo (2019).

- *Potential for misuse:* Improved face recognition could be deployed in jurisdictions without adequate privacy laws or used by unauthorized actors for surveillance, profiling, or discrimination.

- *False rejections:* Incorrect quality assessments could deny legitimate users access to essential services, which could disproportionately affect vulnerable populations.

**Mitigation strategies:** To address these concerns, we propose several mitigation strategies:

- *Legal and regulatory compliance:* We emphasize that facial recognition technology should only be deployed within well-defined legal frameworks (e.g., GDPR Voigt & Bussche (2017), BIPA 740 ILCS/14 (2008)) that protect individual privacy and require explicit consent.

- *Diverse and representative datasets:* Future work should ensure FIQA methods are trained and evaluated on diverse datasets that represent various demographic groups to minimize potential biases.

- *Transparent deployment:* Organizations deploying these technologies should clearly inform users about how their biometric data will be processed, stored, and protected.

- *Human oversight:* Critical decisions based on automated facial recognition should include human review processes, especially in high-stakes contexts like law enforcement or border control.

- *Ongoing monitoring:* Systems using FIQA should be continuously monitored for performance disparities across demographic groups and adjusted if biases are detected.

While we firmly reject any malicious or illegal applications of our research, we recognize our responsibility to acknowledge these potential impacts. Our work is intended to improve legitimate biometric applications while encouraging the responsible and ethical deployment of face recognition technology within appropriate legal and ethical frameworks.

Table 3: Comparison of AUC-EDC values for non-cumulative and cumulative quality estimation approaches across different face recognition models and benchmarks. Lower AUC-EDC values indicate better performance. Results are shown for FMR thresholds of $1e-3$ and $1e-4$. The best-performing epoch for each approach is highlighted in red for non-cumulative and blue for cumulative approaches. Final results are shown in violet to demonstrate the stability of cumulative averaging over time. While the cumulative approach generally outperforms non-cumulative methods across most datasets, we observe some limitations with cross-age (AgeDB-30, CALFW) and cross-pose (CPLFW) datasets, where performance is comparable but occasionally slightly worse at the final epoch.

| FR | Quality | Epoch | Adience Eidinger et al. (2014) | | AgeDB-30 Moschoglou et al. (2017) | | CFP-FP Sengupta et al. (2016) | | LFW Huang et al. (2007) | | CALFW Zheng et al. (2017) | | CPLFW Zheng & Deng (2018) | | XQLFW Knoche et al. (2021) | | Mean AUC | |
|---|---|---|---|---|---|---|---|---|---|---|---|---|---|---|---|---|---|---|
| | | | 1e−3 | 1e−4 | 1e−3 | 1e−4 | 1e−3 | 1e−4 | 1e−3 | 1e−4 | 1e−3 | 1e−4 | 1e−3 | 1e−4 | 1e−3 | 1e−4 | 1e−3 | 1e−4 |
| ArcFace Deng et al. (2019) | non-cumulative | 5 | 0.0379 | 0.0888 | 0.0275 | 0.0315 | 0.0087 | 0.0155 | 0.0028 | 0.0033 | 0.0658 | 0.0710 | 0.0457 | 0.0637 | 0.2159 | 0.2684 | 0.0578 | 0.0775 |
| | | 10 | 0.0222 | 0.0516 | 0.0245 | 0.0304 | 0.0088 | 0.0134 | 0.0026 | 0.0031 | 0.0622 | 0.0679 | 0.0384 | 0.0574 | 0.2197 | 0.2651 | 0.0541 | 0.0698 |
| | | 15 | 0.0274 | 0.0658 | 0.0166 | 0.0216 | 0.0067 | 0.0111 | 0.0021 | 0.0026 | 0.0619 | 0.0666 | 0.0420 | 0.0616 | 0.2181 | 0.2488 | 0.0535 | 0.0683 |
| | | 20 | 0.0229 | 0.0503 | 0.0160 | 0.0184 | 0.0061 | 0.0103 | 0.0024 | 0.0029 | 0.0570 | 0.0603 | 0.0404 | 0.0591 | 0.2103 | 0.2642 | 0.0507 | 0.0665 |
| | | 25 | 0.0231 | 0.0525 | 0.0150 | 0.0182 | 0.0071 | 0.0115 | 0.0022 | 0.0029 | 0.0531 | 0.0570 | 0.0417 | 0.0584 | 0.2847 | 0.3222 | 0.0610 | 0.0747 |
| | | 30 | 0.0255 | 0.0587 | 0.0177 | 0.0224 | 0.0082 | 0.0137 | 0.0024 | 0.0029 | 0.0551 | 0.0585 | 0.0461 | 0.0640 | 0.2728 | 0.3107 | 0.0611 | 0.0758 |
| | cumulative | 5 | 0.0600 | 0.1262 | 0.0332 | 0.0376 | 0.0470 | 0.0562 | 0.0032 | 0.0037 | 0.0723 | 0.0775 | 0.2004 | 0.2373 | 0.5317 | 0.5823 | 0.1354 | 0.1601 |
| | | 10 | 0.0234 | 0.0521 | 0.0239 | 0.0271 | 0.0086 | 0.0126 | 0.0031 | 0.0036 | 0.0644 | 0.0709 | 0.0511 | 0.0702 | 0.2943 | 0.3309 | 0.0670 | 0.0811 |
| | | 15 | 0.0240 | 0.0495 | 0.0179 | 0.0244 | 0.0097 | 0.0149 | 0.0019 | 0.0025 | 0.0595 | 0.0655 | 0.0435 | 0.0620 | 0.2252 | 0.2547 | 0.0545 | 0.0676 |
| | | 20 | 0.0207 | 0.0424 | 0.0174 | 0.0213 | 0.0066 | 0.0106 | 0.0023 | 0.0028 | 0.0573 | 0.0618 | 0.0427 | 0.0598 | 0.1921 | 0.2284 | 0.0484 | 0.0610 |
| | | 25 | 0.0214 | 0.0430 | 0.0178 | 0.0243 | 0.0059 | 0.0091 | 0.0019 | 0.0025 | 0.0602 | 0.0643 | 0.0430 | 0.0601 | 0.2234 | 0.2634 | 0.0534 | 0.0667 |
| | | 30 | 0.0223 | 0.0440 | 0.0187 | 0.0261 | 0.0058 | 0.0084 | 0.0017 | 0.0023 | 0.0582 | 0.0619 | 0.0432 | 0.0607 | 0.2171 | 0.2681 | 0.0524 | 0.0674 |
| ElasticFace Boutros et al. (2022) | non-cumulative | 5 | 0.0431 | 0.0862 | 0.0282 | 0.0304 | 0.0094 | 0.0115 | 0.0026 | 0.0033 | 0.0628 | 0.0641 | 0.0439 | 0.0577 | 0.1876 | 0.2329 | 0.0539 | 0.0694 |
| | | 10 | 0.0241 | 0.0460 | 0.0240 | 0.0264 | 0.0086 | 0.0107 | 0.0024 | 0.0031 | 0.0600 | 0.0617 | 0.0357 | 0.0481 | 0.1901 | 0.2218 | 0.0493 | 0.0597 |
| | | 15 | 0.0323 | 0.0594 | 0.0155 | 0.0170 | 0.0064 | 0.0093 | 0.0018 | 0.0025 | 0.0600 | 0.0616 | 0.0399 | 0.0520 | 0.1907 | 0.2318 | 0.0495 | 0.0619 |
| | | 20 | 0.0253 | 0.0454 | 0.0160 | 0.0171 | 0.0059 | 0.0084 | 0.0016 | 0.0023 | 0.0553 | 0.0576 | 0.0385 | 0.0621 | 0.1891 | 0.2838 | 0.0474 | 0.0681 |
| | | 25 | 0.0262 | 0.0473 | 0.0150 | 0.0158 | 0.0068 | 0.0094 | 0.0016 | 0.0023 | 0.0511 | 0.0531 | 0.0399 | 0.0510 | 0.2733 | 0.3326 | 0.0591 | 0.0731 |
| | | 30 | 0.0287 | 0.0523 | 0.0183 | 0.0195 | 0.0084 | 0.0115 | 0.0019 | 0.0025 | 0.0538 | 0.0551 | 0.0424 | 0.0545 | 0.2306 | 0.2974 | 0.0549 | 0.0704 |
| | cumulative | 5 | 0.0664 | 0.1331 | 0.0328 | 0.0351 | 0.0403 | 0.0500 | 0.0029 | 0.0036 | 0.0678 | 0.0698 | 0.1852 | 0.3531 | 0.4988 | 0.5649 | 0.1277 | 0.1728 |
| | | 10 | 0.0257 | 0.0466 | 0.0245 | 0.0261 | 0.0088 | 0.0111 | 0.0029 | 0.0036 | 0.0627 | 0.0644 | 0.0467 | 0.1021 | 0.2767 | 0.3014 | 0.0640 | 0.0793 |
| | | 15 | 0.0258 | 0.0461 | 0.0177 | 0.0186 | 0.0088 | 0.0118 | 0.0015 | 0.0022 | 0.0573 | 0.0593 | 0.0411 | 0.0521 | 0.1984 | 0.2447 | 0.0501 | 0.0621 |
| | | 20 | 0.0221 | 0.0383 | 0.0172 | 0.0185 | 0.0063 | 0.0087 | 0.0018 | 0.0025 | 0.0560 | 0.0575 | 0.0392 | 0.0628 | 0.1756 | 0.2165 | 0.0455 | 0.0578 |
| | | 25 | 0.0227 | 0.0394 | 0.0173 | 0.0184 | 0.0057 | 0.0080 | 0.0016 | 0.0024 | 0.0576 | 0.0599 | 0.0397 | 0.0632 | 0.2145 | 0.2904 | 0.0513 | 0.0688 |
| | | 30 | 0.0239 | 0.0412 | 0.0180 | 0.0192 | 0.0057 | 0.0078 | 0.0013 | 0.0022 | 0.0566 | 0.0581 | 0.0395 | 0.0634 | 0.2062 | 0.2477 | 0.0502 | 0.0628 |
| MagFace Meng et al. (2021a) | non-cumulative | 5 | 0.0402 | 0.0926 | 0.0324 | 0.0461 | 0.0121 | 0.0167 | 0.0029 | 0.0036 | 0.0645 | 0.0670 | 0.0480 | 0.1098 | 0.2428 | 0.2694 | 0.0633 | 0.0865 |
| | | 10 | 0.0226 | 0.0498 | 0.0293 | 0.0451 | 0.0098 | 0.0182 | 0.0026 | 0.0032 | 0.0621 | 0.0638 | 0.0389 | 0.0813 | 0.2492 | 0.2818 | 0.0592 | 0.0776 |
| | | 15 | 0.0285 | 0.0654 | 0.0182 | 0.0284 | 0.0081 | 0.0102 | 0.0022 | 0.0027 | 0.0615 | 0.0633 | 0.0432 | 0.0856 | 0.2474 | 0.2735 | 0.0584 | 0.0756 |
| | | 20 | 0.0235 | 0.0495 | 0.0190 | 0.0275 | 0.0071 | 0.0090 | 0.0025 | 0.0030 | 0.0566 | 0.0578 | 0.0414 | 0.0958 | 0.2432 | 0.2795 | 0.0562 | 0.0746 |
| | | 25 | 0.0237 | 0.0507 | 0.0187 | 0.0285 | 0.0082 | 0.0114 | 0.0023 | 0.0028 | 0.0526 | 0.0537 | 0.0423 | 0.0749 | 0.3100 | 0.3637 | 0.0654 | 0.0837 |
| | | 30 | 0.0260 | 0.0577 | 0.0201 | 0.0340 | 0.0105 | 0.0129 | 0.0025 | 0.0030 | 0.0540 | 0.0554 | 0.0466 | 0.0806 | 0.3007 | 0.3312 | 0.0658 | 0.0821 |
| | cumulative | 5 | 0.0624 | 0.1393 | 0.0378 | 0.0494 | 0.0578 | 0.0750 | 0.0032 | 0.0040 | 0.0695 | 0.0710 | 0.3001 | 0.5208 | 0.5532 | 0.5881 | 0.1549 | 0.2068 |
| | | 10 | 0.0240 | 0.0518 | 0.0285 | 0.0384 | 0.0104 | 0.0185 | 0.0032 | 0.0038 | 0.0648 | 0.0667 | 0.0547 | 0.1851 | 0.3245 | 0.3530 | 0.0729 | 0.1025 |
| | | 15 | 0.0246 | 0.0485 | 0.0209 | 0.0348 | 0.0125 | 0.0238 | 0.0019 | 0.0026 | 0.0585 | 0.0597 | 0.0443 | 0.0939 | 0.2504 | 0.2987 | 0.0590 | 0.0803 |
| | | 20 | 0.0211 | 0.0413 | 0.0204 | 0.0310 | 0.0076 | 0.0095 | 0.0023 | 0.0029 | 0.0566 | 0.0578 | 0.0419 | 0.0957 | 0.2381 | 0.2854 | 0.0554 | 0.0748 |
| | | 25 | 0.0219 | 0.0426 | 0.0205 | 0.0355 | 0.0065 | 0.0097 | 0.0019 | 0.0025 | 0.0593 | 0.0606 | 0.0422 | 0.0968 | 0.2504 | 0.2867 | 0.0575 | 0.0763 |
| | | 30 | 0.0228 | 0.0440 | 0.0213 | 0.0384 | 0.0070 | 0.0088 | 0.0017 | 0.0023 | 0.0576 | 0.0586 | 0.0426 | 0.0973 | 0.2505 | 0.2993 | 0.0576 | 0.0784 |
| CurricularFace Huang et al. (2020) | non-cumulative | 5 | 0.0341 | 0.0785 | 0.0323 | 0.0356 | 0.0111 | 0.0156 | 0.0028 | 0.0033 | 0.0637 | 0.0679 | 0.0416 | 0.0605 | 0.1798 | 0.2232 | 0.0522 | 0.0692 |
| | | 10 | 0.0205 | 0.0416 | 0.0288 | 0.0321 | 0.0094 | 0.0132 | 0.0026 | 0.0031 | 0.0601 | 0.0642 | 0.0336 | 0.0498 | 0.1956 | 0.2304 | 0.0501 | 0.0621 |
| | | 15 | 0.0258 | 0.0561 | 0.0192 | 0.0233 | 0.0075 | 0.0112 | 0.0021 | 0.0026 | 0.0608 | 0.0642 | 0.0365 | 0.0533 | 0.1825 | 0.2027 | 0.0478 | 0.0591 |
| | | 20 | 0.0214 | 0.0424 | 0.0178 | 0.0210 | 0.0078 | 0.0110 | 0.0024 | 0.0029 | 0.0563 | 0.0590 | 0.0362 | 0.0659 | 0.1748 | 0.2320 | 0.0452 | 0.0620 |
| | | 25 | 0.0220 | 0.0441 | 0.0170 | 0.0205 | 0.0086 | 0.0124 | 0.0023 | 0.0029 | 0.0525 | 0.0551 | 0.0374 | 0.0518 | 0.2554 | 0.2979 | 0.0565 | 0.0692 |
| | | 30 | 0.0240 | 0.0513 | 0.0198 | 0.0238 | 0.0102 | 0.0147 | 0.0030 | 0.0030 | 0.0545 | 0.0569 | 0.0397 | 0.0854 | 0.2309 | 0.2703 | 0.0545 | 0.0679 |
| | cumulative | 5 | 0.0529 | 0.1158 | 0.0382 | 0.0423 | 0.0492 | 0.0688 | 0.0032 | 0.0037 | 0.0682 | 0.0738 | 0.1814 | 0.3807 | 0.4678 | 0.4964 | 0.1230 | 0.1688 |
| | | 10 | 0.0219 | 0.0432 | 0.0288 | 0.0320 | 0.0095 | 0.0132 | 0.0031 | 0.0036 | 0.0632 | 0.0671 | 0.0452 | 0.1135 | 0.2696 | 0.3049 | 0.0630 | 0.0825 |
| | | 15 | 0.0226 | 0.0428 | 0.0198 | 0.0240 | 0.0110 | 0.0162 | 0.0019 | 0.0025 | 0.0577 | 0.0618 | 0.0382 | 0.0526 | 0.2005 | 0.2252 | 0.0502 | 0.0607 |
| | | 20 | 0.0196 | 0.0355 | 0.0190 | 0.0220 | 0.0080 | 0.0111 | 0.0024 | 0.0030 | 0.0569 | 0.0605 | 0.0370 | 0.0665 | 0.1667 | 0.1875 | 0.0442 | 0.0552 |
| | | 25 | 0.0202 | 0.0361 | 0.0189 | 0.0219 | 0.0068 | 0.0094 | 0.0019 | 0.0025 | 0.0589 | 0.0624 | 0.0376 | 0.0666 | 0.1937 | 0.2288 | 0.0483 | 0.0611 |
| | | 30 | 0.0213 | 0.0388 | 0.0195 | 0.0224 | 0.0071 | 0.0101 | 0.0017 | 0.0023 | 0.0574 | 0.0603 | 0.0373 | 0.0670 | 0.1886 | 0.2200 | 0.0476 | 0.0601 |

Table 4: The achieved AUC of EDC by using two approaches compared in this paper, non-cumulative and cumulative, with two different loss functions (ArcFace, CurricularFace) and under different settings for epoch 33. The results are reported under two operation thresholds, FMR= $1e-3$ and FMR= $1e-4$ and under two protocols.

| FR | Quality Metric | Loss | Adience Eidinger et al. (2014) | | AgeDB-30 Moschoglou et al. (2017) | | CFP-FP Sengupta et al. (2016) | | LFW Huang et al. (2007) | | CALFW Zheng et al. (2017) | | CPLFW Zheng & Deng (2018) | | XQLFW Knoche et al. (2021) | | Mean AUC | |
|---|---|---|---|---|---|---|---|---|---|---|---|---|---|---|---|---|---|---|
| | | | 1e−3 | 1e−4 | 1e−3 | 1e−4 | 1e−3 | 1e−4 | 1e−3 | 1e−4 | 1e−3 | 1e−4 | 1e−3 | 1e−4 | 1e−3 | 1e−4 | 1e−3 | 1e−4 |
| ArcFace Deng et al. (2019) | non-cumulative | ArcFaceDeng et al. (2019) | 0.0251 | 0.0577 | 0.0172 | 0.0224 | 0.0078 | 0.0135 | 0.0025 | 0.0030 | 0.0561 | 0.0598 | 0.0463 | 0.0661 | 0.2727 | 0.3155 | 0.0611 | 0.0769 |
| | | CurricularFaceHuang et al. (2020) | 0.0243 | 0.0576 | 0.0187 | 0.0226 | 0.0073 | 0.0118 | 0.0019 | 0.0025 | 0.0555 | 0.0611 | 0.0462 | 0.0640 | 0.2371 | 0.2766 | 0.0559 | 0.0709 |
| | cumulative | ArcFaceDeng et al. (2019) | 0.0221 | 0.0431 | 0.0190 | 0.0262 | 0.0059 | 0.0084 | 0.0026 | 0.0023 | 0.0587 | 0.0576 | 0.0608 | 0.0417 | 0.2382 | 0.2226 | 0.0549 | 0.0676 |
| | | CurricularFaceHuang et al. (2020) | 0.0220 | 0.0456 | 0.0199 | 0.0240 | 0.0066 | 0.0105 | 0.0026 | 0.0033 | 0.0548 | 0.0576 | 0.0415 | 0.0589 | 0.2226 | 0.2752 | 0.0529 | 0.0679 |
| ElasticFace Boutros et al. (2022) | non-cumulative | ArcFaceDeng et al. (2019) | 0.0284 | 0.0517 | 0.0179 | 0.0191 | 0.0084 | 0.0115 | 0.0019 | 0.0026 | 0.0549 | 0.0562 | 0.0427 | 0.0674 | 0.2479 | 0.3101 | 0.0574 | 0.0741 |
| | | CurricularFaceHuang et al. (2020) | 0.0268 | 0.0506 | 0.0189 | 0.0202 | 0.0095 | 0.0129 | 0.0017 | 0.0025 | 0.0530 | 0.0553 | 0.0427 | 0.0672 | 0.2171 | 0.2573 | 0.0528 | 0.0666 |
| | cumulative | ArcFaceDeng et al. (2019) | 0.0235 | 0.0403 | 0.0184 | 0.0197 | 0.0056 | 0.0077 | 0.0014 | 0.0023 | 0.0528 | 0.0541 | 0.0427 | 0.0513 | 0.2058 | 0.2461 | 0.0500 | 0.0608 |
| | | CurricularFaceHuang et al. (2020) | 0.0236 | 0.0416 | 0.0195 | 0.0204 | 0.0062 | 0.0088 | 0.0022 | 0.0030 | 0.0528 | 0.0541 | 0.0399 | 0.0513 | 0.2058 | 0.2461 | 0.0500 | 0.0608 |
| MagFace Meng et al. (2021a) | non-cumulative | ArcFaceDeng et al. (2019) | 0.0257 | 0.0560 | 0.0196 | 0.0341 | 0.0103 | 0.0128 | 0.0025 | 0.0031 | 0.0550 | 0.0565 | 0.0469 | 0.1039 | 0.3065 | 0.3488 | 0.0666 | 0.0879 |
| | | CurricularFaceHuang et al. (2020) | 0.0248 | 0.0559 | 0.0211 | 0.0324 | 0.0111 | 0.0156 | 0.0020 | 0.0027 | 0.0550 | 0.0563 | 0.0460 | 0.1024 | 0.3250 | 0.3250 | 0.0612 | 0.0843 |
| | cumulative | ArcFaceDeng et al. (2019) | 0.0226 | 0.0426 | 0.0214 | 0.0389 | 0.0070 | 0.0088 | 0.0017 | 0.0023 | 0.0579 | 0.0590 | 0.0429 | 0.0748 | 0.2537 | 0.3005 | 0.0580 | 0.0745 |
| | | CurricularFaceHuang et al. (2020) | 0.0225 | 0.0456 | 0.0214 | 0.0313 | 0.0078 | 0.0095 | 0.0027 | 0.0034 | 0.0552 | 0.0566 | 0.0429 | 0.0974 | 0.2666 | 0.3143 | 0.0599 | 0.0805 |
| CurricularFace Huang et al. (2020) | non-cumulative | ArcFaceDeng et al. (2019) | 0.0238 | 0.0511 | 0.0192 | 0.0231 | 0.0100 | 0.0149 | 0.0027 | 0.0033 | 0.0555 | 0.0581 | 0.0398 | 0.0707 | 0.2484 | 0.2758 | 0.0571 | 0.0710 |
| | | CurricularFaceHuang et al. (2020) | 0.0228 | 0.0493 | 0.0201 | 0.0231 | 0.0100 | 0.0154 | 0.0019 | 0.0025 | 0.0545 | 0.0590 | 0.0399 | 0.0694 | 0.2114 | 0.2551 | 0.0515 | 0.0677 |
| | cumulative | ArcFaceDeng et al. (2019) | 0.0210 | 0.0380 | 0.0198 | 0.0228 | 0.0070 | 0.0100 | 0.0017 | 0.0023 | 0.0579 | 0.0607 | 0.0370 | 0.0670 | 0.2078 | 0.2346 | 0.0503 | 0.0622 |
| | | CurricularFaceHuang et al. (2020) | 0.0208 | 0.0394 | 0.0215 | 0.0250 | 0.0072 | 0.0110 | 0.0027 | 0.0033 | 0.0547 | 0.0569 | 0.0371 | 0.0518 | 0.1945 | 0.2270 | 0.0484 | 0.0592 |

Table 5: The AUCs of EDC achieved by our *CARPM-FIQA* and the SOTA methods under different experimental settings. The notions of $1e-3$ and $1e-4$ indicate the value of the fixed FMR at which the EDC curves(FNMR vs. reject) were calculated. The results are compared to three IQA and eleven FIQA approaches. The XQLFW dataset uses SER-FIQ(marked with *) as FIQ labeling method.

| FR | | Method | Adience Eidinger et al. (2014) 1e-3 | 1e-4 | AgeDB-30 Moschoglou et al. (2017) 1e-3 | 1e-4 | CFP-FP Sengupta et al. (2016) 1e-3 | 1e-4 | LFW Huang et al. (2007) 1e-3 | 1e-4 | CALFW Zheng et al. (2017) 1e-3 | 1e-4 | CPLFW Zheng & Deng (2018) 1e-3 | 1e-4 | XQLFW Knoche et al. (2021) 1e-3 | 1e-4 | IJB-C Maze et al. (2018) 1e-3 | 1e-4 |
|---|---|---|---|---|---|---|---|---|---|---|---|---|---|---|---|---|---|---|
| AdaFaceDeng et al. (2019) | IQA | BRISQUEMittal et al. (2012) | 0.0565 | 0.1285 | 0.0400 | 0.0585 | 0.0343 | 0.0433 | 0.0043 | 0.0049 | 0.0755 | 0.0813 | 0.2558 | 0.3037 | 0.6680 | 0.7122 | 0.0381 | 0.0656 |
| | | RankIQALiu et al. (2017b) | 0.0400 | 0.0933 | 0.0372 | 0.0523 | 0.0301 | 0.0384 | 0.0039 | 0.0045 | 0.0846 | 0.0915 | 0.2437 | 0.2969 | 0.6584 | 0.7039 | 0.0385 | 0.0640 |
| | | DeepIQABosse et al. (2018) | 0.0568 | 0.1372 | 0.0403 | 0.0523 | 0.0238 | 0.0292 | 0.0049 | 0.0056 | 0.0793 | 0.0850 | 0.2309 | 0.2856 | 0.5958 | 0.6458 | 0.0383 | 0.0640 |
| | FIQA | RankIQChen et al. (2015) | 0.0353 | 0.0873 | 0.0322 | 0.0420 | 0.0152 | 0.0260 | 0.0018 | 0.0024 | 0.0608 | 0.0672 | 0.0633 | 0.0848 | 0.2789 | 0.3332 | 0.0227 | 0.0342 |
| | | PFEShi & Jain (2019) | 0.0212 | 0.0428 | 0.0172 | 0.0226 | 0.0092 | 0.0129 | 0.0023 | 0.0028 | 0.0647 | 0.0681 | 0.0450 | 0.0638 | 0.2302 | 0.2710 | 0.0176 | 0.0248 |
| | | SER-FIQTerhörst et al. (2020) | 0.0223 | 0.0434 | 0.0167 | 0.0223 | 0.0065 | 0.0103 | 0.0023 | 0.0028 | 0.0595 | 0.0627 | 0.0389 | 0.0584 | 0.1812(1)* | 0.2295(2)* | 0.0161 | 0.0241 |
| | | FaceQnetHernandez-Ortega et al. (2019; 2020) | 0.0346 | 0.0734 | 0.0197 | 0.0245 | 0.0240 | 0.0273 | 0.0022 | 0.0027 | 0.0774 | 0.0822 | 0.1504 | 0.1751 | 0.5829 | 0.6136 | 0.0270 | 0.0376 |
| | | MagFaceMeng et al. (2021a) | 0.0207(3) | 0.0425 | 0.0156 | 0.0198 | 0.0073 | 0.0105 | 0.0016(3) | 0.0021(2) | 0.0568(3) | 0.0602(2) | 0.0492 | 0.0642 | 0.4022 | 0.4636 | 0.0171 | 0.0254 |
| | | SDD-FIQAOu et al. (2021) | 0.0248 | 0.0562 | 0.0186 | 0.0206 | 0.0122 | 0.0193 | 0.0021 | 0.0027 | 0.0641 | 0.0698 | 0.0517 | 0.0670 | 0.3090 | 0.3561 | 0.0186 | 0.0270 |
| | | CR-FIQA(S) Boutros et al. (2023) | 0.0241 | 0.0517 | 0.0144(1) | 0.0187(3) | 0.0090 | 0.0145 | 0.0020 | 0.0025 | 0.0521(1) | 0.0554(1) | 0.0391 | 0.0567 | 0.2377 | 0.2740 | 0.0171 | 0.0250 |
| | | CR-FIQA(L) Boutros et al. (2023) | 0.0204(2) | 0.0353(1) | 0.0159 | 0.0189 | 0.0050(2) | 0.0082(1) | 0.0023 | 0.0029 | 0.0616 | 0.0632 | 0.0360(2) | 0.0515(2) | 0.2084 | 0.2441 | 0.0138(1) | 0.0207(1) |
| | | DifFIQA(R) Babnik et al. (2023b) | 0.0251 | 0.0619 | 0.0194 | 0.0262 | 0.0053 | 0.0091 | 0.0020 | 0.0025 | 0.0629 | 0.0688 | 0.0365(3) | 0.0531 | 0.1847(2) | 0.2397(3) | 0.0216 | 0.0323 |
| | | eDifFIQA(L) Babnik et al. (2024) | 0.0210 | 0.0402(3) | 0.0148(2) | 0.0176(1) | 0.0049(1) | 0.0083(2) | 0.0014(1) | 0.0019(1) | 0.0574 | 0.0627 | 0.0342(1) | 0.0500(1) | 0.1917(3) | 0.2469 | 0.0180 | 0.0269 |
| | | GraFIQs (S) Kolf et al. (2024) | 0.0258 | 0.0522 | 0.0286 | 0.0377 | 0.0142 | 0.0235 | 0.0028 | 0.0036 | 0.0732 | 0.0786 | 0.0607 | 0.0927 | 0.2409 | 0.2905 | 0.0206 | 0.0308 |
| | | GraFIQs (L) Kolf et al. (2024) | 0.0225 | 0.0403 | 0.0176 | 0.0219 | 0.0070 | 0.0111 | 0.0032 | 0.0038 | 0.0644 | 0.0692 | 0.0415 | 0.0612 | 0.2058 | 0.2447 | 0.0162 | 0.0237 |
| | | CLIB-FIQA Ou et al. (2024) | 0.0217 | 0.0424 | 0.0151(3) | 0.0180(2) | 0.0052 | 0.0089 | 0.0015(2) | 0.0019(1) | 0.0561(2) | 0.0620 | 0.0369 | 0.0530(3) | 0.1951 | 0.2277(1) | 0.0144(2) | 0.0212(2) |
| | | CARPM-FIQA (S) | 0.0223 | 0.044 | 0.0187 | 0.0261 | 0.0058 | 0.0084(3) | 0.0017 | 0.0023(3) | 0.0582 | 0.0619(3) | 0.0432 | 0.0607 | 0.2171 | 0.2681 | 0.0176 | 0.0258 |
| | | CARPM-FIQA (L) | 0.0191(1) | 0.0363(2) | 0.0171 | 0.0203 | 0.0051(3) | 0.0087 | 0.0018 | 0.0023(3) | 0.0616 | 0.0632 | 0.0379 | 0.0532 | 0.2008 | 0.2741 | 0.0153(3) | 0.0225(3) |
| ElasticFaceBoutros et al. (2022) | IQA | BRISQUEMittal et al. (2012) | 0.0644 | 0.1184 | 0.0375 | 0.0403 | 0.0281 | 0.0372 | 0.0034 | 0.0047 | 0.0726 | 0.0747 | 0.2641 | 0.4688 | 0.6343 | 0.6964 | 0.0357 | 0.0622 |
| | | RankIQALiu et al. (2017b) | 0.0433 | 0.0862 | 0.0374 | 0.0436 | 0.0269 | 0.0318 | 0.0033 | 0.0045 | 0.0810 | 0.0835 | 0.2325 | 0.4306 | 0.6189 | 0.6856 | 0.0366 | 0.0599 |
| | | DeepIQABosse et al. (2018) | 0.0645 | 0.1203 | 0.0384 | 0.0411 | 0.0191 | 0.0256 | 0.0043 | 0.0056 | 0.0756 | 0.0772 | 0.2401 | 0.4541 | 0.5400 | 0.5832 | 0.038 | 0.0599 |
| | FIQA | RankIQChen et al. (2015) | 0.0400 | 0.0777 | 0.0309 | 0.0337 | 0.0149 | 0.0180 | 0.0013(1) | 0.0020(3) | 0.0598 | 0.0614 | 0.0581 | 0.0727 | 0.2468 | 0.2776 | 0.0226 | 0.0334 |
| | | PFEShi & Jain (2019) | 0.0222 | 0.0381 | 0.0163 | 0.0172 | 0.0088 | 0.0113 | 0.0018 | 0.0025 | 0.0628 | 0.0643 | 0.0419 | 0.0895 | 0.2112 | 0.2436 | 0.0171 | 0.0247 |
| | | SER-FIQTerhörst et al. (2020) | 0.0240 | 0.0417 | 0.0163 | 0.0179 | 0.0061 | 0.0085 | 0.0021 | 0.0028 | 0.0574 | 0.0590 | 0.0387 | 0.0513 | 0.1576(2)* | 0.1868(1)* | 0.0156 | 0.0235 |
| | | FaceQnetHernandez-Ortega et al. (2019; 2020) | 0.0369 | 0.0667 | 0.0194 | 0.0207 | 0.0227 | 0.0247 | 0.0021 | 0.0026 | 0.0763 | 0.0777 | 0.1420 | 0.2880 | 0.5549 | 0.5844 | 0.0263 | 0.0370 |
| | | MagFaceMeng et al. (2021a) | 0.0225 | 0.0385 | 0.0150 | 0.0158(2) | 0.0069 | 0.0095 | 0.0014(2) | 0.0021 | 0.0553(2) | 0.0563(2) | 0.0474 | 0.0597 | 0.3973 | 0.4282 | 0.0166 | 0.0243 |
| | | SDD-FIQAOu et al. (2021) | 0.0277 | 0.0512 | 0.0187 | 0.0200 | 0.0098 | 0.0118 | 0.0019 | 0.0027 | 0.0624 | 0.0638 | 0.0493 | 0.0634 | 0.3052 | 0.3562 | 0.0183 | 0.0266 |
| | | CR-FIQA(S) Boutros et al. (2023) | 0.0257 | 0.0465 | 0.0146(2) | 0.0160 | 0.0070 | 0.0096 | 0.0015(3) | 0.0022 | 0.0509(1) | 0.0522(1) | 0.0383 | 0.0502 | 0.2093 | 0.2835 | 0.0167 | 0.0244 |
| | | CR-FIQA(L) Boutros et al. (2023) | 0.0214(2) | 0.0357(2) | 0.0149(3) | 0.0159(3) | 0.0045(2) | 0.0065(2) | 0.0018 | 0.0025 | 0.0594 | 0.0608 | 0.0350 | 0.0462 | 0.1798 | 0.2060 | 0.0135(3) | 0.0203(1) |
| | | DifFIQA(R) Babnik et al. (2023b) | 0.0278 | 0.0536 | 0.0194 | 0.0207 | 0.0050 | 0.0073 | 0.0019 | 0.0025 | 0.0616 | 0.0634 | 0.0330(2) | 0.0445(2) | 0.1599(1) | 0.1890(2) | 0.0212 | 0.0312 |
| | | eDifFIQA(L) Babnik et al. (2024) | 0.0222(3) | 0.0374 | 0.0139(1) | 0.0148(1) | 0.0043(1) | 0.0066(3) | 0.0014(2) | 0.0019(2) | 0.0564 | 0.0576 | 0.0323(1) | 0.0440(1) | 0.1688(3) | 0.1996(3) | 0.0178 | 0.0262 |
| | | GraFIQs (S) Kolf et al. (2024) | 0.0278 | 0.0500 | 0.0317 | 0.0353 | 0.0132 | 0.0165 | 0.0027 | 0.0036 | 0.0711 | 0.0732 | 0.0577 | 0.0852 | 0.2183 | 0.2555 | 0.0199 | 0.0307 |
| | | GraFIQs (L) Kolf et al. (2024) | 0.0233 | 0.0394 | 0.0182 | 0.0200 | 0.0070 | 0.0091 | 0.0029 | 0.0037 | 0.0614 | 0.0632 | 0.0393 | 0.0633 | 0.1930 | 0.2319 | 0.0158 | 0.0235 |
| | | CLIB-FIQA Ou et al. (2024) | 0.0229 | 0.0396 | 0.0151 | 0.0160 | 0.0048(3) | 0.0072 | 0.0013(1) | 0.0018(1) | 0.0558(3) | 0.0570(3) | 0.0347(3) | 0.0459(3) | 0.1740 | 0.2292 | 0.0141(2) | 0.0208(2) |
| | | CARPM-FIQA (S) | 0.0239 | 0.0412 | 0.0180 | 0.0192 | 0.0057 | 0.0078 | 0.0013(1) | 0.0022 | 0.0566 | 0.0581 | 0.0395 | 0.0634 | 0.2062 | 0.2477 | 0.0173 | 0.0253 |
| | | CARPM-FIQA (L) | 0.0199(1) | 0.0344 | 0.0161 | 0.0169 | 0.0045(2) | 0.0064(1) | 0.0015(3) | 0.0021 | 0.0597 | 0.0616 | 0.0359 | 0.0467 | 0.1759 | 0.2055 | 0.0148(3) | 0.0222(3) |
| MagFaceMeng et al. (2021a) | IQA | BRISQUEMittal et al. (2012) | 0.0594 | 0.1308 | 0.0442 | 0.0799 | 0.0422 | 0.0589 | 0.0043 | 0.0058 | 0.0758 | 0.0788 | 0.4649 | 0.6809 | 0.6911 | 0.7229 | 0.0462 | 0.0787 |
| | | RankIQALiu et al. (2017b) | 0.0407 | 0.0889 | 0.0370 | 0.0681 | 0.0369 | 0.0543 | 0.0041 | 0.0056 | 0.0829 | 0.0857 | 0.3251 | 0.6475 | 0.6706 | 0.7046 | 0.0462 | 0.0750 |
| | | DeepIQABosse et al. (2018) | 0.0571 | 0.1302 | 0.0417 | 0.0721 | 0.0322 | 0.0545 | 0.0048 | 0.0059 | 0.0787 | 0.0809 | 0.3672 | 0.6632 | 0.6162 | 0.6519 | 0.0474 | 0.0765 |
| | FIQA | RankIQChen et al. (2015) | 0.0359 | 0.0837 | 0.0361 | 0.0531 | 0.0213 | 0.0332 | 0.0019 | 0.0027 | 0.0602 | 0.0629 | 0.0659 | 0.1642 | 0.3076 | 0.3475 | 0.0270 | 0.0383 |
| | | PFEShi & Jain (2019) | 0.0215 | 0.0423 | 0.0192 | 0.0317 | 0.0107 | 0.0138 | 0.0023 | 0.0029 | 0.0640 | 0.0652 | 0.0449 | 0.1435 | 0.2615 | 0.2926 | 0.0200 | 0.0283 |
| | | SER-FIQTerhörst et al. (2020) | 0.0233 | 0.0451 | 0.0185 | 0.0293 | 0.0080 | 0.0139 | 0.0025 | 0.0033 | 0.0590 | 0.0607 | 0.0397 | 0.0821 | 0.2139(1)* | 0.2562(1)* | 0.0189 | 0.0270 |
| | | FaceQnetHernandez-Ortega et al. (2019; 2020) | 0.0365 | 0.0720 | 0.0217 | 0.0314 | 0.0271 | 0.0351 | 0.0022 | 0.0027 | 0.0763 | 0.0773 | 0.2988 | 0.5218 | 0.6016 | 0.6210 | 0.0305 | 0.0423 |
| | | MagFaceMeng et al. (2021a) | 0.0212(3) | 0.0417 | 0.0167 | 0.0195(2) | 0.0069 | 0.0129 | 0.0017(3) | 0.0023(3) | 0.0562(2) | 0.0578(2) | 0.0506 | 0.0887 | 0.4500 | 0.4900 | 0.0195 | 0.0307 |
| | | SDD-FIQAOu et al. (2021) | 0.0253 | 0.0562 | 0.0216 | 0.0305 | 0.0146 | 0.0201 | 0.0021 | 0.0028 | 0.0643 | 0.0657 | 0.0525 | 0.1188 | 0.3404 | 0.3928 | 0.0215 | 0.0307 |
| | | CR-FIQA(S) Boutros et al. (2023) | 0.0244 | 0.0507 | 0.0165(2) | 0.0234(1) | 0.0102 | 0.0121 | 0.0020 | 0.0028 | 0.0516(1) | 0.0528(1) | 0.0409 | 0.0840 | 0.2670 | 0.3336 | 0.0199 | 0.0284 |
| | | CR-FIQA(L) Boutros et al. (2023) | 0.0211(2) | 0.0372(2) | 0.0174 | 0.0235(2) | 0.0062(2) | 0.0080(1) | 0.0023 | 0.0028 | 0.0614 | 0.0628 | 0.0374(3) | 0.0679(1) | 0.2369 | 0.2839 | 0.0163(1) | 0.0236(1) |
| | | DifFIQA(R) Babnik et al. (2023b) | 0.0256 | 0.0585 | 0.0223 | 0.0363 | 0.0066(3) | 0.0150 | 0.0020 | 0.0025 | 0.0638 | 0.0660 | 0.0371(2) | 0.0851 | 0.2177(2) | 0.2642(3) | 0.0250 | 0.0356 |
| | | eDifFIQA(L) Babnik et al. (2024) | 0.0216 | 0.0403(3) | 0.0168(3) | 0.0246(3) | 0.0058(1) | 0.0121 | 0.0014(1) | 0.0019(1) | 0.0580 | 0.0595 | 0.0357(1) | 0.0810(3) | 0.2278 | 0.2792 | 0.0211 | 0.0301 |
| | | GraFIQs (S) Kolf et al. (2024) | 0.0274 | 0.0552 | 0.0319 | 0.0486 | 0.0206 | 0.0328 | 0.0029 | 0.0041 | 0.0736 | 0.0759 | 0.0673 | 0.1461 | 0.2728 | 0.3099 | 0.0249 | 0.0359 |
| | | GraFIQs (L) Kolf et al. (2024) | 0.0233 | 0.0419 | 0.0182 | 0.0253 | 0.0087 | 0.0186 | 0.0033 | 0.0041 | 0.0640 | 0.0652 | 0.0428 | 0.0987 | 0.2524 | 0.3018 | 0.0191 | 0.0273 |
| | | CLIB-FIQA Ou et al. (2024) | 0.0223 | 0.0431 | 0.0175 | 0.0261 | 0.0066(3) | 0.0136 | 0.0015(2) | 0.0021 | 0.0572(3) | 0.0574 | 0.0384 | 0.0845 | 0.2196(3) | 0.2725 | 0.0169(2) | 0.0240(2) |
| | | CARPM-FIQA (S) | 0.0238 | 0.0440 | 0.0213 | 0.0384 | 0.0070 | 0.0103 | 0.0017(3) | 0.0023 | 0.0618 | 0.0634 | 0.0426 | 0.0974 | 0.2297 | 0.2993 | 0.0202 | 0.0288 |
| | | CARPM-FIQA (L) | 0.0195(1) | 0.0361(1) | 0.0187 | 0.0267 | 0.0062(2) | 0.0103 | 0.0019 | 0.0023 | 0.0618 | 0.0634 | 0.0384 | 0.0681(2) | 0.2297 | 0.2635(2) | 0.0176(3) | 0.0253(3) |
| CurricularFaceHuang et al. (2020) | IQA | BRISQUEMittal et al. (2012) | 0.0502 | 0.1095 | 0.0433 | 0.0491 | 0.0323 | 0.0357 | 0.0041 | 0.0054 | 0.0755 | 0.0784 | 0.2709 | 0.5057 | 0.6146 | 0.6336 | 0.0363 | 0.0589 |
| | | RankIQALiu et al. (2017b) | 0.0359 | 0.0752 | 0.0394 | 0.0510 | 0.0298 | 0.0356 | 0.0039 | 0.0045 | 0.0806 | 0.0865 | 0.2346 | 0.4654 | 0.5900 | 0.6212 | 0.0361 | 0.0556 |
| | | DeepIQABosse et al. (2018) | 0.0492 | 0.1070 | 0.0407 | 0.0476 | 0.0227 | 0.0278 | 0.0050 | 0.0056 | 0.0764 | 0.0786 | 0.2488 | 0.4961 | 0.5165 | 0.5526 | 0.0376 | 0.0571 |
| | FIQA | RankIQChen et al. (2015) | 0.0314 | 0.0715 | 0.0365 | 0.0417 | 0.0186 | 0.0249 | 0.0018 | 0.0024 | 0.0590 | 0.0640 | 0.0541 | 0.0730 | 0.2449 | 0.2880 | 0.0220 | 0.0320 |
| | | PFEShi & Jain (2019) | 0.0198(2) | 0.0365 | 0.0197 | 0.0227 | 0.0100 | 0.0134 | 0.0024 | 0.0028 | 0.0630 | 0.0657 | 0.0402 | 0.0983 | 0.1982 | 0.2220 | 0.0170 | 0.0238 |
| | | SER-FIQTerhörst et al. (2020) | 0.0211 | 0.0381 | 0.0167 | 0.0193(1) | 0.0074 | 0.0111 | 0.0025 | 0.0040 | 0.0587 | 0.0610 | 0.0356 | 0.0520 | 0.1558(1)* | 0.1866(1)* | 0.0153 | 0.0228 |
| | | FaceQnetHernandez-Ortega et al. (2019; 2020) | 0.0326 | 0.0626 | 0.0221 | 0.0267 | 0.0226 | 0.0274 | 0.0022 | 0.0027 | 0.0767 | 0.0799 | 0.1384 | 0.3229 | 0.5035 | 0.5411 | 0.0259 | 0.0354 |
| | | MagFaceMeng et al. (2021a) | 0.0200 | 0.0364 | 0.0167 | 0.0195(2) | 0.0078 | 0.0111 | 0.0016(3) | 0.0021(2) | 0.0637 | 0.0675 | 0.0465 | 0.0607 | 0.3758 | 0.4178 | 0.0163 | 0.0232 |
| | | SDD-FIQAOu et al. (2021) | 0.0230 | 0.0462 | 0.0219 | 0.0254 | 0.0138 | 0.0185 | 0.0021 | 0.0027 | 0.0637 | 0.0671 | 0.0465 | 0.0671 | 0.2649 | 0.3053 | 0.0178 | 0.0255 |
| | | CR-FIQA(S) Boutros et al. (2023) | 0.0227 | 0.0446 | 0.0156(1) | 0.0198(3) | 0.0097 | 0.0148 | 0.0020 | 0.0025 | 0.0513(1) | 0.0534(1) | 0.0340 | 0.0501 | 0.2101 | 0.2470 | 0.0165 | 0.0234 |
| | | CR-FIQA(L) Boutros et al. (2023) | 0.0198(2) | 0.0336(2) | 0.0162(2) | 0.0200 | 0.0054(3) | 0.0080(1) | 0.0022 | 0.0028 | 0.0608 | 0.0617 | 0.0305(2) | 0.0441(1) | 0.1716 | 0.2318 | 0.0134(1) | 0.0194(1) |
| | | DifFIQA(R) Babnik et al. (2023b) | 0.0230 | 0.0475 | 0.0227 | 0.0260 | 0.0055 | 0.0092 | 0.0020 | 0.0025 | 0.0608 | 0.0667 | 0.0305(3) | 0.0441(2) | 0.1600(2) | 0.1871(2) | 0.0209 | 0.0303 |
| | | eDifFIQA(L) Babnik et al. (2024) | 0.0199(3) | 0.0338(3) | 0.0170 | 0.0195(2) | 0.0048(2) | 0.0084(2) | 0.0014(1) | 0.0019(1) | 0.0566(3) | 0.0601 | 0.0310 | 0.0455(2) | 0.1717 | 0.2080 | 0.0174 | 0.0251 |
| | | GraFIQs (S) Kolf et al. (2024) | 0.0246 | 0.0455 | 0.0306 | 0.0361 | 0.0153 | 0.0198 | 0.0029 | 0.0036 | 0.0703 | 0.0731 | 0.0516 | 0.0912 | 0.2029 | 0.2320 | 0.0194 | 0.0292 |
| | | GraFIQs (L) Kolf et al. (2024) | 0.0220 | 0.0365 | 0.0167(3) | 0.0200 | 0.0068 | 0.0099 | 0.0033 | 0.0038 | 0.0610 | 0.0641 | 0.0369 | 0.0663 | 0.1713 | 0.1959(3) | 0.0156 | 0.0223 |
| | | CLIB-FIQA Ou et al. (2024) | 0.0206 | 0.0406 | 0.0169 | 0.0204 | 0.0052(1) | 0.0091 | 0.0015(2) | 0.0019(1) | 0.0569 | 0.0595(3) | 0.0320(3) | 0.0458(3) | 0.1650(3) | 0.1951 | 0.0160 | 0.0201(2) |
| | | CARPM-FIQA (S) | 0.0213 | 0.0388 | 0.0195 | 0.0224 | 0.0071 | 0.0101 | 0.0017 | 0.0023(3) | 0.0574 | 0.0603 | 0.0373 | 0.0670 | 0.1886 | 0.2200 | 0.0171 | 0.0244 |
| | | CARPM-FIQA (L) | 0.0183(1) | 0.0330(1) | 0.0185 | 0.0215 | 0.0061 | 0.0087(3) | 0.0018 | 0.0023(3) | 0.0608 | 0.0623 | 0.0334 | 0.0470 | 0.1747 | 0.2065 | 0.0147(3) | 0.0213(3) |

Table 6: Comparison of pAUC-EDC values for non-cumulative and cumulative quality estimation approaches across different face recognition models and benchmarks. Lower pAUC-EDC values indicate better performance. Results are shown for FMR thresholds of $1e-3$ and $1e-4$. The best-performing epoch for each approach is highlighted in red for non-cumulative and blue for cumulative approaches. Final results are shown in violet to demonstrate the stability of cumulative averaging over time. While the cumulative approach generally outperforms non-cumulative methods across most datasets, we observe some limitations with cross-age (AgeDB-30, CALFW) and cross-pose (CPLFW) datasets, where performance is comparable but occasionally slightly worse at the final epoch.

| FR | Quality | Epoch | Adience (Eidinger et al. 2014) $1e-3$ | $1e-4$ | AgeDB-30 (Moschoglou et al. 2017) $1e-3$ | $1e-4$ | CFP-FP (Sengupta et al. 2016) $1e-3$ | $1e-4$ | LFW (Huang et al. 2007) $1e-3$ | $1e-4$ | CALFW (Zheng et al. 2017) $1e-3$ | $1e-4$ | CPLFW (Zheng & Deng 2018) $1e-3$ | $1e-4$ | XQLFW (Knoche et al. 2021) $1e-3$ | $1e-4$ | Mean pAUC $1e-3$ | $1e-4$ |
|---|---|---|---|---|---|---|---|---|---|---|---|---|---|---|---|---|---|---|
| ArcFace Deng et al. (2019) | non-cumulative | 5 | 0.0114 | 0.0278 | 0.0085 | 0.0112 | 0.0049 | 0.0084 | 0.0007 | 0.0008 | 0.0181 | 0.0202 | 0.0227 | 0.0343 | 0.1216 | 0.1383 | 0.0268 | 0.0344 |
| | | 10 | 0.0097 | 0.0252 | 0.0084 | 0.0124 | 0.0046 | 0.0074 | 0.0007 | 0.0008 | 0.0184 | 0.0206 | 0.0210 | 0.0332 | 0.1260 | 0.1484 | 0.0270 | 0.0354 |
| | | 15 | 0.0103 | 0.0255 | 0.0073 | 0.0111 | 0.0040 | 0.0074 | 0.0005 | 0.0007 | 0.0184 | 0.0205 | 0.0208 | 0.0330 | 0.1266 | 0.1472 | 0.0268 | 0.0351 |
| | | 20 | 0.0096 | 0.0242 | 0.0067 | 0.0084 | 0.0035 | 0.0066 | 0.0007 | 0.0008 | 0.0183 | 0.0201 | 0.0206 | 0.0341 | 0.1258 | 0.1474 | 0.0265 | 0.0345 |
| | | 25 | 0.0098 | 0.0248 | 0.0066 | 0.0089 | 0.0038 | 0.0068 | 0.0007 | 0.0009 | 0.0177 | 0.0196 | 0.0212 | 0.0326 | 0.1325 | 0.1484 | 0.0275 | 0.0346 |
| | | 30 | 0.0105 | 0.0262 | 0.0076 | 0.0113 | 0.0040 | 0.0074 | 0.0008 | 0.0009 | 0.0178 | 0.0196 | 0.0218 | 0.0340 | 0.1394 | 0.1597 | 0.0288 | 0.0370 |
| | cumulative | 5 | 0.0148 | 0.0335 | 0.0088 | 0.0120 | 0.0098 | 0.0121 | 0.0009 | 0.0011 | 0.0196 | 0.0216 | 0.0509 | 0.0647 | 0.1561 | 0.1723 | 0.0373 | 0.0453 |
| | | 10 | 0.0096 | 0.0246 | 0.0078 | 0.0099 | 0.0058 | 0.0085 | 0.0007 | 0.0009 | 0.0186 | 0.0209 | 0.0310 | 0.0448 | 0.1359 | 0.1527 | 0.0299 | 0.0375 |
| | | 15 | 0.0098 | 0.0242 | 0.0075 | 0.0114 | 0.0054 | 0.0086 | 0.0007 | 0.0008 | 0.0187 | 0.0208 | 0.0213 | 0.0331 | 0.1246 | 0.1357 | 0.0269 | 0.0335 |
| | | 20 | 0.0091 | 0.0231 | 0.0069 | 0.0102 | 0.0037 | 0.0068 | 0.0007 | 0.0008 | 0.0185 | 0.0205 | 0.0225 | 0.0345 | 0.1175 | 0.1300 | 0.0256 | 0.0323 |
| | | 25 | 0.0093 | 0.0235 | 0.0070 | 0.0111 | 0.0035 | 0.0060 | 0.0006 | 0.0007 | 0.0184 | 0.0204 | 0.0225 | 0.0344 | 0.1263 | 0.1436 | 0.0268 | 0.0342 |
| | | 30 | 0.0093 | 0.0238 | 0.0073 | 0.0112 | 0.0033 | 0.0053 | 0.0006 | 0.0008 | 0.0184 | 0.0205 | 0.0224 | 0.0347 | 0.1219 | 0.1473 | 0.0262 | 0.0348 |
| ElasticFace Boutros et al. (2022) | non-cumulative | 5 | 0.0128 | 0.0254 | 0.0085 | 0.0092 | 0.0044 | 0.0058 | 0.0006 | 0.0008 | 0.0171 | 0.0177 | 0.0217 | 0.0338 | 0.1174 | 0.1384 | 0.0261 | 0.0330 |
| | | 10 | 0.0109 | 0.0225 | 0.0079 | 0.0089 | 0.0042 | 0.0058 | 0.0006 | 0.0008 | 0.0179 | 0.0183 | 0.0198 | 0.0306 | 0.1121 | 0.1338 | 0.0248 | 0.0315 |
| | | 15 | 0.0118 | 0.0231 | 0.0068 | 0.0076 | 0.0039 | 0.0058 | 0.0004 | 0.0007 | 0.0179 | 0.0184 | 0.0197 | 0.0303 | 0.1189 | 0.1351 | 0.0256 | 0.0316 |
| | | 20 | 0.0109 | 0.0216 | 0.0066 | 0.0070 | 0.0034 | 0.0050 | 0.0006 | 0.0008 | 0.0174 | 0.0181 | 0.0194 | 0.0416 | 0.1210 | 0.1535 | 0.0256 | 0.0354 |
| | | 25 | 0.0111 | 0.0221 | 0.0065 | 0.0069 | 0.0034 | 0.0051 | 0.0006 | 0.0009 | 0.0168 | 0.0176 | 0.0202 | 0.0295 | 0.1319 | 0.1556 | 0.0272 | 0.0340 |
| | | 30 | 0.0118 | 0.0231 | 0.0073 | 0.0078 | 0.0039 | 0.0059 | 0.0007 | 0.0009 | 0.0172 | 0.0178 | 0.0205 | 0.0304 | 0.1277 | 0.1547 | 0.0270 | 0.0344 |
| | cumulative | 5 | 0.0165 | 0.0309 | 0.0088 | 0.0098 | 0.0092 | 0.0119 | 0.0008 | 0.0011 | 0.0187 | 0.0193 | 0.0439 | 0.1057 | 0.1462 | 0.1683 | 0.0349 | 0.0496 |
| | | 10 | 0.0110 | 0.0222 | 0.0076 | 0.0083 | 0.0053 | 0.0069 | 0.0006 | 0.0009 | 0.0180 | 0.0185 | 0.0278 | 0.0403 | 0.1314 | 0.1498 | 0.0288 | 0.0412 |
| | | 15 | 0.0110 | 0.0217 | 0.0070 | 0.0076 | 0.0044 | 0.0064 | 0.0006 | 0.0008 | 0.0179 | 0.0185 | 0.0203 | 0.0297 | 0.1169 | 0.1456 | 0.0254 | 0.0329 |
| | | 20 | 0.0101 | 0.0204 | 0.0067 | 0.0074 | 0.0035 | 0.0051 | 0.0006 | 0.0008 | 0.0179 | 0.0184 | 0.0199 | 0.0422 | 0.1127 | 0.1486 | 0.0245 | 0.0347 |
| | | 25 | 0.0103 | 0.0206 | 0.0069 | 0.0075 | 0.0031 | 0.0047 | 0.0005 | 0.0007 | 0.0176 | 0.0182 | 0.0200 | 0.0419 | 0.1266 | 0.1534 | 0.0264 | 0.0353 |
| | | 30 | 0.0105 | 0.0212 | 0.0070 | 0.0075 | 0.0031 | 0.0046 | 0.0005 | 0.0008 | 0.0177 | 0.0184 | 0.0199 | 0.0420 | 0.1213 | 0.1462 | 0.0257 | 0.0344 |
| MagFace Meng et al. (2021a) | non-cumulative | 5 | 0.0116 | 0.0277 | 0.0093 | 0.0160 | 0.0066 | 0.0107 | 0.0008 | 0.0010 | 0.0177 | 0.0186 | 0.0252 | 0.0779 | 0.1296 | 0.1460 | 0.0287 | 0.0426 |
| | | 10 | 0.0099 | 0.0250 | 0.0092 | 0.0181 | 0.0056 | 0.0120 | 0.0008 | 0.0010 | 0.0183 | 0.0192 | 0.0227 | 0.0548 | 0.1320 | 0.1486 | 0.0284 | 0.0398 |
| | | 15 | 0.0106 | 0.0253 | 0.0077 | 0.0162 | 0.0056 | 0.0071 | 0.0006 | 0.0008 | 0.0182 | 0.0190 | 0.0226 | 0.0545 | 0.1358 | 0.1513 | 0.0287 | 0.0392 |
| | | 20 | 0.0099 | 0.0242 | 0.0070 | 0.0122 | 0.0047 | 0.0061 | 0.0007 | 0.0009 | 0.0179 | 0.0184 | 0.0226 | 0.0687 | 0.1360 | 0.1556 | 0.0284 | 0.0409 |
| | | 25 | 0.0101 | 0.0245 | 0.0073 | 0.0132 | 0.0049 | 0.0076 | 0.0007 | 0.0008 | 0.0173 | 0.0178 | 0.0226 | 0.0465 | 0.1471 | 0.1616 | 0.0300 | 0.0389 |
| | | 30 | 0.0107 | 0.0256 | 0.0079 | 0.0167 | 0.0058 | 0.0076 | 0.0008 | 0.0010 | 0.0174 | 0.0183 | 0.0232 | 0.0479 | 0.1476 | 0.1552 | 0.0305 | 0.0389 |
| | cumulative | 5 | 0.0151 | 0.0343 | 0.0097 | 0.0167 | 0.0127 | 0.0221 | 0.0010 | 0.0013 | 0.0193 | 0.0201 | 0.0704 | 0.1646 | 0.1646 | 0.1759 | 0.0418 | 0.0621 |
| | | 10 | 0.0100 | 0.0248 | 0.0087 | 0.0139 | 0.0070 | 0.0131 | 0.0007 | 0.0010 | 0.0185 | 0.0195 | 0.0353 | 0.1500 | 0.1504 | 0.1647 | 0.0329 | 0.0553 |
| | | 15 | 0.0101 | 0.0238 | 0.0080 | 0.0171 | 0.0068 | 0.0140 | 0.0007 | 0.0009 | 0.0184 | 0.0190 | 0.0235 | 0.0619 | 0.1335 | 0.1554 | 0.0287 | 0.0417 |
| | | 20 | 0.0093 | 0.0227 | 0.0076 | 0.0148 | 0.0050 | 0.0064 | 0.0007 | 0.0008 | 0.0183 | 0.0189 | 0.0230 | 0.0691 | 0.1306 | 0.1603 | 0.0278 | 0.0419 |
| | | 25 | 0.0096 | 0.0230 | 0.0079 | 0.0164 | 0.0044 | 0.0072 | 0.0006 | 0.0008 | 0.0181 | 0.0187 | 0.0229 | 0.0690 | 0.1327 | 0.1533 | 0.0280 | 0.0412 |
| | | 30 | 0.0097 | 0.0237 | 0.0077 | 0.0164 | 0.0046 | 0.0059 | 0.0006 | 0.0007 | 0.0182 | 0.0186 | 0.0230 | 0.0693 | 0.1337 | 0.1630 | 0.0282 | 0.0425 |
| CurricularFace Huang et al. (2020) | non-cumulative | 5 | 0.0099 | 0.0233 | 0.0093 | 0.0113 | 0.0052 | 0.0077 | 0.0007 | 0.0008 | 0.0172 | 0.0188 | 0.0203 | 0.0366 | 0.1116 | 0.1262 | 0.0249 | 0.0321 |
| | | 10 | 0.0084 | 0.0203 | 0.0089 | 0.0109 | 0.0046 | 0.0066 | 0.0007 | 0.0008 | 0.0179 | 0.0192 | 0.0181 | 0.0322 | 0.1143 | 0.1294 | 0.0247 | 0.0313 |
| | | 15 | 0.0091 | 0.0213 | 0.0075 | 0.0095 | 0.0046 | 0.0067 | 0.0005 | 0.0007 | 0.0178 | 0.0193 | 0.0176 | 0.0317 | 0.1125 | 0.1247 | 0.0242 | 0.0306 |
| | | 20 | 0.0084 | 0.0199 | 0.0074 | 0.0090 | 0.0040 | 0.0059 | 0.0007 | 0.0008 | 0.0177 | 0.0191 | 0.0177 | 0.0450 | 0.1081 | 0.1293 | 0.0234 | 0.0327 |
| | | 25 | 0.0087 | 0.0202 | 0.0072 | 0.0088 | 0.0048 | 0.0072 | 0.0007 | 0.0009 | 0.0172 | 0.0184 | 0.0184 | 0.0305 | 0.1259 | 0.1365 | 0.0260 | 0.0317 |
| | | 30 | 0.0091 | 0.0217 | 0.0078 | 0.0097 | 0.0048 | 0.0072 | 0.0008 | 0.0009 | 0.0173 | 0.0184 | 0.0187 | 0.0315 | 0.1167 | 0.1319 | 0.0251 | 0.0316 |
| | cumulative | 5 | 0.0129 | 0.0288 | 0.0099 | 0.0125 | 0.0108 | 0.0145 | 0.0009 | 0.0011 | 0.0187 | 0.0204 | 0.0421 | 0.1163 | 0.1375 | 0.1493 | 0.0333 | 0.0490 |
| | | 10 | 0.0085 | 0.0201 | 0.0087 | 0.0104 | 0.0058 | 0.0080 | 0.0007 | 0.0009 | 0.0181 | 0.0196 | 0.0265 | 0.0932 | 0.1232 | 0.1359 | 0.0274 | 0.0412 |
| | | 15 | 0.0087 | 0.0200 | 0.0079 | 0.0099 | 0.0056 | 0.0081 | 0.0007 | 0.0008 | 0.0180 | 0.0195 | 0.0185 | 0.0309 | 0.1158 | 0.1284 | 0.0250 | 0.0311 |
| | | 20 | 0.0080 | 0.0188 | 0.0076 | 0.0093 | 0.0043 | 0.0061 | 0.0007 | 0.0008 | 0.0180 | 0.0194 | 0.0182 | 0.0455 | 0.1069 | 0.1213 | 0.0234 | 0.0316 |
| | | 25 | 0.0082 | 0.0188 | 0.0076 | 0.0093 | 0.0038 | 0.0055 | 0.0006 | 0.0007 | 0.0178 | 0.0192 | 0.0186 | 0.0456 | 0.1126 | 0.1293 | 0.0242 | 0.0326 |
| | | 30 | 0.0083 | 0.0196 | 0.0076 | 0.0094 | 0.0040 | 0.0059 | 0.0006 | 0.0008 | 0.0179 | 0.0194 | 0.0182 | 0.0457 | 0.1093 | 0.1284 | 0.0237 | 0.0327 |

Table 7: The achieved pAUC of EDC by using two approaches compared in this paper, non-cumulative and cumulative, with two different loss functions (ArcFace, CurricularFace) and under different settings for epoch 33. The results are reported under two operation thresholds, FMR$= 1e-3$ and FMR$= 1e-4$ and under two protocols.

| FR | Quality Metric | Loss | Adience (Eidinger et al. 2014) $1e-3$ | $1e-4$ | AgeDB-30 (Moschoglou et al. 2017) $1e-3$ | $1e-4$ | CFP-FP (Sengupta et al. 2016) $1e-3$ | $1e-4$ | LFW (Huang et al. 2007) $1e-3$ | $1e-4$ | CALFW (Zheng et al. 2017) $1e-3$ | $1e-4$ | CPLFW (Zheng & Deng 2018) $1e-3$ | $1e-4$ | XQLFW (Knoche et al. 2021) $1e-3$ | $1e-4$ | Mean pAUC $1e-3$ | $1e-4$ |
|---|---|---|---|---|---|---|---|---|---|---|---|---|---|---|---|---|---|---|
| ArcFace Deng et al. (2019) | non-cumulative | ArcFace Deng et al. (2019) | 0.0105 | 0.0262 | 0.0075 | 0.0112 | 0.0040 | 0.0076 | 0.0008 | 0.0009 | 0.0178 | 0.0196 | 0.0220 | 0.0360 | 0.1352 | 0.1597 | 0.0283 | 0.0373 |
| | | CurricularFace Huang et al. (2020) | 0.0103 | 0.0258 | 0.0073 | 0.0102 | 0.0044 | 0.0076 | 0.0009 | 0.0009 | 0.0181 | 0.0200 | 0.0229 | 0.0355 | 0.1266 | 0.1488 | 0.0272 | 0.0355 |
| | cumulative | ArcFace Deng et al. (2019) | 0.0093 | 0.0236 | 0.0072 | 0.0111 | 0.0034 | 0.0053 | 0.0006 | 0.0008 | 0.0184 | 0.0204 | 0.0208 | 0.0347 | 0.1255 | 0.1383 | 0.0265 | 0.0335 |
| | | CurricularFace Huang et al. (2020) | 0.0099 | 0.0249 | 0.0075 | 0.0110 | 0.0037 | 0.0064 | 0.0007 | 0.0009 | 0.0184 | 0.0203 | 0.0205 | 0.0323 | 0.1246 | 0.1490 | 0.0265 | 0.0350 |
| ElasticFace Boutros et al. (2022) | non-cumulative | ArcFace Deng et al. (2019) | 0.0118 | 0.0230 | 0.0072 | 0.0077 | 0.0040 | 0.0060 | 0.0007 | 0.0009 | 0.0170 | 0.0178 | 0.0202 | 0.0433 | 0.1186 | 0.1549 | 0.0256 | 0.0362 |
| | | CurricularFace Huang et al. (2020) | 0.0115 | 0.0230 | 0.0070 | 0.0075 | 0.0040 | 0.0059 | 0.0007 | 0.0009 | 0.0170 | 0.0179 | 0.0202 | 0.0427 | 0.1186 | 0.1402 | 0.0256 | 0.0340 |
| | cumulative | ArcFace Deng et al. (2019) | 0.0103 | 0.0210 | 0.0069 | 0.0074 | 0.0031 | 0.0046 | 0.0005 | 0.0008 | 0.0176 | 0.0183 | 0.0194 | 0.0291 | 0.1262 | 0.1515 | 0.0255 | 0.0333 |
| | | CurricularFace Huang et al. (2020) | 0.0111 | 0.0219 | 0.0072 | 0.0077 | 0.0034 | 0.0052 | 0.0006 | 0.0009 | 0.0177 | 0.0183 | 0.0194 | 0.0291 | 0.1194 | 0.1499 | 0.0255 | 0.0333 |
| MagFace Meng et al. (2021a) | non-cumulative | ArcFace Deng et al. (2019) | 0.0107 | 0.0255 | 0.0079 | 0.0167 | 0.0058 | 0.0077 | 0.0008 | 0.0010 | 0.0174 | 0.0184 | 0.0235 | 0.0708 | 0.1431 | 0.1572 | 0.0299 | 0.0425 |
| | | CurricularFace Huang et al. (2020) | 0.0105 | 0.0259 | 0.0079 | 0.0151 | 0.0059 | 0.0098 | 0.0009 | 0.0011 | 0.0177 | 0.0185 | 0.0234 | 0.0703 | 0.1343 | 0.1529 | 0.0287 | 0.0419 |
| | cumulative | ArcFace Deng et al. (2019) | 0.0096 | 0.0231 | 0.0077 | 0.0164 | 0.0046 | 0.0059 | 0.0006 | 0.0008 | 0.0182 | 0.0185 | 0.0223 | 0.0693 | 0.1404 | 0.1611 | 0.0282 | 0.0422 |
| | | CurricularFace Huang et al. (2020) | 0.0102 | 0.0249 | 0.0080 | 0.0164 | 0.0048 | 0.0061 | 0.0008 | 0.0011 | 0.0181 | 0.0185 | 0.0223 | 0.0461 | 0.1333 | 0.1537 | 0.0282 | 0.0382 |
| CurricularFace Huang et al. (2020) | non-cumulative | ArcFace Deng et al. (2019) | 0.0091 | 0.0217 | 0.0080 | 0.0097 | 0.0048 | 0.0072 | 0.0008 | 0.0009 | 0.0172 | 0.0184 | 0.0190 | 0.0469 | 0.1219 | 0.1345 | 0.0258 | 0.0342 |
| | | CurricularFace Huang et al. (2020) | 0.0091 | 0.0215 | 0.0075 | 0.0092 | 0.0050 | 0.0074 | 0.0008 | 0.0009 | 0.0174 | 0.0190 | 0.0186 | 0.0460 | 0.1148 | 0.1287 | 0.0247 | 0.0332 |
| | cumulative | ArcFace Deng et al. (2019) | 0.0082 | 0.0194 | 0.0075 | 0.0093 | 0.0039 | 0.0058 | 0.0006 | 0.0008 | 0.0178 | 0.0193 | 0.0181 | 0.0458 | 0.1155 | 0.1300 | 0.0245 | 0.0329 |
| | | CurricularFace Huang et al. (2020) | 0.0087 | 0.0206 | 0.0079 | 0.0096 | 0.0041 | 0.0063 | 0.0008 | 0.0009 | 0.0180 | 0.0193 | 0.0175 | 0.0299 | 0.1139 | 0.1295 | 0.0244 | 0.0309 |

Table 8: The pAUCs of EDC achieved by our *CARPM-FIQA* and the SOTA methods under different experimental settings. The notions of $1e-3$ and $1e-4$ indicate the value of the fixed FMR at which the EDC curves(FNMR vs. reject) were calculated. The results are compared to three IQA and eleven FIQA approaches. The XQLFW dataset uses SER-FIQ(marked with *) as FIQ labeling method.

| FR | | Method | Adience Eidinger et al. (2014) | | AgeDB-30 Moschoglou et al. (2017) | | CFP-FP Sengupta et al. (2016) | | LFW Huang et al. (2007) | | CALFW Zheng et al. (2017) | | CPLFW Zheng & Deng (2018) | | XQLFW Knoche et al. (2021) | | IJB-C Maze et al. (2018) | |
| --- | --- | --- | --- | --- | --- | --- | --- | --- | --- | --- | --- | --- | --- | --- | --- | --- | --- | --- |
| | | | 1e−3 | 1e−4 | 1e−3 | 1e−4 | 1e−3 | 1e−4 | 1e−3 | 1e−4 | 1e−3 | 1e−4 | 1e−3 | 1e−4 | 1e−3 | 1e−4 | 1e−3 | 1e−4 |
| ArcFaceDeng et al. (2019) | IQA | BRISQUEMittal et al. (2012) | 0.0143 | 0.0333 | 0.0096 | 0.0146 | 0.0095 | 0.0136 | 0.0009 | 0.0010 | 0.0200 | 0.0225 | 0.0501 | 0.0638 | 0.1512 | 0.1689 | 0.0072 | 0.0113 |
| | | RankIQALiu et al. (2017b) | 0.0124 | 0.0302 | 0.0087 | 0.0141 | 0.0088 | 0.0135 | 0.0009 | 0.0010 | 0.0209 | 0.0234 | 0.0506 | 0.0658 | 0.1532 | 0.1709 | 0.0071 | 0.0112 |
| | | DeepIQABosse et al. (2018) | 0.0145 | 0.0337 | 0.0093 | 0.0140 | 0.0088 | 0.0119 | 0.0009 | 0.0010 | 0.0207 | 0.0230 | 0.0504 | 0.0653 | 0.1487 | 0.1668 | 0.0072 | 0.0114 |
| | FIQA | RankIQChen et al. (2015) | 0.0125 | 0.0304 | 0.0090 | 0.0143 | 0.0071 | 0.0114 | **0.0006(1)** | 0.0008(2) | 0.0191 | 0.0215 | 0.0306 | 0.0427 | 0.1270 | 0.1500 | 0.0066 | 0.0102 |
| | | PFEShi & Jain (2019) | 0.0096 | 0.0242 | 0.0071 | 0.0109 | 0.0053 | 0.0082 | 0.0007(2) | 0.0008(2) | 0.0187 | 0.0206 | 0.0248 | 0.0398 | 0.1247 | 0.1523 | 0.0063 | 0.0094 |
| | | SER-FIQTerhörst et al. (2020) | 0.0102 | 0.0244 | 0.0066 | 0.0107 | 0.0035(3) | 0.0057(3) | 0.0007(2) | 0.0008(2) | 0.0187 | 0.0205 | 0.0199 | 0.0319 | **0.1175(1)*** | **0.1385(3)*** | **0.0056(1)** | 0.0087(2) |
| | | FaceQNetHernandez-Ortega et al. (2019; 2020) | 0.0130 | 0.0303 | 0.0076 | 0.0113 | 0.0077 | 0.0100 | 0.0008(3) | 0.0009(3) | 0.0196 | 0.0216 | 0.0428 | 0.0554 | 0.1523 | 0.1686 | 0.0071 | 0.0106 |
| | | MagFaceMeng et al. (2021a) | 0.0099 | 0.0247 | 0.0065(3) | 0.0098 | 0.0045 | 0.0068 | **0.0006(1)** | **0.0007(1)** | **0.0177(1)** | 0.0193(2) | 0.0249 | 0.0360 | 0.1359 | 0.1614 | 0.0061 | 0.0092 |
| | | SDD-FIQAOu et al. (2021) | 0.0104 | 0.0259 | 0.0073 | 0.0088(3) | 0.0068 | 0.0100 | 0.0007(2) | 0.0008(2) | 0.0188 | 0.0205 | 0.0279 | 0.0377 | 0.1356 | 0.1525 | 0.0061 | 0.0094 |
| | | CR-FIQA(S) Boutros et al. (2023) | 0.0101 | 0.0257 | 0.0068 | 0.0100 | 0.0042 | 0.0078 | **0.0006(1)** | 0.0008(2) | 0.0178(2) | 0.0197 | 0.0207 | 0.0324 | 0.1242 | 0.1391 | 0.0060 | 0.0092 |
| | | CR-FIQA(L) Boutros et al. (2023) | 0.0097 | **0.0201(1)** | 0.0066 | 0.0089 | 0.0035(3) | 0.0058 | **0.0007(1)** | 0.0009(3) | **0.0177(1)** | **0.0186(1)** | 0.0190 | **0.0307(1)** | 0.1213(3) | 0.1378(2) | 0.0057(2) | 0.0087(2) |
| | | DifFIQA(R) Babnik et al. (2023b) | 0.0100 | 0.0256 | 0.0076 | 0.0120 | 0.0036 | 0.0060 | **0.0006(1)** | 0.0007(1) | 0.0186 | 0.0208 | **0.0188(1)** | 0.0308(2) | 0.1193(2) | 0.1454 | 0.0061 | 0.0094 |
| | | eDifFIQA(L) Babnik et al. (2024) | **0.0090(1)** | 0.0230 | **0.0060(1)** | **0.0079(1)** | **0.0033(1)** | 0.0056(2) | **0.0006(1)** | 0.0007 | **0.0177(1)** | 0.0199 | 0.0189(2) | 0.0308(2) | 0.1223 | 0.1474 | 0.0058(3) | 0.0089 |
| | | GraFIQs (S) Kolf et al. (2024) | 0.0119 | 0.0274 | 0.0087 | 0.0138 | 0.0074 | 0.0117 | 0.0007(2) | 0.0010 | 0.0192 | 0.0218 | 0.0313 | 0.0434 | 0.1277 | 0.1495 | 0.0059 | 0.0091 |
| | | GraFIQs (L) Kolf et al. (2024) | 0.0093(2) | 0.0215(2) | 0.0067 | 0.0099 | 0.0040 | 0.0064 | 0.0007(2) | 0.0009(3) | 0.0181 | 0.0202 | 0.0208 | 0.0346 | 0.1262 | 0.1389 | 0.0059 | 0.0089 |
| | | CLIB-FIQA Ou et al. (2024) | 0.0095(3) | 0.0240 | 0.0062(2) | 0.0083(2) | 0.0037 | 0.0062 | **0.0006(1)** | **0.0007(1)** | 0.0179(3) | 0.0199 | 0.0192(3) | 0.0311 | 0.1221 | **0.1285(1)** | 0.0057(2) | **0.0086(1)** |
| | | *CARPM-FIQA* (S) | 0.0093(2) | 0.0238 | 0.0073 | 0.0112 | **0.0033(1)** | **0.0053(1)** | **0.0006(1)** | 0.0008(2) | 0.0185 | 0.0205 | 0.0224 | 0.0347 | 0.1219 | **0.1473** | 0.0061 | 0.0093 |
| | | *CARPM-FIQA* (L) | **0.0090(1)** | 0.0220(3) | 0.0065(3) | 0.0093 | 0.0034(2) | 0.0058 | 0.0007(2) | 0.0008(2) | 0.0185 | 0.0195(3) | 0.0194 | 0.0403(3) | 0.1249 | 0.1590 | 0.0058(3) | 0.0088(3) |
| ElasticFaceBoutros et al. (2022) | IQA | BRISQUEMittal et al. (2012) | 0.0160 | 0.0302 | 0.0090 | 0.0099 | 0.0082 | 0.0107 | 0.0007(3) | 0.0010 | 0.0195 | 0.0203 | 0.0427 | 0.1055 | 0.1412 | 0.1638 | 0.0069 | 0.0108 |
| | | RankIQALiu et al. (2017b) | 0.0138 | 0.0274 | 0.0085 | 0.0096 | 0.0082 | 0.0105 | 0.0007(3) | 0.0010 | 0.0203 | 0.0209 | 0.0433 | 0.1086 | 0.1428 | 0.1661 | 0.0068 | 0.0106 |
| | | DeepIQABosse et al. (2018) | 0.0162 | 0.0308 | 0.0088 | 0.0097 | 0.0074 | 0.0100 | 0.0008 | 0.0010 | 0.0201 | 0.0208 | 0.0431 | 0.1082 | 0.1379 | 0.1621 | 0.0070 | 0.0108 |
| | FIQA | RankIQChen et al. (2015) | 0.0139 | 0.0276 | 0.0089 | 0.0097 | 0.0067 | 0.0089 | **0.0005(1)** | 0.0008(2) | 0.0182 | 0.0188 | 0.0291 | 0.0394 | 0.1163 | 0.1342 | 0.0065 | 0.0100 |
| | | PFEShi & Jain (2019) | 0.0106 | 0.0211 | 0.0064 | 0.0069 | 0.0049 | 0.0065 | 0.0006(2) | 0.0008(2) | 0.0181 | 0.0186 | 0.0219 | 0.0682 | 0.1180 | 0.1401 | 0.0060 | 0.0091 |
| | | SER-FIQTerhörst et al. (2020) | 0.0114 | 0.0227 | 0.0064 | 0.0072 | 0.0031(2) | 0.0044(2) | 0.0006(2) | 0.0008(2) | 0.0177 | 0.0184 | 0.0185 | 0.0292 | **0.1057(1)*** | **0.1283(3)*** | **0.0054(1)** | **0.0083(1)** |
| | | FaceQNetHernandez-Ortega et al. (2019; 2020) | 0.0143 | 0.0274 | 0.0075 | 0.0082 | 0.0071 | 0.0084 | 0.0007(3) | 0.0009(3) | 0.0189 | 0.0196 | 0.0237 | 0.0345 | 0.1331 | 0.1445 | 0.0058 | 0.0089 |
| | | MagFaceMeng et al. (2021a) | 0.0110 | 0.0211 | 0.0060(2) | 0.0134 | 0.0057 | 0.0059 | **0.0005(1)** | **0.0007(1)** | 0.0173(3) | 0.0177(3) | 0.0237 | 0.0345 | 0.1331 | 0.1445 | 0.0059 | 0.0090 |
| | | SDD-FIQAOu et al. (2021) | 0.0115 | 0.0231 | 0.0074 | 0.0080 | 0.0054 | 0.0067 | 0.0006(2) | 0.0008(2) | 0.0181 | 0.0186 | 0.0255 | 0.0377 | 0.1336 | 0.1564 | 0.0059 | 0.0090 |
| | | CR-FIQA(S) Boutros et al. (2023) | 0.0112 | 0.0223 | 0.0067 | 0.0073 | 0.0038 | 0.0057 | **0.0005(1)** | 0.0008(2) | 0.0172(2) | 0.0176(2) | 0.0197 | 0.0301 | 0.1166 | 0.1411 | 0.0058 | 0.0087 |
| | | CR-FIQA(L) Boutros et al. (2023) | 0.0105 | 0.0206 | 0.0064 | 0.0069 | 0.0031(2) | 0.0045(3) | 0.0006(2) | 0.0009(3) | **0.0171(1)** | **0.0175(1)** | 0.0178(3) | 0.0275(2) | 0.1094(2) | **0.1265(1)** | 0.0055(2) | 0.0084(2) |
| | | DifFIQA(R) Babnik et al. (2023b) | 0.0113 | 0.0227 | 0.0076 | 0.0082 | 0.0031(2) | 0.0045(3) | 0.0006(2) | 0.0008(2) | 0.0180 | 0.0186 | **0.0174(1)** | **0.0271(1)** | 0.0942(2) | 0.1279(2) | 0.0060 | 0.0091 |
| | | eDifFIQA(L) Babnik et al. (2024) | 0.0099(2) | 0.0205(3) | **0.0058(1)** | **0.0062(1)** | **0.0029(1)** | 0.0044(2) | 0.0006(2) | **0.0007(1)** | **0.0171(1)** | 0.0176(2) | 0.0176(2) | 0.0280 | 0.1129(3) | 0.1322 | 0.0056(3) | 0.0086(3) |
| | | GraFIQs (S) Kolf et al. (2024) | 0.0132 | 0.0257 | 0.0088 | 0.0098 | 0.0067 | 0.0088 | 0.0007(3) | 0.0010 | 0.0185 | 0.0190 | 0.0271 | 0.0523 | 0.1255 | 0.1381 | 0.0057 | 0.0088 |
| | | GraFIQs (L) Kolf et al. (2024) | 0.0101(3) | 0.0203(2) | 0.0066 | 0.0073 | 0.0034 | 0.0047 | 0.0006(2) | 0.0008(2) | 0.0176 | 0.0181 | 0.0194 | 0.0415 | 0.1270 | 0.1528 | 0.0056(3) | 0.0086(3) |
| | | CLIB-FIQA Ou et al. (2024) | 0.0103 | 0.0214 | 0.0062 | 0.0066 | 0.0032(3) | 0.0048 | **0.0005(1)** | **0.0007(1)** | **0.0171(1)** | 0.0176(2) | 0.0179 | 0.0276(3) | 0.1140 | 0.1532 | 0.0056(3) | 0.0084(2) |
| | | *CARPM-FIQA* (S) | 0.0105 | 0.0212 | 0.0070 | 0.0075 | 0.0031(2) | 0.0046 | **0.0005(1)** | 0.0008(2) | 0.0177 | 0.0184 | 0.0199 | 0.0420 | 0.1213 | 0.1462 | 0.0059 | 0.0090 |
| | | *CARPM-FIQA* (L) | **0.0096(1)** | **0.0195(1)** | 0.0061(3) | 0.0065(3) | **0.0029(1)** | **0.0042(1)** | 0.0006(2) | 0.0008(2) | 0.0179 | 0.0184 | 0.0182 | 0.0278 | 0.1159 | 0.1310 | 0.0055(2) | 0.0086(3) |
| MagFaceMeng et al. (2021a) | IQA | BRISQUEMittal et al. (2012) | 0.0148 | 0.0334 | 0.0101 | 0.0207 | 0.0117 | 0.0205 | 0.0009 | 0.0013 | 0.0199 | 0.0211 | 0.0700 | 0.1672 | 0.1601 | 0.1727 | 0.0084 | 0.0131 |
| | | RankIQALiu et al. (2017b) | 0.0128 | 0.0291 | 0.0092 | 0.0212 | 0.0119 | 0.0207 | 0.0009 | 0.0013 | 0.0208 | 0.0217 | 0.0518 | 0.1695 | 0.1619 | 0.1744 | 0.0083 | 0.0130 |
| | | DeepIQABosse et al. (2018) | 0.0149 | 0.0335 | 0.0100 | 0.0204 | 0.0111 | 0.0199 | 0.0009 | 0.0014 | 0.0206 | 0.0215 | 0.0710 | 0.1691 | 0.1576 | 0.1706 | 0.0085 | 0.0131 |
| | FIQA | RankIQChen et al. (2015) | 0.0125 | 0.0302 | 0.0100 | 0.0199 | 0.0096 | 0.0178 | 0.0007(2) | 0.0010 | 0.0188 | 0.0198 | 0.0336 | 0.1133 | 0.1392 | 0.1514 | 0.0077 | 0.0117 |
| | | PFEShi & Jain (2019) | 0.0098 | 0.0239 | 0.0074 | 0.0161 | 0.0066 | 0.0092 | 0.0007(2) | 0.0008(2) | 0.0186 | 0.0192 | 0.0253 | 0.1178 | 0.1386 | 0.1558 | 0.0072 | 0.0106 |
| | | SER-FIQTerhörst et al. (2020) | 0.0107 | 0.0241 | 0.0074 | 0.0160 | 0.0045(3) | 0.0099 | 0.0007(2) | 0.0011 | 0.0183 | 0.0187 | 0.0219 | 0.0541(3) | **0.1264(1)*** | **0.1440(1)*** | **0.0066(1)** | **0.0097(1)** |
| | | FaceQNetHernandez-Ortega et al. (2019; 2020) | 0.0133 | 0.0292 | 0.0082 | 0.0159 | 0.0096 | 0.0162 | 0.0008(3) | 0.0010 | 0.0193 | 0.0194 | 0.0602 | 0.1589 | 0.1496 | 0.1603 | 0.0070 | 0.0104 |
| | | MagFaceMeng et al. (2021a) | 0.0100 | 0.0233 | **0.0066(1)** | 0.0134 | 0.0057 | 0.0096 | **0.0006(1)** | 0.0008(2) | 0.0178(2) | 0.0184(3) | 0.0268 | 0.0579 | 0.1458 | 0.1652 | 0.0070 | 0.0104 |
| | | SDD-FIQAOu et al. (2021) | 0.0106 | 0.0257 | 0.0081 | 0.0122(3) | 0.0083 | 0.0128 | 0.0007(2) | 0.0009(3) | 0.0186 | 0.0194 | 0.0284 | 0.0834 | 0.1525 | 0.1656 | 0.0071 | 0.0106 |
| | | CR-FIQA(S) Boutros et al. (2023) | 0.0103 | 0.0246 | 0.0074 | 0.0137 | 0.0055 | 0.0068(3) | 0.0007(2) | 0.0010 | **0.0177(1)** | **0.0182(1)** | 0.0225 | 0.0548 | 0.1339 | 0.1619 | 0.0070 | 0.0102 |
| | | CR-FIQA(L) Boutros et al. (2023) | 0.0100 | **0.0210(1)** | 0.0071 | 0.0128 | 0.0048 | 0.0061(2) | 0.0007(2) | 0.0008(2) | **0.0177(1)** | 0.0183(2) | 0.0209(2) | 0.0454(2) | 0.1296 | 0.1506 | 0.0067(2) | 0.0099(3) |
| | | DifFIQA(R) Babnik et al. (2023b) | 0.0103 | 0.0251 | 0.0086 | 0.0171 | 0.0047 | 0.0104 | **0.0006(1)** | 0.0009(3) | 0.0185 | 0.0194 | **0.0207(1)** | 0.0598 | 0.1280(2) | 0.1501(3) | 0.0072 | 0.0107 |
| | | eDifFIQA(L) Babnik et al. (2024) | 0.0093(2) | 0.0224(3) | 0.0067(2) | **0.0113(1)** | **0.0042(1)** | 0.0097 | **0.0006(1)** | 0.0008(2) | **0.0177(1)** | **0.0182(1)** | 0.0213(3) | 0.0596 | 0.1323 | 0.1498(2) | 0.0068(3) | 0.0102 |
| | | GraFIQs (S) Kolf et al. (2024) | 0.0125 | 0.0273 | 0.0097 | 0.0204 | 0.0100 | 0.0181 | 0.0008(3) | 0.0013 | 0.0191 | 0.0201 | 0.0344 | 0.1017 | 0.1386 | 0.1557 | 0.0070 | 0.0104 |
| | | GraFIQs (L) Kolf et al. (2024) | 0.0079(3) | 0.0217(2) | 0.0070 | 0.0136 | 0.0049 | 0.0111 | 0.0008(3) | 0.0011 | 0.0179(3) | 0.0186 | 0.0230 | 0.0603 | 0.1414 | 0.1576 | 0.0068(3) | 0.0101(3) |
| | | CLIB-FIQA Ou et al. (2024) | 0.0098 | 0.0234 | 0.0071 | 0.0119(2) | 0.0048 | 0.0105 | 0.0007(2) | 0.0009(3) | **0.0177(1)** | 0.0183 | 0.0212(3) | 0.0599 | 0.1284(3) | 0.1554 | 0.0067(2) | 0.0098(2) |
| | | *CARPM-FIQA* (S) | 0.0097(3) | 0.0237 | 0.0077 | 0.0164 | 0.0046 | **0.0059(1)** | **0.0006(1)** | **0.0007(1)** | 0.0182 | 0.0186 | 0.0230 | 0.0693 | 0.1337 | 0.1630 | 0.0070 | 0.0104 |
| | | *CARPM-FIQA* (L) | **0.0092(1)** | **0.0210(1)** | 0.0070(3) | 0.0138 | 0.0044(2) | 0.0080 | 0.0007(2) | 0.0008(2) | 0.0185 | 0.0192 | 0.0212(3) | 0.0451(1) | 0.1368 | 0.1568 | 0.0067(2) | 0.0101(3) |
| CurricularFaceHuang et al. (2020) | IQA | BRISQUEMittal et al. (2012) | 0.0126 | 0.0283 | 0.0100 | 0.0124 | 0.0093 | 0.0107 | 0.0009 | 0.0011 | 0.0196 | 0.0211 | 0.0409 | 0.1172 | 0.1346 | 0.1444 | 0.0068 | 0.0102 |
| | | RankIQALiu et al. (2017b) | 0.0109 | 0.0254 | 0.0091 | 0.0122 | 0.0095 | 0.0127 | 0.0009 | 0.0010 | 0.0202 | 0.0219 | 0.0411 | 0.1198 | 0.1361 | 0.1461 | 0.0067 | 0.0102 |
| | | DeepIQABosse et al. (2018) | 0.0127 | 0.0284 | 0.0097 | 0.0117 | 0.0087 | 0.0117 | 0.0009 | 0.0010 | 0.0201 | 0.021 | 0.0412 | 0.1198 | 0.1297 | 0.1416 | 0.0069 | 0.0102 |
| | FIQA | RankIQChen et al. (2015) | 0.0107 | 0.0247 | 0.0096 | 0.0118 | 0.0078 | 0.0107 | **0.0006(1)** | 0.0008(2) | 0.0182 | 0.0200 | 0.0275 | 0.0402 | 0.1129 | 0.1292 | 0.0064 | 0.0094 |
| | | PFEShi & Jain (2019) | 0.0091 | 0.0207 | 0.0067(3) | 0.0089 | 0.0035(2) | **0.0053(1)** | 0.0007(2) | 0.0008(2) | 0.0183 | 0.0195 | 0.0208 | 0.0772 | **0.1048(2)** | **0.1215(1)** | 0.0066 | 0.0087 |
| | | SER-FIQTerhörst et al. (2020) | 0.0091 | 0.0207 | 0.0067(3) | 0.0083 | 0.0035(2) | **0.0053(1)** | 0.0007(2) | 0.0008(2) | 0.0179 | 0.0192 | 0.0169 | 0.0308 | **0.1054(3)*** | **0.1217(2)*** | **0.0054(1)** | **0.0079(1)** |
| | | FaceQNetHernandez-Ortega et al. (2019; 2020) | 0.0116 | 0.0254 | 0.0082 | 0.0101 | 0.0074 | 0.0099 | 0.0008(3) | 0.0009(3) | 0.0191 | 0.0204 | 0.0355 | 0.1066 | 0.1322 | 0.1459 | 0.0067 | 0.0097 |
| | | MagFaceMeng et al. (2021a) | 0.0089 | 0.0198 | 0.0066(2) | 0.0082(3) | 0.0048 | 0.0068 | **0.0006(1)** | **0.0007(1)** | 0.0175 | 0.0185(2) | 0.0219 | 0.0357 | 0.1257 | 0.1373 | 0.0057 | 0.0085 |
| | | SDD-FIQAOu et al. (2021) | 0.0091 | 0.0212 | 0.0080 | 0.0098 | 0.0073 | 0.0097 | 0.0007(2) | 0.0008(2) | 0.0183 | 0.0197 | 0.0237 | 0.0413 | 0.1219 | 0.1372 | 0.0059 | 0.0085 |
| | | CR-FIQA(S) Boutros et al. (2023) | 0.0088 | 0.0211 | 0.0071 | 0.0089 | 0.0047 | 0.0071 | **0.0006(1)** | 0.0008(2) | 0.0174(3) | 0.0186(3) | 0.0176 | 0.0317 | 0.1138 | 0.1301 | 0.0058 | 0.0083 |
| | | CR-FIQA(L) Boutros et al. (2023) | 0.0089 | 0.0189(3) | 0.0066(2) | 0.0083 | 0.0038 | 0.0056(3) | 0.0007(2) | 0.0009(3) | 0.0175 | **0.0181(1)** | 0.0161(3) | 0.0283(2) | **0.1043(1)** | 0.1279 | **0.0054(1)** | **0.0079(1)** |
| | | DifFIQA(R) Babnik et al. (2023b) | 0.0087 | 0.0206 | 0.0087 | 0.0104 | 0.0034(1) | 0.0054(2) | **0.0006(1)** | **0.0007(1)** | 0.0178 | 0.0196 | **0.0158(1)** | **0.0280(1)** | 0.1109 | 0.1250 | 0.0059 | 0.0086 |
| | | eDifFIQA(L) Babnik et al. (2024) | **0.0079(1)** | **0.0182(1)** | 0.0067(3) | 0.0081(2) | **0.0034(1)** | **0.0053(1)** | **0.0006(1)** | **0.0007(1)** | **0.0172(1)** | 0.0186(3) | 0.0160(2) | 0.0299 | 0.1129 | 0.1267 | 0.0056(3) | 0.0082(3) |
| | | GraFIQs (S) Kolf et al. (2024) | 0.0107 | 0.0235 | 0.0089 | 0.0116 | 0.0077 | 0.0106 | 0.0008(3) | 0.0010 | 0.0185 | 0.0200 | 0.0253 | 0.0577 | 0.1173 | 0.1300 | 0.0057 | 0.0089 |
| | | GraFIQs (L) Kolf et al. (2024) | 0.0085 | 0.0186(2) | **0.0065(1)** | **0.0080(1)** | 0.0037(3) | 0.0056(3) | 0.0007(2) | 0.0009(3) | 0.0176 | 0.0190 | 0.0180 | 0.0453 | 0.1097 | 0.1225(3) | 0.0056(3) | 0.0081(2) |
| | | CLIB-FIQA Ou et al. (2024) | 0.0084 | 0.0190 | 0.0070 | 0.0085 | 0.0037(3) | 0.0057 | **0.0006(1)** | 0.0008(2) | 0.0173(2) | 0.0186(3) | 0.0162 | 0.0285(3) | 0.1100 | 0.1275 | 0.0055(2) | 0.0081(2) |
| | | *CARPM-FIQA* (S) | 0.0083(3) | 0.0196 | 0.0076 | 0.0094 | 0.0040 | 0.0059 | **0.0006(1)** | 0.0008(2) | 0.0179 | 0.0194 | 0.0182 | 0.0457 | 0.1093 | 0.1284 | 0.0059 | 0.0086 |
| | | *CARPM-FIQA* (L) | 0.0081(2) | 0.0190 | 0.0068 | 0.0083 | 0.0037(3) | **0.0053(1)** | 0.0007(2) | 0.0008(2) | 0.0182 | 0.0189 | 0.0165 | 0.0286 | 0.1123 | 0.1283 | 0.0055(2) | 0.0081(2) |

Table 9: Verification performance on the IJB-C (1:1 mixed verification) Maze et al. (2018).

| | Quality Estimation | 1:1 mixed Verification: TAR (%) at | | | | | | | | | | | |
| --- | --- | --- | --- | --- | --- | --- | --- | --- | --- | --- | --- | --- | --- |
| | | ArcFace Deng et al. (2019) | | | ElasticFace Boutros et al. (2022) | | | MagFace Meng et al. (2021a) | | | CurricularFace Huang et al. (2020) | | |
| | | FAR=1e-6 | FAR=1e-5 | FAR=1e-4 | FAR=1e-6 | FAR=1e-5 | FAR=1e-4 | FAR=1e-6 | FAR=1e-5 | FAR=1e-4 | FAR=1e-6 | FAR=1e-5 | FAR=1e-4 |
| | Quality-based aggregation is not used. | 89.85 | 94.47 | 96.28 | 89.15 | 94.54 | 96.49 | 86.57 | 93.08 | 96.65 | 90.46 | 94.89 | 96.58 |
| IQA | BRISQUEMittal et al. (2012) | 86.65 | 93.62 | 95.98 | 85.68 | 93.51 | 95.65 | 81.11 | 90.64 | 94.82 | 88.16 | 93.98 | 96.29 |
| | RankIQALiu et al. (2017b) | 86.37 | 93.61 | 95.83 | 86.71 | 93.46 | 96.00 | 80.78 | 90.75 | 94.86 | 88.16 | 94.11 | 96.22 |
| | DeepIQABosse et al. (2018) | 81.97 | 91.64 | 94.67 | 78.93 | 91.59 | 94.81 | 73.53 | 86.34 | 92.90 | 82.65 | 92.04 | 95.00 |
| FIQA | RankIQChen et al. (2015) | 88.78 | 94.42 | 96.28 | 88.88 | 94.64 | 96.45 | 85.63 | 92.66 | 95.70 | 90.00 | 94.93 | 96.53 |
| | PFEShi & Jain (2019) | 89.50 | 94.51 | 96.31 | 89.10 | 94.67 | 96.51 | 84.93 | 92.44 | 95.60 | 90.36 | 95.04 | 96.54 |
| | SER-FIQTerhörst et al. (2020) | 89.74 | 94.65 | 96.44 | 90.05 | 94.79 | 96.57 | 86.02 | 93.35 | 95.80 | 90.54 | 95.11 | 96.58 |
| | FaceQNetHernandez-Ortega et al. (2019; 2020) | 87.87 | 94.04 | 96.12 | 86.26 | 94.09 | 96.25 | 82.91 | 90.56 | 95.03 | 89.61 | 94.65 | 96.36 |
| | MagFaceMeng et al. (2021a) | 89.49 | 94.41 | 96.24 | 89.37 | 94.69 | 96.46 | 85.75 | 92.71 | 95.54 | 90.34 | 95.02 | 96.50 |
| | SDD-FIQAOu et al. (2021) | 89.39 | 94.61 | 96.34 | 88.07 | 94.82 | 96.49 | 84.69 | 92.83 | 95.73 | 89.91 | 95.12 | 96.63 |
| | CR-FIQA(S)Boutros et al. (2023) | 89.59 | 94.78 | 96.33 | 90.30 | 94.97 | 96.63 | 86.45 | 93.48 | 95.95 | 90.82 | 95.13 | 96.64 |
| | CR-FIQA(L)Boutros et al. (2023) | 90.16 | 94.75 | 96.36 | 90.00 | 94.92 | 96.58 | 87.12 | 93.67 | 95.90 | 90.79 | 95.12 | 96.58 |
| | DifFIQA(R)Babnik et al. (2023b) | 88.14 | 93.95 | 96.18 | 88.36 | 94.14 | 96.32 | 84.11 | 92.26 | 95.59 | 88.91 | 94.60 | 96.36 |
| | eDifFIQA(L)Babnik et al. (2024) | 88.60 | 94.24 | 96.24 | 89.03 | 94.47 | 96.42 | 85.21 | 92.68 | 95.74 | 89.72 | 94.82 | 96.50 |
| | GraFIQs (S)Kolf et al. (2024) | 85.07 | 93.41 | 95.93 | 85.00 | 93.80 | 96.14 | 82.29 | 90.95 | 95.23 | 87.78 | 94.24 | 96.36 |
| | GraFIQs (L)Kolf et al. (2024) | 86.31 | 93.48 | 95.95 | 85.20 | 93.83 | 96.21 | 82.71 | 91.26 | 95.30 | 88.08 | 94.31 | 96.38 |
| | CLIB-FIQAOu et al. (2024) | 89.10 | 94.79 | 96.37 | 90.35 | 95.01 | 96.58 | 87.23 | 93.99 | 96.05 | 90.56 | 95.22 | 96.59 |
| | *CARPM-FIQA* (S) | 88.56 | 94.32 | 96.28 | 88.43 | 94.56 | 96.47 | 85.23 | 92.87 | 92.87 | 89.04 | 94.94 | 96.51 |
| | *CARPM-FIQA* (L) | 89.66 | 94.55 | 96.34 | 89.83 | 94.72 | 96.57 | 86.50 | 93.54 | 95.93 | 89.66 | 95.02 | 96.57 |

Table 10: The verification performances of *CARPM-FIQA* (L), only used as a feature extraction model, are compared to the recent SOTA face recognition models on mainstream benchmarks.

| Model | LFW Huang et al. (2007) Acc(%) | AgeDB-30 Moschoglou et al. (2017) Acc(%) | CFP-FP Sengupta et al. (2016) Acc(%) | CALFW Zheng et al. (2017) Acc(%) | CPLFW Zheng & Deng (2018) Acc(%) | IJB-C Maze et al. (2018) TAR@FAR=1e-4 |
| --- | --- | --- | --- | --- | --- | --- |
| ArcFaceDeng et al. (2019) | 99.82 | 98.15 | 98.27 | 95.45 | 92.08 | 96.28 |
| ElasticFaceBoutros et al. (2022) | 99.80 | 98.35 | 98.67 | 96.17 | 93.27 | 96.49 |
| MagFaceMeng et al. (2021a) | 99.83 | 98.17 | 98.46 | 96.15 | 92.87 | 96.65 |
| CurricularFaceHuang et al. (2020) | 99.80 | 98.32 | 98.37 | 96.20 | 93.13 | 96.58 |
| *CARPM-FIQA* (L) | 99.83 | 98.20 | 98.51 | 96.08 | 92.88 | 96.12 |

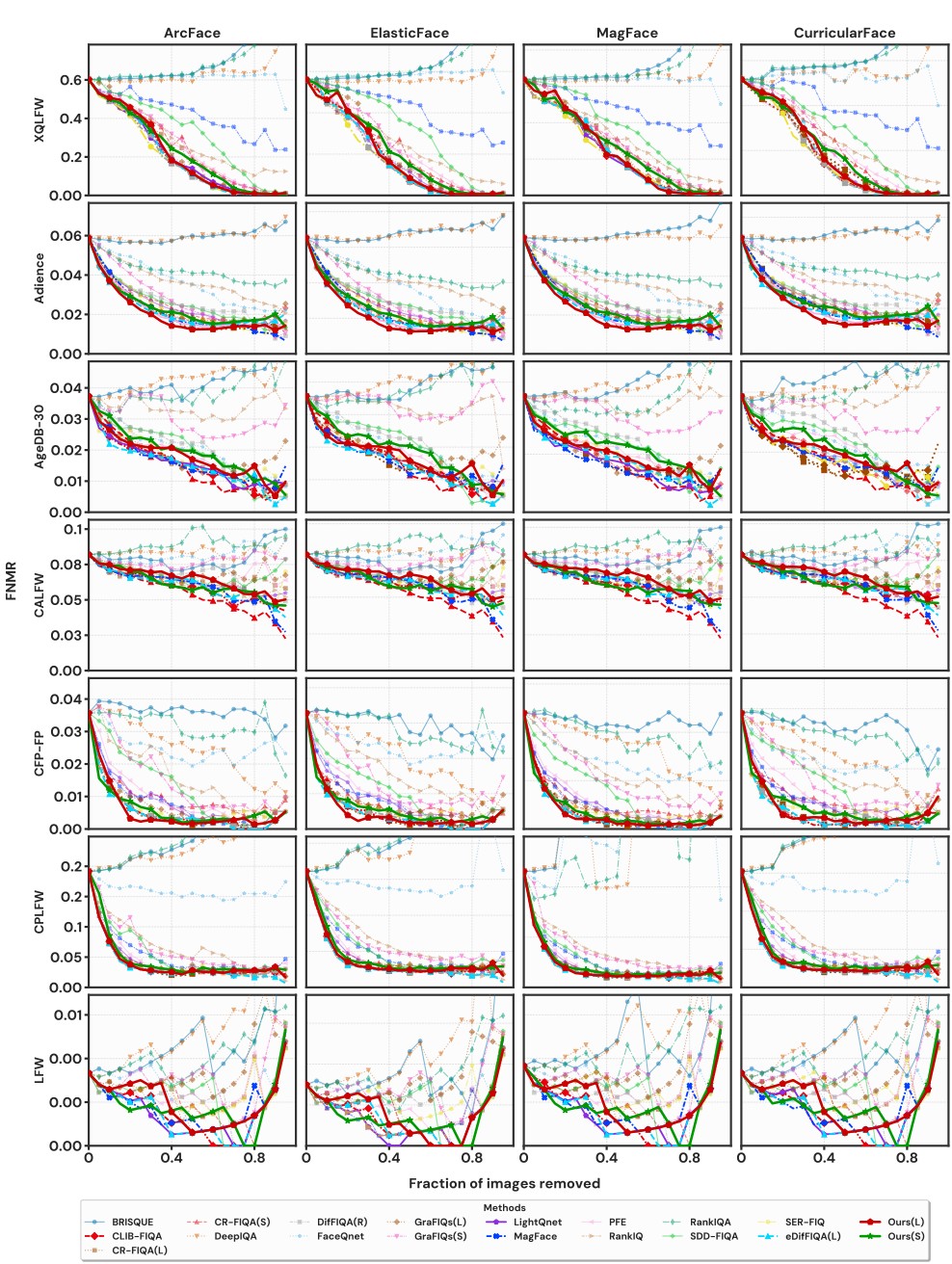

Figure 2: Error-versus-Discard Characteristic (EDC) curves for FNMR@FMR=$1e-3$ of our proposed method *CARPM-FIQA* in comparison to SOTA. Results shown on eight benchmark datasets: LFW Huang et al. (2007), AgeDB-30 Moschoglou et al. (2017), CFP-FP Sengupta et al. (2016), CALFW Zheng et al. (2017), Adience Eidinger et al. (2014), CPLFW Zheng & Deng (2018), XQLFW Knoche et al. (2021), and IJB-C Maze et al. (2018), using ArcFace Deng et al. (2019), ElasticFace Boutros et al. (2022), MagFace Meng et al. (2021a), and CurricularFace Huang et al. (2020) FR models. The proposed *CARPM-FIQA* (S) and *CARPM-FIQA* (L) are marked with solid green and solid red lines, respectively.

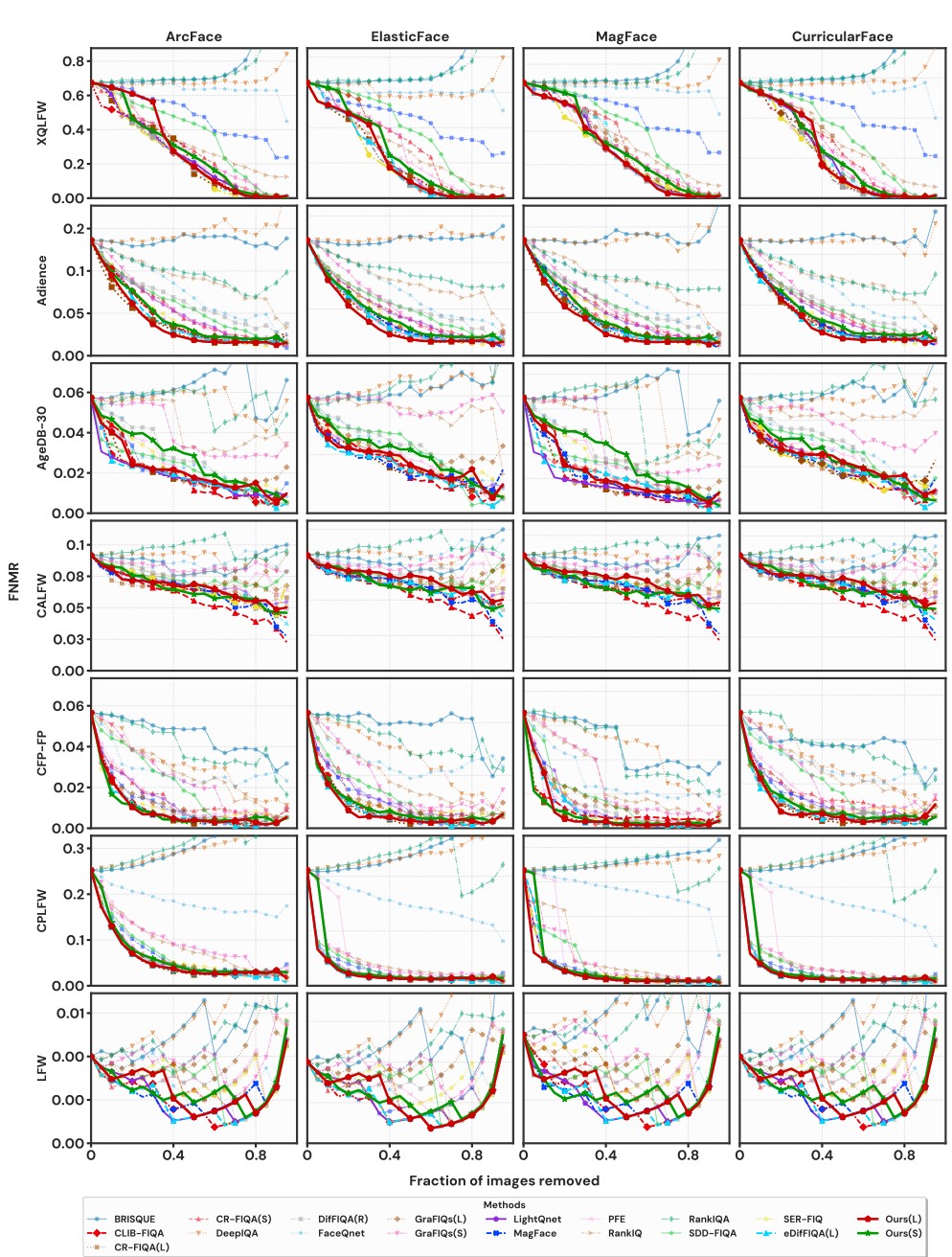

Figure 3: Error-versus-Discard Characteristic (EDC) curves for FNMR@FMR=$1e-4$ of our proposed method *CARPM-FIQA* in comparison to SOTA. Results shown on eight benchmark datasets: LFW Huang et al. (2007), AgeDB-30 Moschoglou et al. (2017), CFP-FP Sengupta et al. (2016), CALFW Zheng et al. (2017), Adience Eidinger et al. (2014), CPLFW Zheng & Deng (2018), XQLFW Knoche et al. (2021), and IJB-C Maze et al. (2018), using ArcFace Deng et al. (2019), ElasticFace Boutros et al. (2022), MagFace Meng et al. (2021a), and CurricularFace Huang et al. (2020) FR models. The proposed *CARPM-FIQA* (S) and *CARPM-FIQA* (L) are marked with solid green and solid red lines, respectively.

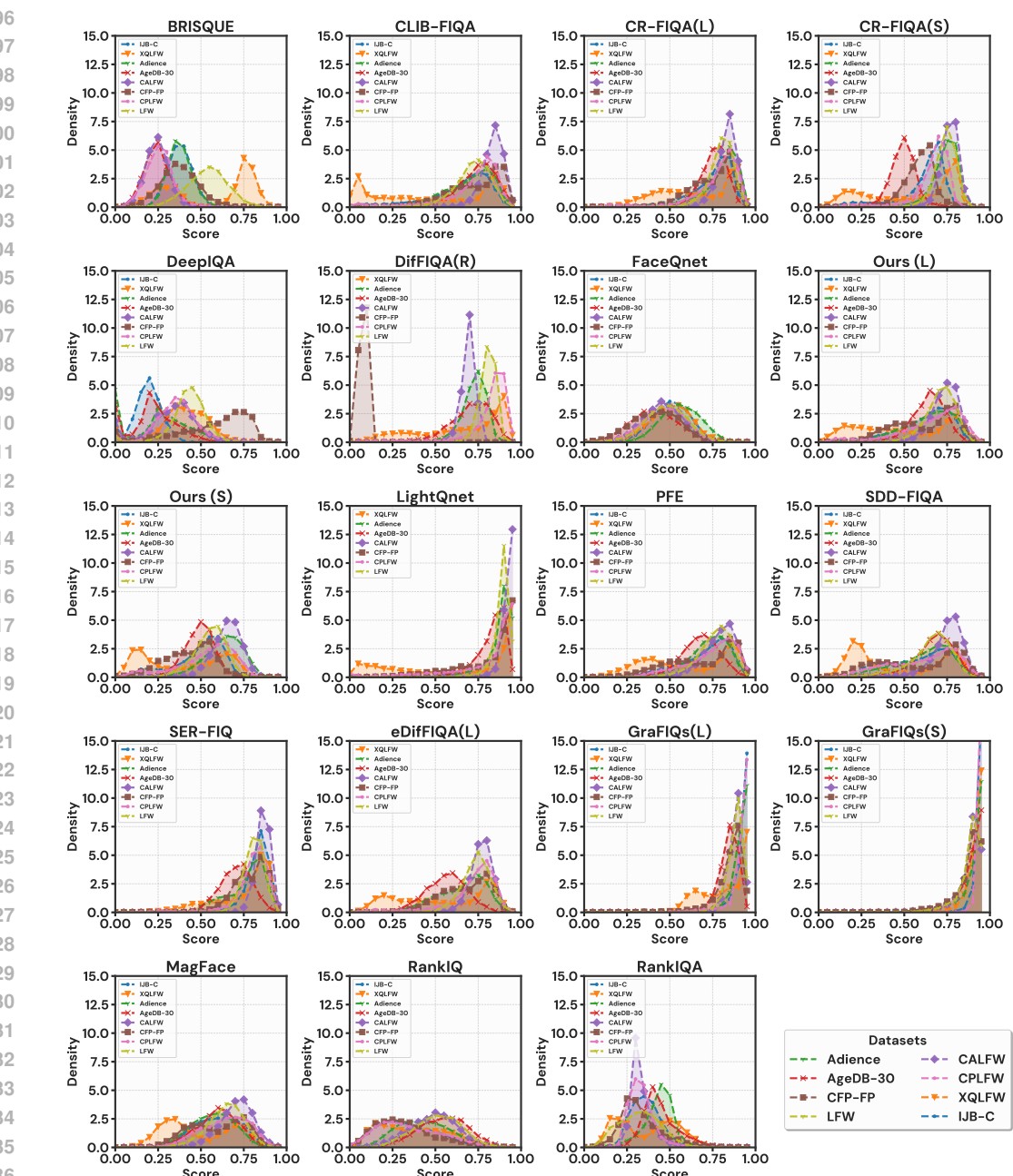

Figure 4: Distribution of quality scores across the evaluation benchmarks, comparing our proposed method (*CARPM-FIQA*) with SOTA methods. All scores are normalized to the range [0, 1].

Table 11: Post-hoc vs. joint training of the quality regression head. AUC-EDC is reported for FMR=$1e{-}3$ and $1e{-}4$ across face recognition benchmarks. "Frozen FR" denotes a frozen pre-trained backbone, while "Cumulative FR" denotes joint online training of FR and quality head with CARPM. Results show that jointly training leveraging embedding dynamics achieves better performance.

| FR Backbone | Training Scheme | Adience Eidinger et al. (2014) | | AgeDB-30 Moschoglou et al. (2017) | | CFP-FP Sengupta et al. (2016) | | LFW Huang et al. (2007) | | CALFW Zheng et al. (2017) | | CPLFW Zheng & Deng (2018) | | XQLFW Knoche et al. (2021) | |
|---|---|---|---|---|---|---|---|---|---|---|---|---|---|---|---|
| | | 1e-3 | 1e-4 | 1e-3 | 1e-4 | 1e-3 | 1e-4 | 1e-3 | 1e-4 | 1e-3 | 1e-4 | 1e-3 | 1e-4 | 1e-3 | 1e-4 |
| ArcFace Deng et al. (2019) | Frozen FR | 0.0231 | 0.0512 | 0.0249 | 0.0317 | 0.0073 | 0.0127 | 0.0029 | 0.0035 | 0.0559 | 0.0592 | 0.0518 | 0.0716 | 0.2653 | 0.3047 |
| | Cumulative FR | 0.0221 | 0.0431 | 0.0190 | 0.0262 | 0.0059 | 0.0084 | 0.0017 | 0.0023 | 0.0587 | 0.0624 | 0.0415 | 0.0608 | 0.2352 | 0.2700 |
| ElasticFace Boutros et al. (2022) | Frozen FR | 0.0248 | 0.0449 | 0.0235 | 0.0249 | 0.0059 | 0.0081 | 0.0025 | 0.0033 | 0.0534 | 0.0547 | 0.0458 | 0.0724 | 0.2268 | 0.2827 |
| | Cumulative FR | 0.0235 | 0.0403 | 0.0184 | 0.0197 | 0.0056 | 0.0077 | 0.0014 | 0.0023 | 0.0569 | 0.0585 | 0.0392 | 0.0634 | 0.2232 | 0.2688 |
| MagFace Meng et al. (2021a) | Frozen FR | 0.0236 | 0.0498 | 0.0274 | 0.0458 | 0.0091 | 0.0127 | 0.0030 | 0.0036 | 0.0552 | 0.0571 | 0.0534 | 0.1189 | 0.2996 | 0.3259 |
| | Cumulative FR | 0.0226 | 0.0426 | 0.0214 | 0.0389 | 0.0070 | 0.0088 | 0.0017 | 0.0023 | 0.0579 | 0.0590 | 0.0422 | 0.0974 | 0.2666 | 0.3143 |
| CurricularFace Huang et al. (2020) | Frozen FR | 0.0216 | 0.0430 | 0.0262 | 0.0310 | 0.0079 | 0.0118 | 0.0029 | 0.0035 | 0.0547 | 0.0569 | 0.0430 | 0.0756 | 0.2222 | 0.2736 |
| | Cumulative FR | 0.0210 | 0.0380 | 0.0198 | 0.0228 | 0.0070 | 0.0100 | 0.0017 | 0.0023 | 0.0579 | 0.0607 | 0.0370 | 0.0670 | 0.2078 | 0.2346 |

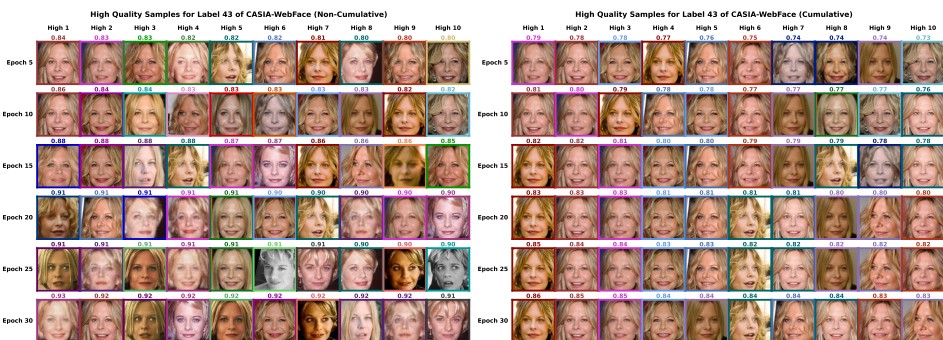

Figure 5: High quality samples from identity #43 across training epochs. **Left:** Non-cumulative quality scores show per-epoch quality estimates. **Right:** Cumulative quality scores average the quality over all epochs, producing more stable rankings. Images maintain consistent color borders across plots to track their position changes. Higher scores indicate better quality face samples.

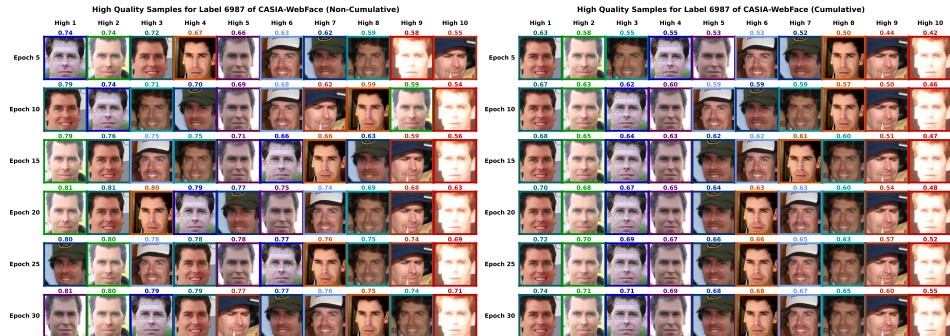

Figure 6: High quality samples from identity #6987 across training epochs. This identity has fewer unique images (15 total). Note how the top-ranked images in the non-cumulative approach (left) tend to stabilize in the cumulative approach (right), showing the benefit of averaging quality scores over time.

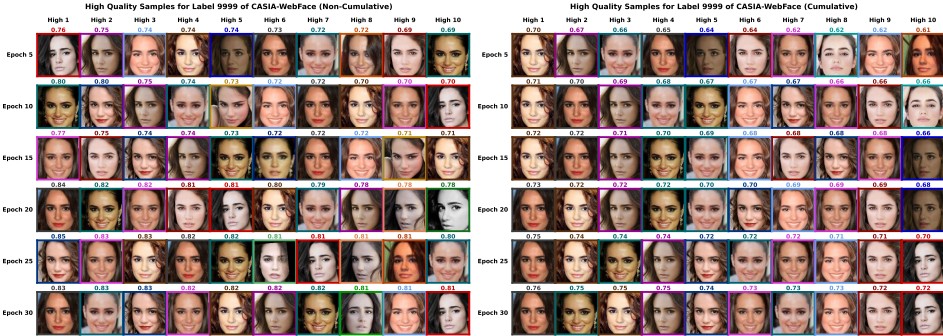

Figure 7: High quality samples from identity #9999 across training epochs. The consistent color coding allows tracking specific images as they move through the rankings. Observe how some images consistently rank high across epochs, while others show greater variability, particularly in the non-cumulative scores (left).

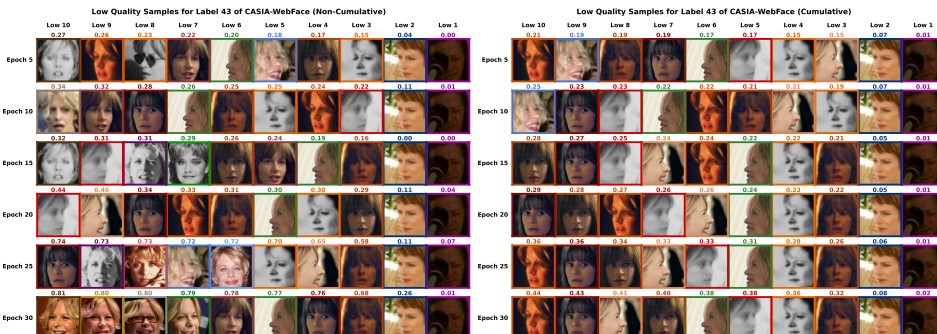

Figure 8: Low quality samples from identity #43 across training epochs. Images are ordered from higher to lower quality (left to right), with the lowest quality samples appearing on the right side. These samples typically exhibit more challenging characteristics such as extreme poses, occlusions, or poor lighting. Notice how the lowest quality samples tend to remain consistent across epochs, particularly in the cumulative approach (right).

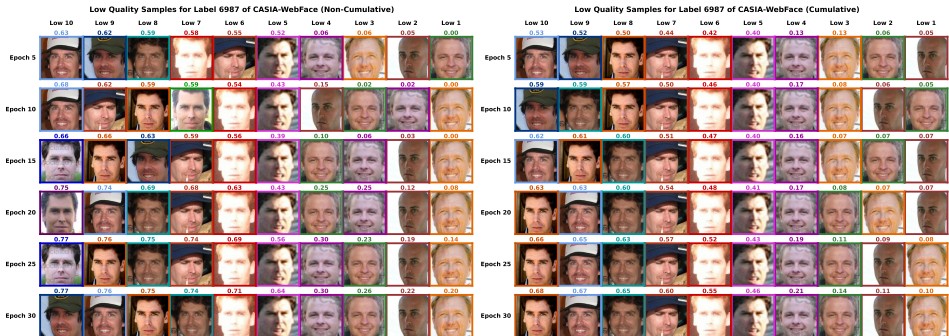

Figure 9: Low quality samples from identity #6987, having few unique images, across training epochs. The non-cumulative approach (left) shows more variability in rankings compared to the more stable cumulative approach (right). The lowest quality samples (rightmost columns) exhibit mislabeled samples that make facial recognition more challenging.

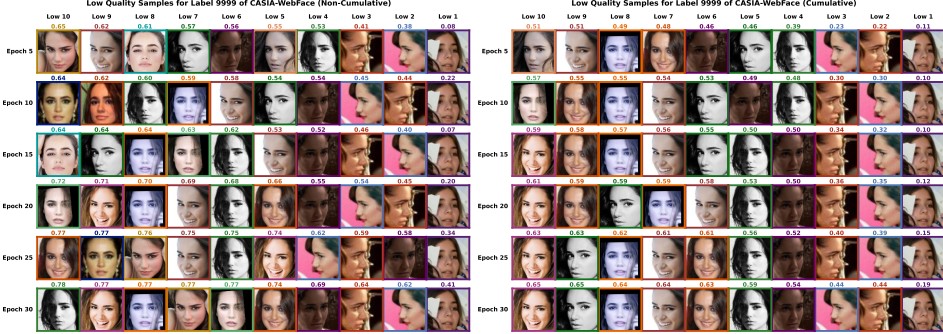

Figure 10: Low quality samples from identity #9999 across training epochs. The consistent color borders across both non-cumulative (left) and cumulative (right) visualizations help track how the quality scores of individual images evolve through training. Lower quality samples typically exhibit characteristics that make them harder to recognize, such as unusual expressions, poor lighting, or non-frontal poses.

Table 12: Post-hoc vs. joint training of the quality regression head. pAUC-EDC is reported for FMR=$1e-3$ and $1e-4$ across face recognition benchmarks. "Frozen FR" denotes a frozen pre-trained backbone, while "Cumulative FR" denotes joint online training of FR and quality head with CARPM. Results show that jointly training leveraging embedding dynamics achieves better performance.

| FR Backbone | Training Scheme | Adience Eidinger et al. (2014) | | AgeDB-30 Moschoglou et al. (2017) | | CFP-FP Sengupta et al. (2016) | | LFW Huang et al. (2007) | | CALFW Zheng et al. (2017) | | CPLFW Zheng & Deng (2018) | | XQLFW Knoche et al. (2021) | |
|---|---|---|---|---|---|---|---|---|---|---|---|---|---|---|---|
| | | 1e-3 | 1e-4 | 1e-3 | 1e-4 | 1e-3 | 1e-4 | 1e-3 | 1e-4 | 1e-3 | 1e-4 | 1e-3 | 1e-4 | 1e-3 | 1e-4 |
| ArcFace Deng et al. (2019) | Frozen FR | 0.0111 | 0.0281 | 0.0074 | 0.0087 | 0.0051 | 0.0083 | 0.0006 | 0.0007 | 0.0192 | 0.0216 | 0.0202 | 0.0321 | 0.1236 | 0.1404 |
| | Cumulative FR | 0.0093 | 0.0236 | 0.0072 | 0.0111 | 0.0034 | 0.0053 | 0.0006 | 0.0008 | 0.0184 | 0.0204 | 0.0208 | 0.0347 | 0.1255 | 0.1383 |
| ElasticFace Boutros et al. (2022) | Frozen FR | 0.0122 | 0.0248 | 0.0074 | 0.0083 | 0.0042 | 0.0061 | 0.0006 | 0.0007 | 0.0181 | 0.0191 | 0.0190 | 0.0305 | 0.1141 | 0.1315 |
| | Cumulative FR | 0.0103 | 0.0210 | 0.0069 | 0.0074 | 0.0031 | 0.0046 | 0.0005 | 0.0008 | 0.0176 | 0.0183 | 0.0197 | 0.0420 | 0.1262 | 0.1515 |
| MagFace Meng et al. (2021a) | Frozen FR | 0.0113 | 0.0276 | 0.0085 | 0.0128 | 0.0063 | 0.0113 | 0.0007 | 0.0009 | 0.0189 | 0.0197 | 0.0219 | 0.0618 | 0.1305 | 0.1486 |
| | Cumulative FR | 0.0096 | 0.0231 | 0.0077 | 0.0164 | 0.0046 | 0.0059 | 0.0006 | 0.0008 | 0.0181 | 0.0185 | 0.0227 | 0.0693 | 0.1404 | 0.1611 |
| CurricularFace Huang et al. (2020) | Frozen FR | 0.0097 | 0.0230 | 0.0083 | 0.0105 | 0.0053 | 0.0076 | 0.0006 | 0.0008 | 0.0186 | 0.0202 | 0.0172 | 0.0328 | 0.1127 | 0.1286 |
| | Cumulative FR | 0.0082 | 0.0194 | 0.0075 | 0.0093 | 0.0039 | 0.0058 | 0.0006 | 0.0008 | 0.0178 | 0.0193 | 0.0181 | 0.0458 | 0.1155 | 0.1300 |

Table 13: Comparison of training conditions across FIQA methods

| Method | Training Dataset | Architecture | Additional Components | Supervision Type | Loss Functions |
|---|---|---|---|---|---|
| **General Image Quality Assessment (IQA) Methods** | | | | | |
| BRISQUEMittal et al. (2012) | LIVE IQA | Feature-based (Non-DL) | Spatial domain NSS features | Regression to DMOS | SVR with RBF kernel |
| RankIQALiu et al. (2017b) | Waterloo + Places2 | VGG-16 pretrained on ImageNet | Siamese network with efficient backpropagation | Self-supervised ranking + supervised fine-tuning | Pairwise ranking hinge loss + squared Euclidean distance |
| DeepIQABosse et al. (2018) | LIVE IQA / TID2013 / CLIVE | VGG-inspired CNN (14 weight layers) | Siamese network with feature fusion (FR) or single-branch CNN (NR) | End-to-end supervised learning with patch sampling | Mean Absolute Error for simple averaging / Weighted MAE for weighted averaging |
| **Face Image Quality Assessment (FIQA) Methods** | | | | | |
| RankIQChen et al. (2015) | FERET, FRGC, Chinese ID photos, LFW, AFLW | Feature fusion approach | CNN for landmark localization + multi-feature fusion (HoG, Gabor, Gist, LBP, CNN) | Learning to rank with relative quality constraints | Max-margin convex optimization with slack variables |
| PFEShi & Jain (2019) | refined MS-Celeb-1M | ResNet-64 | Uncertainty module (2 FC layers) producing Gaussian distributions | Two-stage: pretrained FR model + uncertainty estimation | Negative mutual likelihood score minimization for genuine pairs |
| SER-FIQTerhörst et al. (2020) | N/A (training-free) | ResNet-100 (pretrained) | Monte Carlo dropout (m=100 stochastic forward passes) | Unsupervised, training-free | N/A (directly computes quality from embedding variations) |
| FaceQnetHernandez-Ortega et al. (2019; 2020) | VGGFace2 | ResNet-50 | Two FC layers (2048→32→1) + Dropout (v1) | Supervised using FR scores between ICAO-compliant images and test images | Mean Squared Error (MSE) regression loss |
| MagFaceMeng et al. (2021a) | MS1MV2 | ResNet-100 | No additional networks (uses feature norm as quality) | Identity classification (no explicit quality labels) | Modified ArcFace with adaptive margin m(a) and magnitude regularizer g(a) |
| SDD-FIQAOu et al. (2021) | MS1MV2 | ResNet-50 | Separate quality regression network | Unsupervised with quality pseudo-labels from similarity distributions | Huber loss |
| CR-FIQAuS(Boutros et al. (2023) | CASIA-WebFace | ResNet-50 | Single regression layer for quality prediction | Unsupervised sample relative classifiability estimation | Combined loss: ArcFace + Smooth L1-Loss for certainty ratio prediction |
| CR-FIQAuL(Boutros et al. (2023) | MS1MV2 | ResNet-100 | Single regression layer for quality prediction | Unsupervised sample relative classifiability estimation | Combined loss: ArcFace + Smooth L1-Loss for certainty ratio prediction |
| DifFIQA(R)Babnik et al. (2023b) | VGGFace2 | DDPM(UNet) + ResNet-100 (pretrained) | Quality regression head added to FR model | Supervised regression using distilled DifFIQA labels | Mean Squared Error (MSE) |
| eDifFIQA(L)Babnik et al. (2024) | VGGFace2 | DDPM(UNet) + ResNet-100 (pretrained) | Teacher model trained on VGGFace2 | Supervised regression using distilled DifFIQA labels | Combined loss: distilled DifFIQA labels |
| GRAFIQSUS(Kolf et al. (2024) | N/A (training-free) | ResNet-50 (pretrained) | No additional networks (uses batch normalization statistics) | Unsupervised, training-free | N/A (calculates quality from BN statistics) |
| GRAFIQSUL(Kolf et al. (2024) | N/A (training-free) | ResNet-100 (pretrained) | No additional networks (uses batch normalization statistics) | Unsupervised, training-free | N/A (calculates quality from BN statistics) |
| CLIB-FIQAOu et al. (2024) | MS1MV2 | CLIP + ResNet-50 | Vision-language alignment model with MLP confidence calibration | Joint learning with supervised quality fitting | EMD loss for quality fitting + Focal loss for quality factors |
| CARPM-FIQA(S) | CASIA-WebFace | ResNet-50 | Quality regression layer for accumulated quality prediction | Unsupervised with temporally accumulated relative point margin scores | Combined loss: ArcFace + Smooth L1-Loss for quality prediction |
| CARPM-FIQA(L) | MS1MV2 | ResNet-100 | Quality regression layer for accumulated quality prediction | Unsupervised with temporally accumulated relative point margin scores | Combined loss: ArcFace + Smooth L1-Loss for quality prediction |

