# OpenReview forum: "Learning Steadily: Accumulating Relative Point Margin Scores for Face Image Quality Assessment"
_ICLR.cc/2026/Conference — ICLR 2026 Conference Withdrawn Submission_

### Official Review · Reviewer_y6bK · 2025-10-31

**Soundness:** 3
**Presentation:** 2
**Contribution:** 2
**Rating:** 6
**Confidence:** 3

**Summary:**

Aiming at the problem in Face Image Quality Assessment (FIQA) where existing methods integrated with Face Recognition (FR) training suffer from unstable quality targets due to the dynamic evolution of the feature space, this paper proposes the CARPM-FIQA method. Its core lies in quantifying the ratio between intra-class compactness and inter-class separation of samples by accumulating Relative Point Margin (RPM) scores across training epochs, thereby obtaining a temporally stable quality metric. Meanwhile, it extends the FR architecture by adding a regression layer to enable quality prediction for unseen images. Through experiments on 8 benchmark datasets (such as LFW and IJB-C) and 4 state-of-the-art FR models, CARPM-FIQA consistently ranks among the top-performing methods, with particularly outstanding performance on datasets with significant quality variations. It not only maintains tight integration with the FR pipeline but also addresses the instability issue of traditional non-cumulative methods.

**Strengths:**

1. Innovative and Theoretically Supported Cumulative Strategy
By accumulating Relative Point Margin (RPM) scores across epochs, the method is theoretically proven to have advantages such as unbiasedness, variance decreasing in inverse proportion to the number of epochs, and smaller mean squared error (MSE). It effectively addresses the issue of fluctuations in quality targets caused by the dynamic feature space .
2. Comprehensive Experimental Design
It covers 8 representative benchmark datasets, compares with 14 mainstream IQA/FIQA methods, and simultaneously verifies the impact of different training protocols (small/large-scale datasets, different loss functions), resulting in highly reliable experimental results.

**Weaknesses:**

1.Cumulative RPM scores may lead to excessive resource consumption. FIQA and FR tasks are relatively lightweight, yet the paper uses 4 Nvidia HGX A100 GPUs and 256-core CPUs for training. Thus, the practical application feasibility of the method remains questionable.
2.The compared methods are relatively outdated, with the latest one being from 2024. It is recommended to supplement comparisons with more up-to-date methods, such as CR-FIQA: Face Image Quality Assessment by Learning Sample Relative Classifiability (ICCV Workshop 2025). Additionally, the number of FR backbones selected for comparison is relatively small.
3.Performance bottleneck on cross-attribute datasets: On datasets such as CALFW, CPLFW, and IJB-C, the performance improvement is limited, and it even occasionally performs slightly worse than non-cumulative methods, showing insufficient adaptability to extreme attribute differences.
4.There is an issue with the formatting of the article, especially with the small font size of the comparison experiment results table. Please confirm if it complies with the ICLR submission standards.

**Questions:**

1.In resource-constrained scenarios (such as edge devices), is the training duration and inference speed of CARPM-FIQA competitive?
2.Is there an optimal range for the number of accumulated epochs? Will an excessive number of epochs introduce outdated feature space information and lead to quality assessment bias?
3.The cumulative method shows limited effectiveness on cross-attribute datasets such as CALFW. What is the reason for this? Does this indicate that the method has certain limitations in generalization performance?

---

### Official Review · Reviewer_4LVa · 2025-11-02

**Soundness:** 2
**Presentation:** 3
**Contribution:** 2
**Rating:** 4
**Confidence:** 2

**Summary:**

Face image quality assessment (FIQA) is a critical topic for biometric systems. However, this manuscript does not meet the standards for publication due to fundamental gaps in methodological novelty clarification, theoretical rigor, experimental completeness, and comparative analysis. The core idea of "accumulated relative point margin scores" shows potential, but the submission fails to substantiate its contributions, validate key design choices, or contextualize the work against state-of-the-art (SOTA) FIQA methods. Below are detailed concerns.

The manuscript claims CARPM-FIQA is a "novel approach" to address unstable quality targets in FR-integrated FIQA. However, it provides no clear distinction from prior work on temporal or cumulative quality estimation in FIQA. For example, methods like CR-FIQA (2021) and SER-FIQ (2020) already leverage sample-specific margin-based metrics to link quality to FR performance—yet the submission does not explain how CARPM-FIQA’s "accumulated across epochs" mechanism differs from these static or single-epoch margin-based methods.
The term "accumulated relative point margin scores" is not defined relative to existing metrics (e.g., inter/intra-class margins in FR loss functions like ArcFace or CosFace). Without this context, it is impossible to assess whether the core innovation is incremental or transformative.

**Strengths:**

Propose a novel approach that addresses this limitation by learning to predict accumulated relative point margin scores across training epochs rather than relying on single-epoch estimates.

Quantify image quality by tracking and accumulating the ratio between intra-class compactness and inter-class separation for each sample throughout training, providing a temporally-stable measure of sample utility.

Demonstrate theoretically and empirically that this cumulative averaging approach significantly reduces variance in quality estimates and improves reliability in sample ranking compared to non-cumulative alternatives.

**Weaknesses:**

It claims CARPM-FIQA "consistently ranks among top-performing methods" but does not name the SOTA FIQA methods it is compared against (e.g., FaceQNet, DifFIQA, SDD-FIQA). Without these comparisons, there is no way to judge its competitiveness.

No numerical metrics (e.g., AUC, pAUC, Spearman’s rank correlation with FR accuracy) are provided. For example, how much does the cumulative approach reduce variance in quality estimates? What percentage improvement in sample ranking reliability is observed?

How is the regression layer trained? For example, are accumulated scores from training samples used as labels, and if so, how are these labels normalized to avoid bias from varying epoch counts across training runs?

There is no discussion of generalization: Does the regression layer overfit to the accumulated scores of the training dataset, or can it reliably predict quality for unseen datasets with different quality distributions (e.g., low-light vs. occluded faces)?

**Questions:**

Expand experiments to include explicit comparisons with SOTA FIQA baselines, quantitative metrics, ablation studies for the cumulative component, and full details of FR models used.

Clarify implementation details for the regression layer and evaluate generalization across diverse quality scenarios.

---

### Official Review · Reviewer_2B9N · 2025-11-04

**Soundness:** 2
**Presentation:** 2
**Contribution:** 2
**Rating:** 2
**Confidence:** 3

**Summary:**

The paper presents a method for accumulating relative point margin scores for improved face image quality assessment. The proposed method is shown to perform better than single-epoch estimates. The approach relies on an additive noise model for the epoch-wise quality estimates and uses this assumption for a theoretical analysis.

**Strengths:**

1. The cumulative average approach is interesting and works with standard datasets.
2. The regression model helps with generalization.

**Weaknesses:**

1. The assumptions in the work are unrealistic. For example, any gradient-based parameter update strategy produces a sequence of correlated parameters (by definition). The assumptions become difficult to justify after the model has trained for even a few epochs. The subsequent theoretical analysis is based on these assumptions, which leads to questions about the validity of the results.
2. The theoretical analysis is fairly straightforward for an additive noise model with uncorrelated noise. Therefore, it offers little by way of novelty.
3. The gains offered by the proposed approach are modest and not always consistent.
4. The paper is difficult to read. The typesetting makes it even harder with cramped bullet items, small tables & figures.
5. The quality of the exposition can be improved.

**Questions:**

1. Can you please provide a strong justification for the assumptions?
2. Can you please highlight the instability issue with the single-epoch approach?
3. Can you please explain why the performance of the proposed approach is not much improved compared to a simpler NR-IQA method like BRISQUE?
4. Can you please explain the gains offered by your method in the feature space?

---

### Official Review · Reviewer_sJms · 2025-11-05

**Soundness:** 2
**Presentation:** 2
**Contribution:** 2
**Rating:** 2
**Confidence:** 4

**Summary:**

This paper presents a facial image quality assessment method called CARPM-FIQA, to mitigate the instability resulted from feature space variations during the FR model training, by accumulating the interval scores of each sample during the training process. The authors conducted extensive experiments to validate the performance of their method, claiming that it reduces the variance in facial image quality assessment and enhances the reliability of ranking.

**Strengths:**

1. In their method, the authors introduced the cumulative score margins of samples, taking into account the variations of the feature space during the training.
2. The authors conducted extensive experiments to validate their method.

**Weaknesses:**

1.	I am skeptical about whether it is worthwhile to invest such significant effort into developing methods for assessing facial image quality. I believe more focus should be placed on developing FR methods that are more broadly applicable, rather than on facial image quality assessment methods. While the methods for assessing facial image quality may assist FR models in filtering out certain extreme facial image samples, integrating them into the training of FR models seems somewhat misguided to me.
2.	Facial recognition or facial verification involves comparing the feature similarity of pairs of facial samples to determine whether they exceed a certain threshold; if they do, the paired facial samples/images are considered to come from the same person, otherwise not. In other words, FR checks the absolute value of the feature similarities of pairs of facial samples. The "silhouette score" defined by the author in Equations 1 and 5 is based on the difference in feature similarities between a sample and its within-class samples and between-class samples, making it a relative value. Therefore, I am doubtful that one can accurately assess whether a facial image is suitable for FR based on the silhouette score.
3.	I believe that "intra-class compactness" and "inter-class separability" express the global properties of sample features, at least the attributes of features for all samples within a class. Therefore, I don’t think that the author's assertion that a(i) indicates intra-class compactness and b(i) reflects inter-class separation is accurate.
4.	In a sufficiently large feature space, even if s(i) is close to 0, the features of the sample may be far from the decision boundary.
5.	After redefining the silhouette score using feature cosine similarity in Eq. 5, some of its fundamental properties have changed. The author seems to overlook the distinction between similarity and distance.
6.	Equation 7 redundantly defines the concept as Equations 4 and 5.
7.	The expression in Equation 8 is unclear, as it does not include f(x). The mathematical notation in the method section of the paper is chaotic and inconsistent.
8.	In the discussion around page 5 of the paper, the author emphasizes the characteristics of the proposed method while neglecting the assessment of image quality.
9.	The figures, tables, and text in the experimental section are overly dense, making them difficult to read.

**Questions:**

Could a facial image quality assessment method developed alongside a FR model be applied to other FR models?

---

### Official Review · Reviewer_kE1n · 2025-11-08

**Soundness:** 3
**Presentation:** 2
**Contribution:** 2
**Rating:** 4
**Confidence:** 4

**Summary:**

The paper proposes using a Cumulative Average of the Relative Point Margin (CARPM) as a stable target value for training face image quality assessment (IQA) integrated with Face Recognition (FR) models. The motivation is to counteract the inherent fluctuations and instability of instantaneous quality metrics (like the relative point margin) that arise from the dynamically evolving feature space during training. The authors demonstrate that this simple averaging approach significantly reduces variance in quality estimates, leading to more reliable sample ranking. The approach is validated through extensive experiments demonstrating effectiveness and robustness across four SOTA FR models and eight benchmark datasets.

**Strengths:**

1. **Originality & Significance (Problem Formulation)**: The paper addresses a critical, under-explored limitation of FR-integrated IQA methods: the instability of dynamic target values stemming from evolving feature clusters. The proposed cumulative averaging approach (CARPM) is a simple yet highly effective solution to this problem, offering a practical way to stabilize training and improve the reliability of the quality metric.

2. **Quality (Theoretical Justification)**: The paper includes a commendable effort to provide theoretical analysis explaining the mechanism by which cumulative averaging smooths fluctuations and stabilizes the training process, thereby grounding the empirical results in a formal context.

3. **Quality & Clarity (Empirical Evidence)**: The experimental section is extensive and thorough. The inclusion of an ablation study directly comparing the cumulative and non-cumulative methods (in terms of target distribution, FR performance metrics, and illustrated quality score stability/ranking) provides compelling evidence of the approach's effectiveness and clearly demonstrates the benefits of CARPM.

**Weaknesses:**

## Major: Limitation Analysis and Alternative Design Space

The major weakness lies in the lack of critical analysis of the inherent limitations of simple cumulative averaging. While effective for stabilization, the proposed method suffers from significant conceptual drawbacks that are not addressed:

- **Slow Adaptation/Inertia**: Cumulative averaging gives equal weight to all past epochs (1/n), meaning the current quality signal is heavily diluted by outdated information. This introduces significant lag and may prevent the model from rapidly adapting to crucial, recent breakthroughs in the feature space.

- **Design Rationale vs. Alternatives**: The paper does not discuss or justify the choice of simple cumulative averaging over well-established alternatives for time-series smoothing in dynamic optimization. For instance, methods like the Exponential Moving Average (EMA) or techniques that apply decaying weights to older data are standard solutions that achieve stabilization while allowing for faster adaptation to recent, high-quality information. A discussion and analysis of these alternatives is essential to fully understand the limitations and optimal design space of the proposed stabilization method.

## Minor: Presentation and Clarity

- **Presentation Quality**: Several tables and figures are overly compressed and difficult to interpret at a normal viewing scale, requiring significant zooming. Improving the visual presentation and readability of the main results is necessary.

- **Section Structure/Pacing**: Section 2 dedicates an unusually large portion of space to background explanations and the derivation of the proposed approach, which, given the simplicity of the CARPM formula, feels compressed and unbalanced relative to the empirical results and analysis. The authors should consider streamlining the background to dedicate more space to the experimental analysis or discussion of limitations.

**Questions:**

1. **Comparison to Traditional IQA**: Please clarify the motivation for comparing the proposed integrated approach against traditional, non-FR-integrated IQA methods. Since your method's target is derived directly from the FR training process (relative point margin), it is fundamentally different from general-purpose IQA metrics. Are the standard IQA comparisons purely for completeness, or is there a specific, actionable insight derived from these comparisons?

2. **Alternative Smoothing Techniques** (EMA): Could the authors discuss and ideally experimentally evaluate the use of an Exponential Moving Average (EMA) instead of simple cumulative averaging? EMA is a common technique that achieves stabilization while allowing for faster adaptation by exponentially prioritizing more recent quality scores. This analysis would directly address the slow adaptation limitation of CARPM.

3. **Illustrative Samples** (Initial Epochs): Regarding the illustrated samples of quality ranking stability (Figures 5-10), it would significantly improve the clarity of the initial effect if the authors could show the ranking at Epoch 1 for both the cumulative and non-cumulative approaches. For a truly fair comparison of the stabilization effect, the initial rankings should be similar before the averaging process takes place. Showing the first epoch would help confirm that the non-cumulative approach is indeed noisy from the start and that the difference observed is due to smoothing, not an initial bias.

---

### Note · Authors · 2025-11-13

I have read and agree with the venue's withdrawal policy on behalf of myself and my co-authors.